# AdaRoPE: Not All Attention Heads Should Rotate and Scale Equally

**Shaowen Wang** [1 2 3 *] **Yuke Zheng** [1 *] **Tansheng Zhu** [1 3 *] **Shuang Chen** [2 *] **Shaofan Liu** [4] **Suncong Zheng** [2]
**Jian Li** [1 †]

## Abstract

Rotary Position Embedding (RoPE) is widely adopted in Transformers to encode positional information, yet standard implementations enforce a uniform frequency schedule and scaling across all attention heads. Using simplified retrieval tasks and length generalization scenarios, we show—both empirically and theoretically—that heads with different functional roles require distinct frequency ranges and attention scaling factors to operate effectively. Ignoring this structure leads to suboptimal utilization of embedding dimensions and degraded performance, particularly under long-context settings. To address these limitations, we propose AdaRoPE, which equips each attention head with learnable rotation frequencies and attention scaling factors. Pretrained LLMs with AdaRoPE consistently outperform existing RoPE variants, including partial RoPE and NoPE baselines. For context extension, we further show that uniform frequency and attention scaling, used in methods such as YaRN, are suboptimal. By applying head-specific scaling, AdaRoPE enables better context extension while better preserving short-context performance in both the extrapolation setting and the long-context continued pretraining setting. These results highlight the importance of optimizing rotary position embedding at the level of individual attention heads.

---

 * Core Contributor; † Corresponding author. Work done during Shaowen Wang's internship at Tencent. [1]Tsinghua University [2]Hunyuan Team, Tencent [3]Xiongan AI Institute [4]Fudan University. Correspondence to: Shaowen Wang <wangsw23@mails.tsinghua.edu.cn>, Suncong Zheng <congzheng@tencent.com>, Jian Li <lijian83@mail.tsinghua.edu.cn>.

*Proceedings of the $43^{rd}$ International Conference on Machine Learning*, Seoul, South Korea. PMLR 306, 2026. Copyright 2026 by the author(s).

## 1. Introduction

Large models based on Transformer (Vaswani et al., 2017) architectures have achieved remarkable success. For the self-attention module, explicit positional embedding is critical for enhancing model performance (Shaw et al., 2018; Dai et al., 2019; Su et al., 2021).

Rotary Position Embedding (RoPE) (Su et al., 2021) has become the de facto positional embedding for large models (Grattafiori et al., 2024; Yang et al., 2025a; Team et al., 2025). It works by applying a rotation matrix to each token's keys and queries, where the rotation angle depends on the token's absolute position in the sequence. When two tokens interact through the attention inner product, their rotated representations naturally encode their relative distance.

Current standard RoPE and its popular extensions such as YaRN (Peng et al., 2023), Log Scaling in Bai et al. (2023), and Llama3 Scheme (Grattafiori et al., 2024), typically use a shared frequency schedule and attention scaling factor across all heads. However, this uniform assignment presents two significant limitations:

First, the uniform coupling of rotation frequencies to attention dimensions across all heads in RoPE can lead to underutilization of the model's embedding dimensions. Because each head shares the same frequency schedule, heads that rely on specific frequency bands may leave other dimensions underutilized, resulting in a waste of attention representational capacity (Chiang & Yogatama, 2025). For instance, semantic heads, which only encode non-positional information, primarily utilize the slowest frequencies (Barbero et al., 2025), whereas slash heads (Cheng et al., 2026), which consistently attend to tokens at an approximately constant lag, utilize high frequencies.

Second, in the length extrapolation stage, prevalent methods such as YaRN (Peng et al., 2023) and ABF (Xiong et al., 2024) apply uniform adjustments: they scale RoPE frequencies identically across all attention heads. This approach overlooks the conflicting geometric needs of different head types. For instance, while global heads that aggregate long-range context require positional scaling to extend their range (Chen et al., 2023a), applying the same scaling to off-by-one heads (Cheng et al., 2026) distorts the

precise rotational alignment required for short-range dependencies, causing them to drift focus from the immediate previous token to positions further back in the sequence.

Furthermore, self-attention in long contexts suffers from attention dilution, where the weights for relevant tokens are drowned out by many irrelevant ones in long sequences (Zhang et al., 2024a; Chiang & Cholak, 2022; Li et al., 2025b). Previous works to mitigate this, such as adjusting the attention temperature with a global, length-dependent factor (Bai et al., 2023; Anson et al., 2025), also employ a uniform strategy. However, because heads prioritize distinct RoPE frequencies, they exhibit inherently varying degrees of recency bias. Consequently, a single scaling factor cannot optimally compensate for this heterogeneous attention dilution across the model.

To address these limitations, we propose **AdaRoPE**, a method that jointly learns head-wise RoPE rotation frequencies and length-dependent attention scaling. Our main contributions are summarized as follows:

- **Theoretical Analysis of Head Heterogeneity.** We construct three simplified yet representative scenarios covering retrieval tasks and length extrapolation, and conduct theoretical analyses on these cases to demonstrate that different attention heads require distinct rotation frequencies and attention scaling factors.

- **AdaRoPE.** We propose AdaRoPE, a simple yet effective lightweight extension of RoPE that equips each attention head with learnable rotation frequencies and attention scaling. AdaRoPE is designed to be plug-and-play, easily integrable into existing architectures, and can be directly applied to RoPE-pretrained models for long-context extrapolation.

- **Superior Pre-training Performance.** Through extensive experiments on models scaling up to 2.7B parameters trained on 100B FineWeb tokens, we demonstrate that AdaRoPE consistently outperforms prevalent positional encodings.

- **Effective Context Extension.** We demonstrate that AdaRoPE effectively extends pre-trained RoPE models (e.g., Llama-8B) to longer contexts. Compared to YaRN, AdaRoPE achieves superior performance in both zero-shot extrapolation and long-context continued pre-training settings without compromising short context capability.

## 2. Preliminaries and Related Work

### 2.1. Rotary Position Embedding (RoPE)

RoPE (Su et al., 2021) encodes position information by rotating the query and key vectors in $d/2$ independent 2D subspaces. Specifically, we partition the $d$-dimensional space into $d/2$ pairs of elements. For each subspace index $f \in \{0, \ldots, d/2 - 1\}$, we define a frequency $\theta_f = b^{-2f/d}$ with a constant $b > 0$ (typically $10,000$).

For a token at position $i$, the rotation matrix $\boldsymbol{R}_i \in \mathbb{R}^{d \times d}$ is a block-diagonal matrix, formed by $d/2$ rotation sub-matrices:

$$\boldsymbol{R}_i = \mathrm{diag}\left(\boldsymbol{G}_i(\theta_0), \ldots, \boldsymbol{G}_i(\theta_{d/2-1})\right),$$

$$\text{where } \boldsymbol{G}_i(\theta_f) = \begin{pmatrix} \cos(i\theta_f) & -\sin(i\theta_f) \\ \sin(i\theta_f) & \cos(i\theta_f) \end{pmatrix}. \quad (1)$$

Applying this rotation to the query $\boldsymbol{q}_i$ and key $\boldsymbol{k}_j$ at positions $i$ and $j$, their inner product (attention logit) becomes:

$$(\boldsymbol{R}_i \boldsymbol{q}_i)^\mathsf{T}(\boldsymbol{R}_j \boldsymbol{k}_j) = \boldsymbol{q}_i^\mathsf{T}(\boldsymbol{R}_i^\mathsf{T} \boldsymbol{R}_j)\boldsymbol{k}_j = \boldsymbol{q}_i^\mathsf{T} \boldsymbol{R}_{j-i} \boldsymbol{k}_j. \quad (2)$$

The attention logit is then defined as $s_{i,j} = \frac{1}{\sqrt{d}} \boldsymbol{q}_i^\mathsf{T} \boldsymbol{R}_{j-i} \boldsymbol{k}_j$. The key property $\boldsymbol{R}_i^\mathsf{T} \boldsymbol{R}_j = \boldsymbol{R}_{j-i}$ ensures that the attention score depends solely on the relative distance $(j - i)$ rather than absolute positions (Su et al., 2021). Notably, if $\theta_f = 0$ for all $f$, $\boldsymbol{R}_i$ becomes the identity matrix, equivalent to NoPE (Haviv et al., 2022); whereas setting $\theta_f = 0$ for a subset of frequencies results in Partial RoPE (Barbero et al., 2025).

### 2.2. Attention Scaling

A fundamental limitation of standard softmax attention under increasing context length is the *attention dilution* phenomenon: as the context length $L$ increases, the probability mass assigned to any single token inevitably decreases, leading to an overly "flat" attention distribution.

Existing approaches address this issue by uniformly scaling the raw attention scores with a length-dependent factor $\lambda(L)$ (e.g., $\lambda(L) = \log L$) before applying softmax, where $L$ denotes the context length, to keep the attention distribution's variance or entropy stable as $L$ grows (Han et al., 2023; Nakanishi, 2025; Anson et al., 2025; Bai et al., 2023; Li et al., 2025b). However, as we show in Section 3.2 , such uniform scaling is fundamentally insufficient to accommodate the heterogeneous requirements of attention heads.

## 3. AdaRoPE

In this section, we motivate and introduce AdaRoPE. We first establish the necessity of head-wise rotation frequencies, followed by an analysis showing why head-wise attention scaling is required for length extrapolation. We then present the design of AdaRoPE, which integrates both components into a unified and practical approach.

### 3.1. Necessity of Head-wise Rotation Frequencies

In this subsection, we introduce a simplified retrieval problem and show, both empirically and theoretically, that the

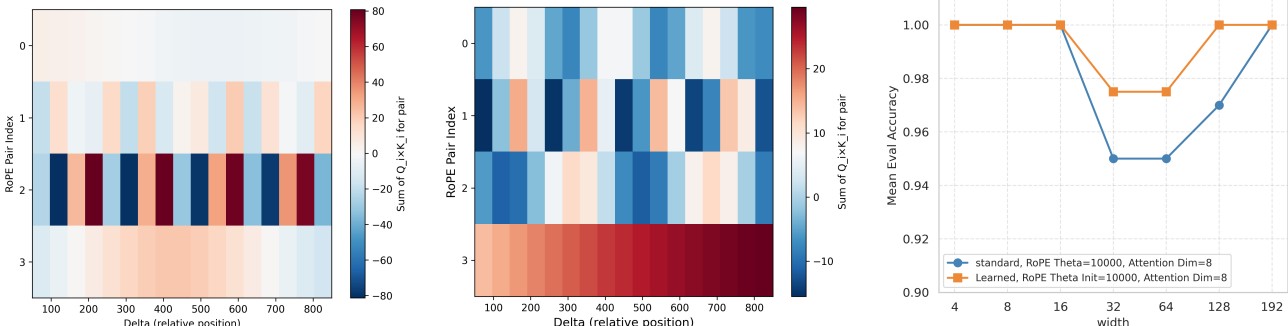

*Figure 1.* **Window-dependent sparse utilization of RoPE frequencies.** We analysis per-dimension-pair contributions to the attention logit between a query $q$ and a key $k$ at different relative distances $\Delta$, where color indicates $\sum_i q_i k_i$ for each RoPE dimension pair. Darker red or blue denotes larger-magnitude logit contributions. It clearly exhibits distinct periodic patterns across different relative distances. We plot the case of Dim = 8, $c = 512$ and $W = 32$. **Left:** Standard RoPE mainly uses the second RoPE Pair, leaving other dimensions almost unused. **Middle:** RoPE with learned frequencies (AdaFreq) effectively utilizes all attention dimensions by adjusting frequency, as evidenced by a larger number of cells with large absolute values. **Right:** Mean evaluation accuracy versus window width $W$. The fixed RoPE baseline exhibits significantly lower accuracy compared to the learned approach, due to its suboptimal utilization of frequencies.

effective frequencies utilized by RoPE depend critically on the target window size. These results provide strong motivation for head-wise frequency specialization. The retrieval task is defined as follows.

**Needle Retrieval from a Window (NRW).** The objective of NRW is to determine whether a designated NEEDLE token lies within a target window. The window is parameterized by its *center $c$* and *width $W$*, defining the interval $[c - W/2, \, c + W/2]$. If the NEEDLE token falls inside this interval, the model outputs YES; otherwise, it outputs NO.

**Empirical Observations.** We train a single-layer Transformer with a single attention head with a vocab embedding layer and a language modeling head on fixed-length sequences over synthetic data. The input format of each synthetic sample is: $[\ldots, \texttt{BG}, \ldots, \texttt{NEEDLE}, \ldots, \texttt{BG}, \ldots, \texttt{QUERY}] \rightarrow \texttt{YES/NO}$. where BG is the background token.

Figure 1 (Left) reveals a highly *sparse* utilization of RoPE dimensions, characterized by the activation of only a confined frequency range rather than the full spectrum. This active band is not fixed but is instead governed by the window width $W$; specifically, an increase in $W$ triggers a systematic downward shift in the selected frequency range, as further evidenced in Figure 5.

**Theoretical Analysis.** In fact, this empirical behavior observed above is consistent with classical Fourier analysis, where broader spatial windows correspond to lower-frequency components (Oppenheim et al., 1997). In Section D, we formalize this intuition using tools from Fourier analysis and show that the frequency mass—largely determined by the rotation frequencies employed by the

model—must be concentrated within a range that explicitly depends on the window length $W$ (see Theorem 1).

It is well known that different attention heads in modern Large Language Models (LLMs) specialize in diverse functions (Clark et al., 2019b; Olsson et al., 2022; Wu et al., 2024). Some heads operate locally, focusing solely on the preceding token (effectively $W \approx 1$), while others attend globally to the entire sequence ($W \approx L$). We also observed heads in LLM with sparse dimension usage, as shown in Figure 2b. Hence, both the empirical evidence and theoretical analysis provide strong motivation of allowing frequency configurations to adapt on a per-head basis.

### 3.2. Necessity of Head-wise Scaling Factor

For length generalization and extrapolation, prior methods apply a uniform scaling factor to stabilize the variance or entropy of the attention logit distribution. In this subsection, we argue that heads with different functionalities impose fundamentally different requirements on their attention distributions during context length extension. To formalize this intuition, we introduce the notion of *effective sequence length*, denoted by $\mathcal{E}$:

$$\mathcal{E}_\beta(\boldsymbol{\alpha}) = \left( \sum_{i=1}^{L} \alpha_i^\beta \right)^{\frac{1}{1-\beta}}, \quad \beta > 1,$$

where $\boldsymbol{\alpha}$ represents the attention weights after softmax, and $\mathcal{E}_\beta(\boldsymbol{\alpha})$ measures the effective number of tokens the model attends to. This metric stems from the generalized effective sample size designed for importance sampling (Huggins & Roy, 2019; Martino et al., 2017).

Intuitively, the metric $\mathcal{E}_\beta(\boldsymbol{\alpha})$ quantifies the effective number of tokens that receive non-negligible attention. We refer to

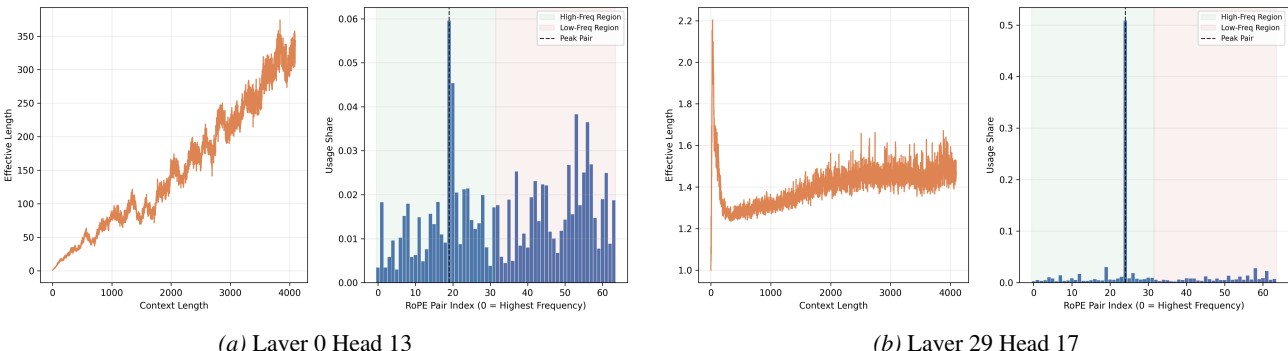

*(a)* Layer 0 Head 13          *(b)* Layer 29 Head 17

*Figure 2.* **Head-wise RoPE utilization and effective length in LLaMA-3-8B.** Each subfigure shows one representative head with *effective length* ($\mathcal{E}_2$; left panel) and RoPE dimension-pair utilization (right panel), measured on 1,024 FineWeb-Edu samples. **a:** $\mathcal{E}_2$ grows with context length and the frequency usage is relatively uniform. **b:** $\mathcal{E}_2$ stays nearly constant across context lengths and usage is highly sparse/spiky. Together, these results show strong head-level heterogeneity in both effective attention span and spectral utilization.

this quantity as the *effective length* . For example, in the case of uniform attention ($\alpha_i = 1/L$), the effective length is maximized, yielding $\mathcal{E}_2(\boldsymbol{\alpha}) = L$, which corresponds to a fully distributed focus over the sequence. In contrast, when attention collapses to a single token (i.e., a one-hot distribution), $\mathcal{E}_\beta(\boldsymbol{\alpha}) = 1$ for any $\beta$, reflecting a highly concentrated focus.

Empirically, the effective length varies dramatically across attention heads, consistent with the sparse-head behavior in Figure 2. For instance, in Llama3-8B, head 13 in layer 0 (Figure 2a) maintains an effective length on the order of a few hundred tokens and grows approximately linearly with the input context length, whereas the head 17 in layer 29 (Figure 2b) remains highly localized, with effective length staying near 1–2 across the entire range. To formally characterize how different attention heads impose distinct requirements on the attention weight distribution $\boldsymbol{\alpha}$ and the effective sequence length $\mathcal{E}(\boldsymbol{\alpha})$, we consider two representative attention head types with contrasting roles.

**Retrieval Heads.** These heads specialize in precise key–value associations, such as identifying a specific "needle" token within an increasingly long "haystack". In particular, during context length extension, it is crucial that the logit corresponding to the target token remains dominant even as additional background tokens are appended. In Theorem 2 (Section E.2), we show that maintaining such sharp focus on the needle token as the sequence length $L$ grows requires a *low and constant* $\mathcal{E}$. This, in turn, necessitates an aggressive scaling factor to counteract the natural dilution of attention weights induced by longer contexts.

**Global Aggregation Heads.** In contrast, these heads are responsible for extracting global statistics or aggregating information across the entire context, which requires attending broadly to all tokens. For such heads, Theorem 3 (Section E.3) shows that a *high and increasing* $\mathcal{E}$ is essential as $L$ grows, allowing the attention distribution to flatten and cover the expanding sequence.

### 3.3. Design of AdaRoPE

We introduce **AdaRoPE**, a drop-in extension of RoPE that equips transformers with (i) *dimension-wise, head-specific* rotary frequencies (**AdaFreq**) and (ii) *head-specific, length-aware* attention temperatures (**AdaScale**). Here we use $h \in \{1, \dots, H\}$ to represent different attention heads.

**AdaFreq: Head-Specific Rotary Frequencies.** AdaFreq replaces the fixed geometric schedule with *learnable*, per-head, per-block angular speeds. For numerical stability, we learn the *log-frequency* parameter $\xi_f^{(h)} \in \mathbb{R}$ for head $h$ and map it to a positive frequency via

$$\theta_f^{(h)} = \exp\left(\xi_f^{(h)}\right).$$

So the head-specific rotation matrix becomes:

$$\boldsymbol{R}_i^{(h)} = \mathrm{diag}\left(\boldsymbol{G}_i\left(\theta_0^{(h)}\right), \dots, \boldsymbol{G}_i\left(\theta_{d/2-1}^{(h)}\right)\right),$$

where $\boldsymbol{G}_i$ is defined in Eq. (1).

**AdaScale: Per-Head, Length-Aware Attention Temperatures.** To counteract attention dilution as the context length $L$ grows, AdaScale multiplies each head's logits by a *head-specific inverse temperature* $\lambda(L)$: the logit (for query $\boldsymbol{q}_i^{(h)}$ and key $\boldsymbol{k}_j^{(h)}$ in head $h$) is computed as

$$s_{i,j}^{(h)} = \frac{\lambda^{(h)}(L)}{\sqrt{d}}(\boldsymbol{q}_i^{(h)})^\mathsf{T}\boldsymbol{R}_{j-i}^{(h)}\boldsymbol{k}_j^{(h)},$$

We use a smooth, positive schedule that is monotone in $L$ and grows when $\gamma^{(h)} > 0$ :

$$\lambda^{(h)}(L) := \underbrace{\frac{1}{\tau^{(h)}}}_{\text{global scale}} \cdot \underbrace{\left[\ln\left(1 + \frac{\max\left\{L, L_{\mathrm{ref}}\right\}}{L_{\mathrm{ref}}}\right)\right]^{\gamma^{(h)}}}_{\text{length growth based on recency strength}}$$

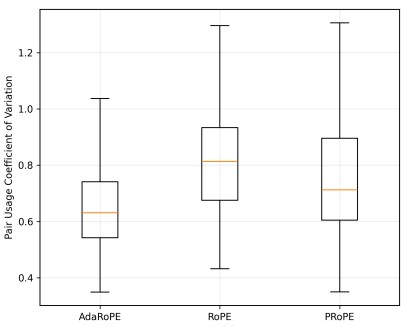 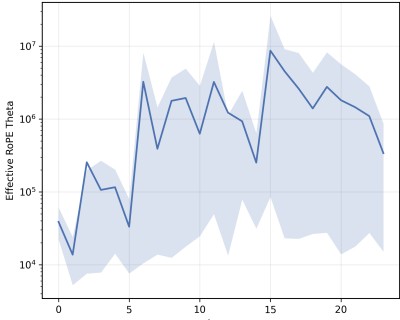 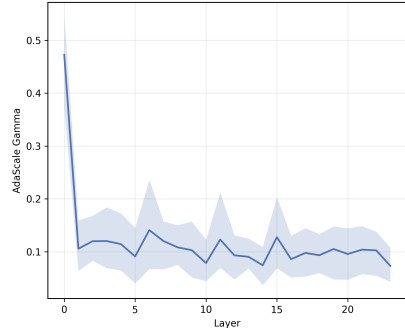

*Figure 3.* **Left:** AdaRoPE yields a lower coefficient of variation (CoV) across dimensions compared to RoPE and PRoPE, indicating a more balanced utilization of frequencies. The box plot represents the min, max, 25th and 75th percentiles, and the mean across all heads. **Middle:** The effective RoPE base $\Theta$ exhibits an increasing trend with network depth, suggesting that deeper layers prioritize learning lower frequencies. **Right:** The AdaScale parameter $\gamma$ peaks in the initial layer and rapidly stabilizes around 0.1 in subsequent layers. In the Middle and Right plots, we plot the mean value across heads in the same layer, and shaded regions denote the 0.1–0.9 quantile range.

where $L_{\text{ref}}$ is a hyperparameter. The choice of temperature is motivated by Theorems 4 and 5 (Section E.4). Larger $\gamma^{(h)}$ yields sharper growth with $L$ (stronger recency), while smaller $\tau^{(h)}$ increases the overall inverse temperature. This design allows heads with different recency profiles to *adaptively* learn distinct $\gamma^{(h)}$ and $\tau^{(h)}$. In practice, we apply $\lambda^{(h)}(L)$ as a per-head scalar to the query vector only ($\boldsymbol{q}_i^{(h)} \leftarrow \lambda^{(h)}(L)\boldsymbol{q}_i^{(h)}$), leaving the key/value cache and the attention kernel unchanged.

### 3.4. Implementation and Overhead

**Implementation.** AdaRoPE is implemented as a drop-in replacement for RoPE: it preserves tensor shapes and interfaces, and remains fully compatible with FlashAttention-style fused attention kernels and standard KV caching (Dao et al., 2022). For grouped-query attention (GQA) (Ainslie et al., 2023), we learn AdaFreq at the *group* level (shared across heads within each KV group), while AdaScale is learned *per head*. For numerical stability, we store AdaRoPE's parameter in fp32.

**AdaRoPE in Context Extension.** Any pre-trained model utilizing RoPE can be seamlessly adapted to AdaRoPE for continued pretraining. This flexibility allows for diverse optimization strategies, ranging from tuning AdaRoPE parameters in isolation to joint training with the backbone. For context extension, we adopt a parameter-efficient strategy: we *freeze* the backbone and fine-tune only the frequency $\{\xi_f^{(h)}\}$ and scaling parameters $\{\gamma^{(h)}, \tau^{(h)}\}$. To accelerate convergence, we set the frequencies and base temperatures to start from a YaRN-style schedule (Peng et al., 2023) targeting the new context length. Crucially, we initialize the length-dependent growth term as $\gamma^{(h)} = 0$, allowing the model to adaptively learn the appropriate temperature growth dynamics from a flat baseline during training. In addition, we set $L_{ref}$ to the model's original context length

to preserve the short context ability.

**Overhead.** The runtime overhead of AdaRoPE is nearly imperceptible in practice. AdaFreq replaces a fixed frequency table with learned frequencies but follows the same compute pattern (elementwise sinusoidal rotations), and AdaScale introduces only a per-head scalar multiplication on the query, leaving the key/value cache layout and the attention kernel unchanged. Parameter overhead is also negligible: AdaRoPE adds only on the order of $10^{-5}$ of the total parameters (typically ~10–50K parameters), and thus does not meaningfully affect model capacity or memory footprint.

## 4. Pretraining Experiment

To validate the effectiveness of AdaRoPE during pretraining, we conduct experiments across models of different scales and architectures, and find that AdaRoPE consistently outperforms strong baselines; we then perform ablations over its individual components and finally analyze the frequency and scaling learned by AdaRoPE.

### 4.1. Experiment Setting

**Data and Model.** We pretrain four model variants on FineWeb-Edu-100B (Penedo et al., 2024): LLaMA-430M (Multi-Head Attention; MHA) (Grattafiori et al., 2024), LLaMA-1.3B (Grouped-Query Attention; GQA) (Grattafiori et al., 2024), OLMoE-2.7B (0.4B activated) (Muennighoff et al., 2024), and Qwen3 Gated Attention-430M (Qiu et al., 2025); full hyperparameter details are provided in Section G.1.

**Evaluation.** We evaluate on seven standard benchmarks (ARC-C (Clark et al., 2018), ARC-E (Clark et al., 2018), BoolQ (Clark et al., 2019a), HellaSwag (Zellers et al., 2019),

*Table 1.* **NLU and LM evaluation.** We report accuracy on seven NLU benchmarks (ARC-C/E, BoolQ, HellaSwag, Lambada, PIQA, WinoGrande); **Avg** denotes the mean accuracy across the seven tasks. For ARC, HellaSwag, and PIQA we report length-normalized accuracy, while the remaining NLU tasks use accuracy. We additionally report Loss (averaged pretraining loss over the last 200 steps; lower is better) and Wiki (WikiText-103 perplexity; lower is better). Best per benchmark within each backbone is boldfaced. AdaRoPE achieves the best average NLU accuracy across all evaluated backbones and improves pretraining loss on average.

| Method | ARC-C | ARC-E | BoolQ | HellaS | Lambada | PIQA | WinoG | Avg | Loss | Wikitext |
|---|---|---|---|---|---|---|---|---|---|---|
| **LLaMA 430M** | | | | | | | | | | |
| AdaRoPE | **33.28** | **61.78** | 59.45 | **45.02** | 37.16 | **68.82** | 53.12 | **51.23** | **2.663** | **22.31** |
| RoPE | 31.66 | 60.82 | 57.16 | 44.94 | 37.94 | 67.95 | 54.06 | 50.65 | 2.692 | 22.92 |
| PRoPE | 32.76 | 60.31 | 60.06 | 44.75 | **38.35** | 67.79 | 54.14 | 51.17 | 2.686 | 22.70 |
| ALiBi | 30.72 | 59.93 | 59.66 | 44.22 | 37.45 | 68.39 | **55.09** | 50.78 | 2.671 | 22.46 |
| NoPE | 32.08 | 59.05 | **60.58** | 41.25 | 36.79 | 66.38 | 52.33 | 49.78 | 2.740 | 23.73 |
| **OLMoE-2.7B (0.4B active)** | | | | | | | | | | |
| AdaRoPE | **40.27** | **70.24** | 60.98 | **53.47** | 43.72 | 71.49 | **56.43** | **56.66** | **2.717** | 17.54 |
| RoPE | 38.99 | 67.59 | 60.55 | 52.45 | **43.95** | 71.33 | 55.49 | 55.76 | 2.725 | 18.00 |
| PRoPE | 35.07 | 68.31 | **62.26** | 52.77 | 43.86 | **71.71** | 54.46 | 55.49 | 2.718 | **17.49** |
| ALiBi | 37.12 | 66.71 | 61.89 | 50.17 | 40.98 | 70.76 | 55.29 | 54.70 | 2.735 | 18.31 |
| NoPE | 33.19 | 64.52 | 60.06 | 48.00 | 38.87 | 69.48 | 53.91 | 52.58 | 2.798 | 19.14 |
| **LLaMA 1.3B** | | | | | | | | | | |
| AdaRoPE | **42.41** | **70.92** | 60.21 | **54.80** | 45.33 | 72.58 | **59.35** | **57.94** | **2.442** | **16.95** |
| RoPE | 38.57 | 69.65 | 59.97 | 54.13 | 44.71 | **72.85** | 57.06 | 56.71 | 2.452 | 17.45 |
| PRoPE | 39.16 | 68.64 | 58.26 | 54.69 | **45.53** | 72.25 | 56.99 | 56.50 | 2.444 | **16.95** |
| ALiBi | 40.02 | 69.19 | 61.01 | 53.79 | 45.41 | 71.49 | 56.83 | 56.82 | 2.451 | 17.06 |
| NoPE | 36.77 | 67.59 | **61.56** | 51.52 | 42.71 | 71.82 | 55.33 | 55.33 | 2.491 | 17.80 |
| **Qwen3 Gated Attention 430M** | | | | | | | | | | |
| AdaRoPE | 32.25 | **61.49** | **60.83** | 45.10 | 37.78 | **69.42** | 52.64 | **51.36** | **2.656** | **20.08** |
| RoPE | 31.91 | 59.64 | 58.87 | 44.64 | 37.36 | 67.30 | 53.20 | 50.42 | 2.671 | 26.20 |
| PRoPE | 29.52 | 59.68 | 59.85 | **46.00** | 37.51 | 68.77 | 53.12 | 50.64 | 2.659 | 21.15 |
| ALiBi | **32.34** | 60.77 | 59.20 | 44.70 | **38.87** | 68.01 | **54.78** | 51.24 | 2.679 | 32.30 |
| NoPE | 29.18 | 58.50 | 59.08 | 41.97 | 34.58 | 66.59 | 53.75 | 49.09 | 2.724 | 34.57 |
| **LLaMA 430M Ablations** | | | | | | | | | | |
| AdaRoPE | **33.28** | 61.78 | 59.45 | **45.02** | 37.16 | **68.82** | 53.12 | **51.23** | **2.663** | **22.31** |
| Share Freq. | 30.20 | 61.49 | **60.73** | 44.64 | 36.97 | 67.25 | **55.01** | 50.90 | 2.685 | 22.45 |
| No AdaScale | 33.02 | 61.53 | 56.73 | 44.97 | **38.83** | 68.50 | 51.78 | 50.77 | 2.683 | 22.40 |
| Fixed Log | 31.66 | **62.54** | 56.76 | 44.71 | 37.45 | 68.01 | 52.96 | 50.58 | 2.687 | 23.14 |
| Learned Base | 31.48 | 61.45 | 58.17 | 44.89 | 37.28 | 68.28 | 53.83 | 50.77 | 2.685 | 22.76 |

Lambada (Paperno et al., 2016), PIQA (Bisk et al., 2019), and WinoGrande (Sakaguchi et al., 2019)) and report accuracy. We additionally report the mean accuracy across these tasks to reduce the evaluation noise (Madaan et al., 2024). Following common evaluation practice, we report length-normalized accuracy for ARC, HellaSwag, and PIQA, and accuracy for the remaining NLU tasks. We use 5-shot evaluation for all NLU tasks except Lambada using 0-shot. In addition, we include two language-modeling metrics: the pretraining loss and WikiText perplexity (Merity et al., 2018).

**Baselines.** We compare AdaRoPE against widely used positional embedding baselines while keeping others fixed, changing *only* the positional embedding mechanism. Specifically, we include: RoPE (Su et al., 2021), Partial RoPE (PRoPE) (Barbero et al., 2025), ALiBi (Press et al., 2021) and NoPE (Chi et al., 2023).

## 4.2. Results

**Main Results.** Table 1 shows that AdaRoPE consistently outperforms RoPE and other positional embedding baselines across models and scales. Compared to RoPE, AdaRoPE improves mean accuracy over the seven NLU tasks by 0.91 points on average and reduces pretraining loss by 0.016 on average.

**Ablation on Head-wise Frequency Selection.** We investigate the necessity of head-wise frequency learning by forcing all heads to share a single learnable frequency schedule (*Share Freq.*). This restriction consistently leads to performance degradation, suggesting that distinct attention heads require unique frequency bands to function effectively.

**Ablation on Head-wise Scaling.** We evaluate the impact of AdaScale by testing two variants: removing the scaling factor entirely (*No AdaScale*) and replacing it with a fixed log-style scaling rule (*Fixed Log*). Both configurations result in inferior performance, with log-style scaling failing to

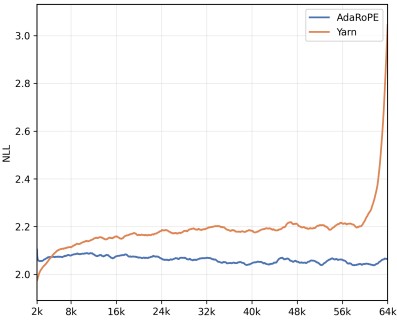 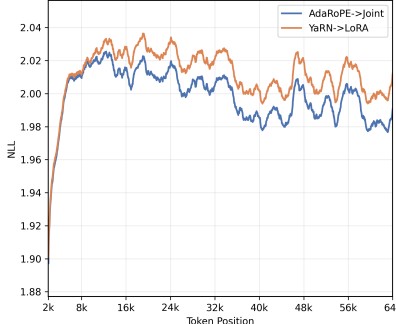 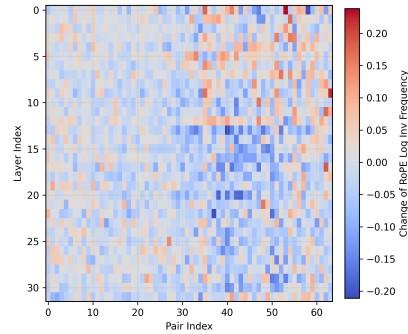

*Figure 4.* Analysis of extrapolation performance and learned frequency patterns. **Left:** NLL (negative log-likelihood) versus token position comparing AdaRoPE and YaRN during the extrapolation stage. **Middle:** NLL comparison during the long-context continued pretraining stage (AdaRoPE vs. YaRN+LoRA). **Right:** Visualization of log-frequency modifications (difference between the YaRN and learned log inverse frequency) across layers and dimension pairs. We observe that the frequencies of early layers tend to shift towards lower frequencies, indicated by postive change of inverse frequency, while deeper layers shift towards higher frequencies. Notably, modifications across all layers largely occur in the low-frequency components.

recover the accuracy loss caused by removing the learned scaling mechanism.

**Ablation on Frequency Parameterization.** We compare our proposed parameterization against an alternative approach that directly learns the base $b$ (*Learned Base*), where frequencies are subsequently recovered via the log-uniform formula $\theta_f = b^{-2f/d}$. This alternative yields worse performance, indicating that our method of learning head-wise frequency selection is more effective than deriving frequencies from an optimized base parameter.

### 4.3. Analysis

**More Sufficient Frequency Utilization.** To quantify how evenly different RoPE frequencies are utilized, we employ the dimension-wise Coefficient of Variation (CoV) of the pair-usage statistics. Formally, the CoV is defined as $C_v = \frac{\sigma}{\mu}$, where $\sigma$ and $\mu$ represent the standard deviation and the mean of the usage statistics across dimensions, respectively. A lower CoV indicates a more balanced distribution with less dispersion. As shown in Figure 3 (left), AdaRoPE attains a noticeably lower CoV than both RoPE and PRoPE, demonstrating that our method promotes a more sufficient utilization of frequency bands rather than concentrating on a small subset of dimensions.

**Learned Frequencies and Scaling Vary Systematically with Layer Depth.** We next examine the learned depthwise parameters that govern effective length specialization. Figure 3 (Middle) shows that the effective RoPE base $\Theta = \exp\left(-\frac{4}{d-2}\sum_{i=0}^{d/2-1}\xi_i^{(h)}\right)$ increases with layer depth, implying that deeper layers operate with slower effective frequencies (i.e., larger effective wavelengths), consistent with a shift toward longer-range positional behavior in later layers. In parallel, the AdaScale parameter $\gamma$ exhibits a

sharp layer dependence (Figure 3, Right): it is largest in the first layer and rapidly decays to around $0.1$ for most subsequent layers, remaining relatively stable thereafter.

## 5. Experiment on Context Extension

In this section, we evaluate the effectiveness of AdaRoPE in two distinct settings: extrapolation with limited data and long-context continued pretraining.

First, we assess the extrapolation capability of AdaRoPE in a data-constrained regime. By fine-tuning only the frequency and scaling parameters on as few as 100 samples (while keeping the backbone frozen), we observe that AdaRoPE achieves significantly better performance on long-context tasks compared to YaRN (Peng et al., 2023).

Second, we explore the potential of AdaRoPE to enhance long-context continued pretraining (Chen et al., 2023b). We investigate different training strategies and find that a two-stage optimization approach—initially optimizing AdaRoPE parameters while freezing the model, followed by jointly training LoRA and AdaRoPE parameters—yields the best improvements.

### 5.1. Overall Experiment Setting

We conduct our experiments using SmolLM-2-1.7B (Allal et al., 2025) and Llama-3-8B (Grattafiori et al., 2024) as the backbone models. Both of these models have a native context length of 8k, and we extend it to 64k in our experiment. We use the PG19 dataset (Rae et al., 2019) for long context finetuning. For general language capabilities, we evaluate the models on the Natural Language benchmarks described in Section 4.1, following the identical setup and reporting the average scores. To assess long-context performance, we utilize the RULER (Hsieh et al., 2024) benchmark and report the mean results across different sequence lengths.

## 5.2. Extrapolation

For context extrapolation, we *freeze* the backbone and fine-tune only AdaRoPE parameters on the first 100 training samples of PG19 (Rae et al., 2019).

**Main Results.** Table 2 presents the performance of LLaMA 8B and SmolLM 1.7B on NLU and RULER benchmarks. A common challenge in context extrapolation is the degradation of short-context capabilities, as evidenced by the performance drops in NLU and Ruler-8k metrics. AdaRoPE partially mitigates the short-context degradation caused by global scaling (e.g., YaRN), especially on LLaMA-8B; on SmolLM, short-context RULER scores still drop, but AdaRoPE remains consistently better than YaRN at all evaluated lengths.

In terms of long-context capabilities (Ruler-16k to Ruler-64k), AdaRoPE also achieves better performance. Notably, on LLaMA 8B, while the YaRN fails almost completely at 64k, AdaRoPE maintains a high retrieval accuracy of 51.87 at 64k length. This implies that context extrapolation can be substantially improved solely by optimizing the RoPE rotation frequencies and attention scaling factors, without modifying model weights.

**Ablation Studies.** We further analyze the contribution of each component using SmolLM 1.7B (bottom section of Table 2). We consider three variants: (1) *w/o AdaFreq*: Removing the adaptive frequency learning (reverting to fixed YaRN angles); (2) *w/o AdaScale*: Removing the adaptive scaling (reverting to a fixed YaRN-style global scaling schedule); (3) *Share Freq.*: Constraining the learned frequencies to be shared across all attention heads.

The results indicate that the full AdaRoPE configuration is essential for robust long-context performance. Removing either adaptive frequency or scaling leads to a significant decline in Ruler scores. Interestingly, we observe a trade-off: ablated versions (e.g., *w/o AdaScale*) yield slightly better short-context scores (NLU). We hypothesize that AdaRoPE utilizes its higher expressivity to better fit the long-context training data, but since we do not mix short texts during fine-tuning, this adaptation incurs a marginal cost on short-context performance.

## 5.3. Long Context Continued Pretraining

We further scale up our investigation to continued pretraining. We employ LoRA (Hu et al., 2021; Chen et al., 2023b) for efficient fine-tuning on a 600M-token subsample of PG19.

**Training Strategies.** To identify the optimal approach for combining parameter-efficient fine-tuning with adaptive rotary embeddings, we compare four strategies: (1)

*Table 2.* Extrapolation with AdaRoPE (NLU and RULER). '–' indicates not applicable due to the backbone's native context limit.

| Method | NLU | R-8k | R-16k | R-32k | R-64k |
|---|---|---|---|---|---|
| **LLaMA 8B** | | | | | |
| AdaRoPE | 63.05 | 82.07 | **77.86** | **70.55** | **51.87** |
| Backbone | **64.79** | **82.37** | – | – | – |
| Backbone+YaRN | 62.53 | 75.76 | 71.10 | 66.84 | 12.80 |
| **SmolLM 1.7B** | | | | | |
| AdaRoPE | 62.73 | 30.07 | **24.16** | **20.21** | **15.83** |
| Backbone | **64.94** | **48.50** | – | – | – |
| Backbone+YaRN | 62.56 | 24.73 | 18.41 | 14.64 | 11.49 |
| **SmolLM 1.7B Ablation** | | | | | |
| AdaRoPE | 62.73 | **30.07** | **24.16** | **20.21** | **15.83** |
| w/o AdaFreq | 63.15 | 27.87 | 20.38 | 15.32 | 13.13 |
| w/o AdaScale | **64.01** | 18.27 | 16.48 | 18.26 | 9.97 |
| Share Freq. | 63.96 | 25.75 | 16.79 | 16.87 | 10.70 |

*YaRN→LoRA:* Directly training LoRA parameters using fixed YaRN embeddings. (2) *AdaRoPE LoRA Joint:* Joint training both AdaRoPE and LoRA parameters. (3) *AdaRoPE → LoRA:* A two-stage strategy where we first warm-up AdaRoPE (using the 100 samples from the extrapolation stage), freeze it, and then train LoRA. (4) *AdaRoPE → Joint:* The proposed strategy, which initializes with the warmed-up AdaRoPE, then jointly trains both AdaRoPE and LoRA. For fair comparison, we control the total number of training tokens to be identical across all single-stage and two-stage methods.

**Main Results.** Table 3 summarizes the performance across different strategies. The *AdaRoPE → Joint* strategy consistently achieves the best performance on both LLaMA 8B and SmolLM 1.7B, particularly in long-context scenarios. We further visualize the perplexity comparison and the evolution of AdaFreq during the training process in Figure 4. We provide further analysis in Section B.3.

We interpret this result through the lens of optimization landscapes. The fixed positional representation (i.e., YaRN) used in *YaRN→LoRA* may not be optimal for extremely long contexts. Directly updating model parameters (LoRA) on top of this suboptimal basis can cause the model to converge into a suboptimal local minimum. By treating the first stage as a "positional alignment" phase, AdaRoPE learns a more suitable frequency spectrum for long contexts. This warmed-up initialization provides a better starting point for the subsequent joint training, allowing the model to refine both its weights and positional view effectively.

**Ablation Studies.** We also conduct ablation studies under the optimal AdaRoPE → Joint setting on SmolLM 1.7B, as shown in Table 4. Consistent with our extrapolation findings, the full AdaRoPE configuration outperforms variants without adaptive frequency *(w/o AdaFreq)*, without adaptive

Table 3. Comparison of context extension strategies. Joint: joint training of AdaRoPE and LoRA; →: two-stage warm-start. '–' indicates not applicable due to the backbone's native context limit.

| Method | NLU | R-8k | R-16k | R-32k | R-64k |
|---|---|---|---|---|---|
| **LLaMA 8B** | | | | | |
| AdaRoPE → Joint | 63.55 | **84.03** | **82.91** | **76.03** | **69.09** |
| AdaRoPE LoRA Joint | 63.94 | 81.53 | 82.32 | 75.49 | 61.62 |
| AdaRoPE → LoRA | 64.07 | 83.58 | 80.40 | 72.91 | 63.27 |
| YaRN→LoRA (YaRN) | 63.99 | 83.34 | 80.23 | 73.84 | 61.60 |
| Backbone | **64.79** | 82.37 | – | – | – |
| **SmolLM 1.7B** | | | | | |
| AdaRoPE → Joint | 63.16 | **51.92** | **35.25** | 32.18 | **31.07** |
| AdaRoPE LoRA Joint | 63.03 | 42.15 | 31.07 | **32.73** | 28.93 |
| AdaRoPE → LoRA | 63.25 | 39.98 | 30.18 | 28.47 | 28.69 |
| YaRN→LoRA (YaRN) | 62.90 | 33.76 | 28.14 | 30.06 | 25.06 |
| Backbone | **64.94** | 48.50 | – | – | – |

Table 4. Ablations and baselines under the AdaRoPE → Joint protocol (SmolLM 1.7B).

| Method | NLU | R-8k | R-16k | R-32k | R-64k |
|---|---|---|---|---|---|
| AdaRoPE | 63.16 | **51.92** | **35.25** | **32.18** | **31.07** |
| w/o AdaFreq | 63.31 | 41.11 | 30.46 | 26.61 | 28.07 |
| w/o AdaScale | **64.43** | 41.99 | 32.38 | 30.57 | 28.35 |
| Share Freq. | 64.37 | 46.85 | 31.63 | 28.12 | 25.49 |
| YaRN→LoRA (YaRN) | 62.90 | 33.76 | 28.14 | 30.06 | 25.06 |

scaling (*w/o AdaScale*), or with shared frequencies (*Share Freq.*). This confirms that even when model parameters (LoRA) are trainable, the flexibility provided by independent, head-wise frequency and scaling adaptation remains crucial for maximizing long-context retrieval capabilities.

## 6. Conclusion

Rotary Position Embedding (RoPE) is widely adopted in modern Transformers, yet their standard formulation uses uniform rotation frequencies and attention scaling across all heads. We show such a design choice is fundamentally misaligned with head heterogeneity, through simplified yet representative theoretical settings. To address this limitation, we propose AdaRoPE, a drop-in extension that learns head-wise, dimension-wise frequencies (AdaFreq) and length-aware attention scaling (AdaScale). Our experiments show that AdaRoPE consistently outperforms strong baselines across a range of model scales, yielding significant gains in both general modeling quality and long-context extrapolation. By optimizing positional embedding at the granularity of individual attention heads, AdaRoPE provides a simple yet powerful and computationally efficient mechanism for robust long-context modeling, while remaining fully compatible with standard hardware and training pipelines.

## Acknowledgements

This work was supported by the Xiongan AI Institute and Tencent. We extend our gratitude to Kaifeng Lyu, Guancheng Du, Xinnuo Chen, Kaiyue Wen, Zhixuan Pan, and Binghui Li for insightful discussions and meticulous proofreading. We also thank the anonymous reviewers for their constructive comments, which helped improve this paper.

## Impact Statement

This work advances the efficiency and capability of Transformer-based language models, specifically regarding long-context generalization. By enabling more effective context extension with reduced computational overhead, AdaRoPE facilitates improvements in high-value applications such as document understanding, long-form summarization, and codebase analysis. Furthermore, because this method achieves strong extrapolation with limited additional training, it may lower the resource barriers for adapting models to long-context tasks, thereby democratizing access to capable long-context processing.

However, enhanced capabilities in processing extensive contexts could amplify risks regarding the large-scale generation of misinformation or the inadvertent leakage of private information found in lengthy sensitive documents. While our method modifies architectural parameters rather than introducing new datasets, the resulting increase in model capability necessitates responsible deployment. We encourage practitioners to pair long-context extensions with robust data governance, privacy-aware training practices, and rigorous evaluation protocols to mitigate potential misuse and privacy risks in long-context regimes.

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

# A. Additional Related Work

**Positional embedding.**   Early Transformer language models commonly use *absolute* positional embedding, typically learned vectors added to token embeddings, as in GPT-style decoders  (Radford & Narasimhan, 2018).  Later work popularized *relative* schemes that modify attention scores as a function of token offsets. T5 introduces a *bucketed relative position bias*, where relative distances are discretized into buckets and a learned scalar bias is added to attention logits  (Raffel et al., 2019). ALiBi instead adds a *linear distance penalty* to attention logits, with different slopes per head, enabling "train short, test long" extrapolation without adding positional vectors  (Press et al., 2021). PaTH replaces fixed position transforms with *data-dependent* (input-conditioned) Householder-like transformations accumulated across positions, improving expressivity while retaining efficient implementations  (Yang et al., 2025b).

**RoPE and RoPE variants.**   RoPE  (Su et al., 2021) encodes positions by rotating queries and keys with position-dependent 2D block rotations, so the attention interaction depends on *relative offsets* through composed rotations. Recent analyses of RoPE's mechanics study how different frequencies are exploited and how RoPE can induce uneven usage patterns across dimensions/heads (e.g., Partial RoPE)  (Barbero et al., 2025). Several works modify how RoPE is applied across dimensions or modalities. RoPE-Mixed adapts RoPE to higher-dimensional settings (e.g., vision), combining axes to better handle resolution changes and serve as a relative position mechanism in 2D  (Heo et al., 2024). LieRE generalizes RoPE using Lie group constructions to support $n$-dimensional inputs, replacing RoPE's canonical 2D plane rotations with rotations derived from learned Lie algebra generators  (Ostmeier et al., 2024). GRAPE provides a unifying group-action framework that subsumes multiplicative (RoPE-like) rotations and additive (ALiBi/FoX-like) logit biases, and extends both via learned subspaces/mixtures  (Zhang et al., 2025). A more directly comparable head-wise RoPE line is HARoPE, which introduces per-head learnable transformations before rotary mapping (via SVD parameterization) for fine-grained image generation  (Li et al., 2025a).

**Scale-invariant attention and entropy-based scaling.**   A separate direction focuses on stabilizing attention behavior as context length grows by modifying *logit transformations/scaling*. Scale-invariant attention proposes desiderata such as scale-invariant total attention and sparsity and derives a position-dependent logit transformation that improves zero-shot generalization from short training contexts to longer validation contexts  (Anson et al., 2025). Complementary theoretical work analyzes length-related failures of self-attention on simple formal languages and shows that normalization and simple length-dependent logit scaling (e.g., scaling attention logits by $\log n$) can mitigate confidence degradation and improve length generalization  (Chiang & Cholak, 2022). Information-entropy invariance methods derive training-free scaling rules (e.g., InfoScale) motivated by preserving entropy properties during length extrapolation  (Li et al., 2025b).

**Context extrapolation for RoPE-based LMs.**   Many approaches extend usable context beyond pretraining by modifying RoPE scaling or position indices. Position Interpolation rescales position indices so long sequences map back into the original trained range, enabling window extension with limited fine-tuning while preserving short-context quality  (Chen et al., 2023a). YaRN is a compute-efficient context extension recipe that combines RoPE-specific heuristics to enable much longer windows with relatively little additional training  (Peng et al., 2023). LongRoPE pushes window extension to the million-token regime via progressive extension and non-uniform RoPE rescaling strategies  (Shang et al., 2025). Ms-PoE proposes a plug-and-play multi-scale positional encoding that assigns different scaling ratios across heads to mitigate "lost-in-the-middle," improving retrieval of mid-context information without fine-tuning  (Zhang et al., 2024b). MoICE introduces routers within heads that dynamically select among multiple RoPE angles ("in-context experts"), training only lightweight routers while freezing the backbone, to enhance long-context awareness  (Lin et al., 2024).

**Closest connections to our work.**   Our method explicitly turns RoPE into a *head-wise, learnable frequency mechanism* (AdaFreq) and introduces *head-wise, length-aware logit temperature schedules* (AdaScale) as a drop-in replacement while preserving cache/kernel compatibility. This differs from analysis-only works that characterize RoPE usage without providing a unified head-wise learnable frequency plus length-aware scaling module  (Barbero et al., 2025), and from modality-driven generalizations (RoPE-Mixed, LieRE) that primarily target multi-dimensional positional structure rather than head-wise long-context specialization in language models  (Heo et al., 2024; Ostmeier et al., 2024).

Among context-extrapolation methods, Ms-PoE and MoICE are closest in spirit because they introduce explicit *head-dependent* mechanisms for long-context behavior  (Zhang et al., 2024b; Lin et al., 2024). Our approach instead learns a *continuous, head-wise frequency spectrum* (rather than selecting among discrete angle experts or using fixed per-head rescaling ratios) and couples this with *head-wise length-aware temperature scaling* inside the standard RoPE attention

computation.

## B. Additional Results

### B.1. Analysis on Synthetic Tasks

Figure 5 displays the attention logit contributions from different RoPE frequency pairs as the target window size $W$ varies.

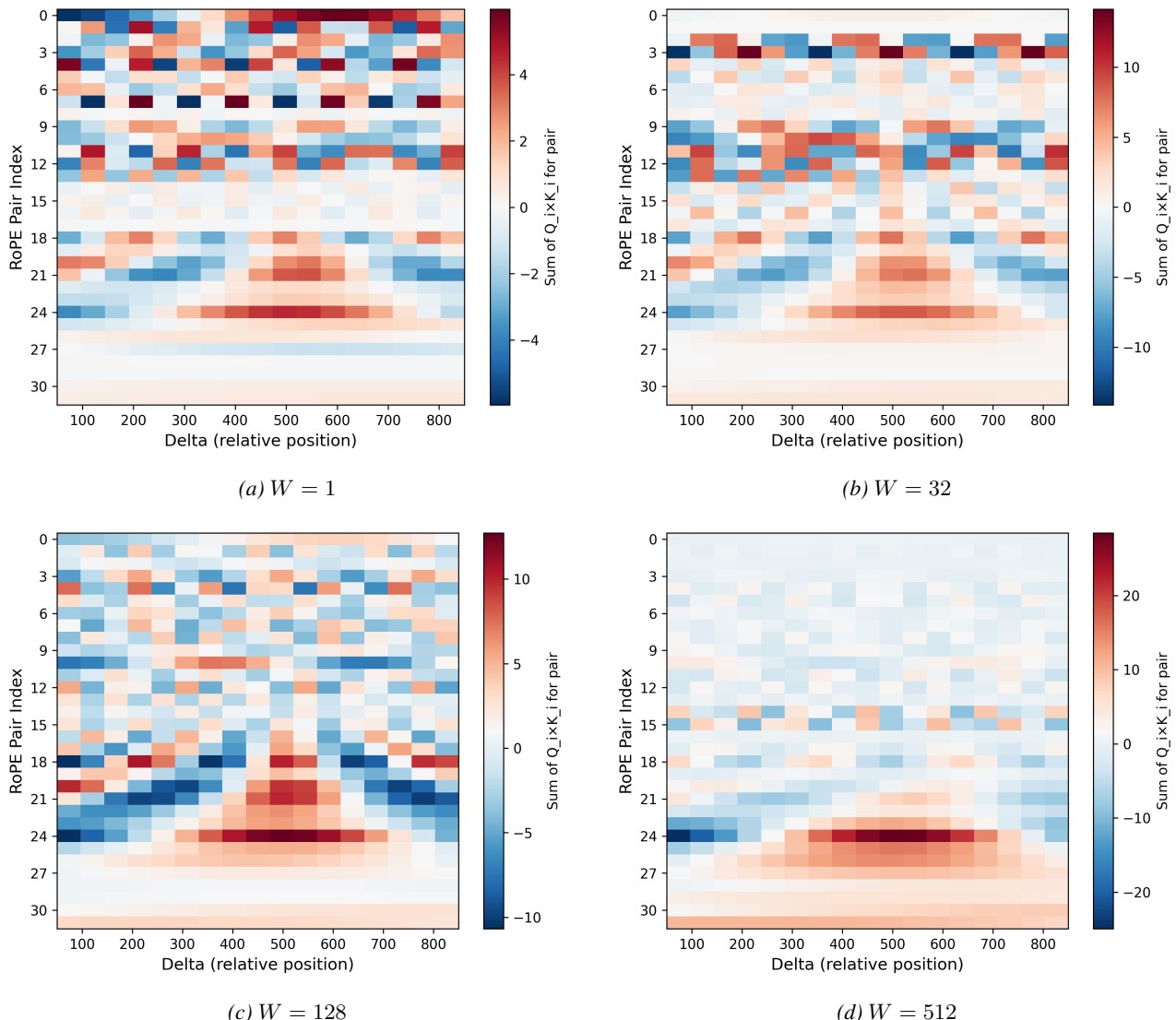

*(a) $W = 1$*

*(b) $W = 32$*

*(c) $W = 128$*

*(d) $W = 512$*

*Figure 5.* **Frequency utilization shifts with different window width $W$ in standard RoPE.** Heatmaps show the contribution of each RoPE dimension pair (y-axis) to the attention score at various relative distances (x-axis) for a trained model. Subfigures correspond to window widths $W \in \{1, 32, 128, 512\}$. As $W$ increases, the contribution of high-frequency pairs (small RoPE pair indices, near the top of each panel) progressively weakens, as indicated by their colors fading toward white. Consequently, the dominant contributions concentrate on lower-frequency bands.

As hypothesized, when $W = 1$ (leftmost), the model utilizes a specific high-frequency band. As $W$ grows to 32, 128, and finally 512, the active frequency band (indicated by darker regions) systematically shifts. This confirms that to optimally perform retrieval at different scales, the attention mechanism must select appropriate frequency components, providing strong empirical support for the head-wise frequency adaptation in AdaRoPE.

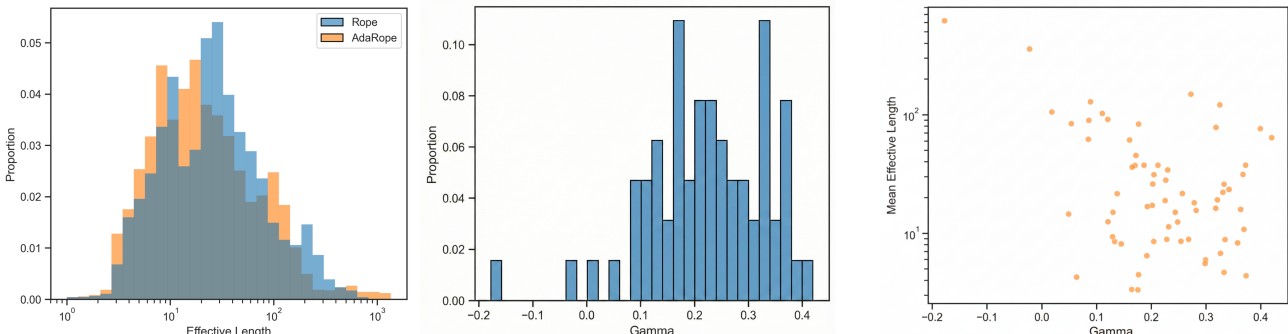

*Figure 6.* **Effective Length Specialization. Left:** AdaRoPE enables a wider distribution of effective lengths across heads compared to RoPE. **Middle:** Distribution of learned $\gamma^{(h)}$ values, which control the $\mathcal{E}_2$ scaling. **Right:** Heads that learn a larger $\gamma^{(h)}$ value maintain a smaller mean effective length.

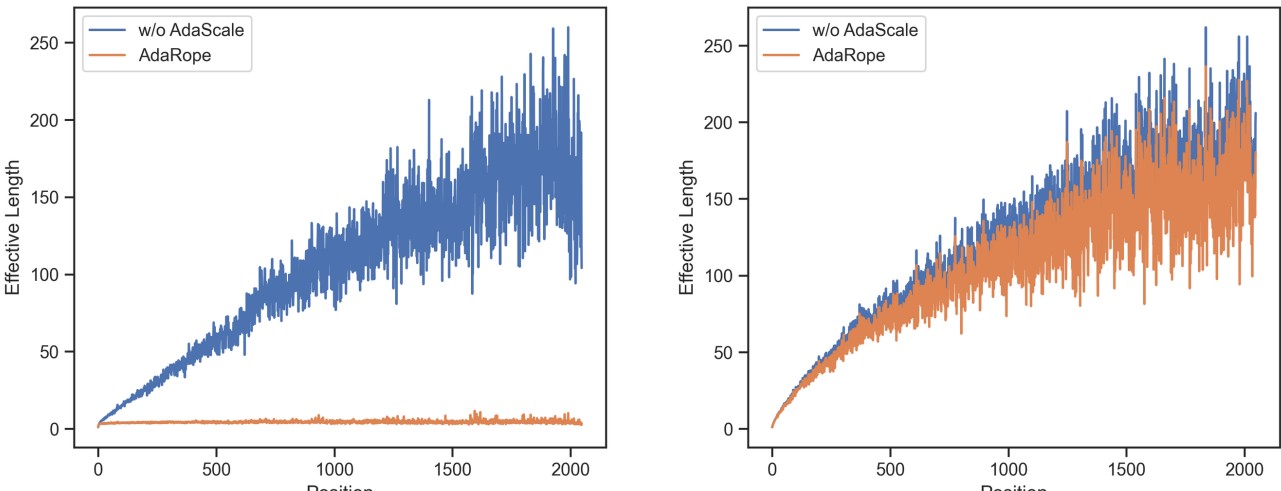

*Figure 7.* **Per-head effective length scaling. Left:** A head with a large learned $\gamma^{(h)} = 0.371$ (Layer 7, Head 1) maintains a small and constant $\mathcal{E}_2$. **Right:** A head with a small learned $\gamma^{(h)} = 0.019$ (Layer 3, Head 7) allows its $\mathcal{E}_2$ to grow with position, acting as a global head.

## B.2. Analysis on Pretraining

**AdaScale enables diverse attention spans.** As context length grows, attention can dilute, as the same query attends to more keys. However, not all heads need to resist this dilution; heads with a global receptive field remain beneficial. We quantify this using $\mathcal{E}_2$, which is the original form of effective sample size. A small $\mathcal{E}_2$ indicates concentrated attention, while a large $\mathcal{E}_2$ indicates distributed attention.

The AdaScale component $\lambda^{(h)}(L) = \frac{1}{\tau^{(h)}} \left[ \ln \left( 1 + \frac{\max\{L, L_{\text{ref}}\}}{L_{\text{ref}}} \right) \right]^{\gamma^{(h)}}$ is designed to manage this, and our results confirm that this $\text{polylog}$-style scaling improves long-context performance.

We find that AdaRoPE allows for greater head specialization in terms of effective length (Figure 6), supporting a wider range of $\mathcal{E}_2$ values than standard RoPE. Some heads remain localized (small $\mathcal{E}_2$) while others become global (large $\mathcal{E}_2$). This differentiation is enabled by the learned parameter $\gamma^{(h)}$, which spans a wide range of values. We observe a strong negative correlation between the learned $\gamma^{(h)}$ and the mean effective length: heads that learn a large $\gamma^{(h)}$ actively suppress attention dilution and maintain a small, focused effective length.

This behavior is clearly visible in individual heads (Figure 7). A head that learns a large $\gamma^{(h)} = 0.371$ (Layer 7, Head 1) maintains a consistently small $\mathcal{E}_2$ regardless of context length. Conversely, a head that learns a small $\gamma^{(h)} = 0.019$ (Layer 3, Head 7) allows its $\mathcal{E}_2$ to grow with the context, effectively behaving as a global-summary head.

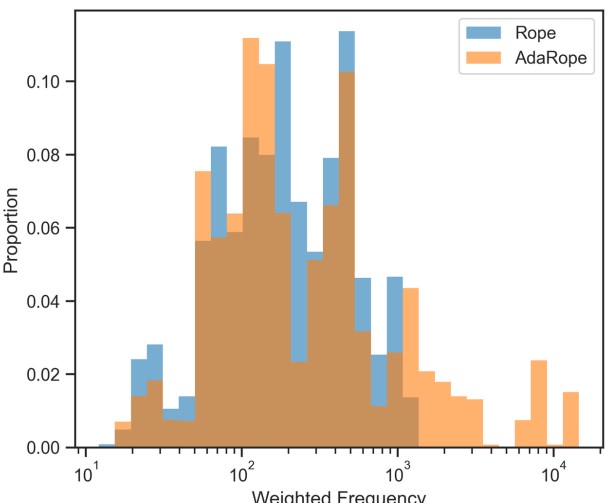 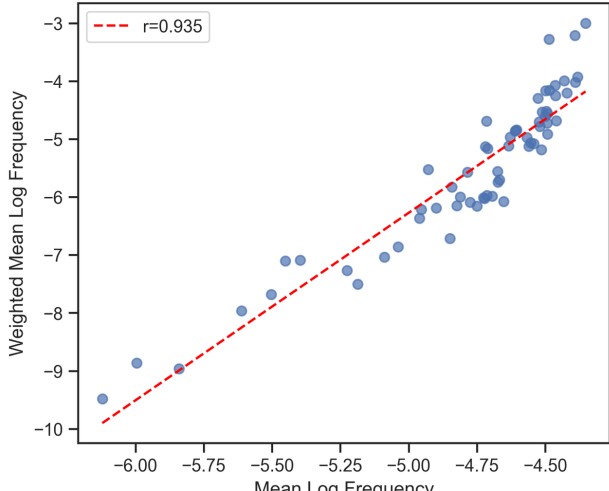

*Figure 8.* **AdaRoPE enables frequency specialization. Left:** The distribution of weighted-average frequencies across heads is broader for AdaRoPE than RoPE. **Right:** High correlation (r=0.935) between the learned mean log-frequency and the utilized weighted mean log-frequency in AdaRoPE.

### B.3. Analysis on Extrapolation

**Setting.**    We conduct a head-level analysis on three SmolLM2-135M variants: the original backbone, YaRN, and AdaRoPE. we utilize the same long-context retrieval corpus used by Wu et al. (2024) to estimate each head's effective attention length $\mathcal{E}_2$.

To identify specific retrieval heads, we also adopt the methodology proposed by Wu et al. (2024). Specifically, we evaluate the models on Needle-in-a-Haystack tasks across a grid of 20 context lengths (up to 10k) and 10 relative depths. Following their approach, we calculate a *retrieval score* for each head, defined as the frequency with which the head's peak attention aligns with the correct needle token matching the generated output during greedy decoding. The head with the highest aggregate score is designated as the *top retrieval head*.

**Sparse but critical heads.**    After fine-tuning AdaRoPE for extrapolation, we quantify the importance of each attention head by a *head influence* score $\Delta_h$: for head $h$, we reset only its AdaRoPE parameters to the pre-finetuning initialization and re-evaluate long-context perplexity, defining $\Delta_h$ as the resulting increase in loss. A histogram of $\Delta_h$ (Figures 10 and 11) shows that most heads have negligible influence, but a small minority form a clear heavy tail. In particular, the head previously identified as a strong position-aware retrieval head lies at the extreme of this distribution, confirming that only a handful of "retrieval-like" heads are indispensable for long-context behavior. At the same time, even the largest $\Delta_h$ is modest compared to the total gap between AdaRoPE and the YaRN baselines, indicating that long-context gains arise from coordinated adjustments across many heads rather than from any single "silver-bullet" head.

**AdaFreq: Non-uniform spectral reshaping.**    To isolate the effect of learned rotary frequencies, we ablate AdaFreq by reverting the log-frequencies $\{\xi_f^{(h)}\}$ of one head at a time while keeping AdaScale fixed. The resulting influence distribution (Figure 10) again exhibits a sparse set of high-impact heads, with the retrieval head being the most sensitive: undoing its learned spectrum noticeably worsens extrapolation perplexity, while perturbing most other heads has almost no effect. Inspecting the learned spectrum of this head reveals that AdaRoPE does not simply apply a uniform rescaling; instead, it selectively amplifies certain mid–high-frequency bands and slightly relaxes others relative to both the original and YaRN-style schedules. This head-specific reshaping of the RoPE spectrum is consistent with our theoretical requirement that different tasks (e.g., sharp local retrieval versus broad global summarization) demand different effective frequency bands per head, and explains why a global frequency rule like YaRN cannot match AdaRoPE 's extrapolation performance.

**AdaScale: Head-wise temperature adaptation.**    We perform an analogous ablation for AdaScale by resetting the temperature parameters $(\gamma^{(h)}, \tau^{(h)})$ of each head to their initialization while keeping AdaFreq fixed. The influence histogram (Figure 11) again shows a sparse set of critical heads whose temperature schedules are essential for long-context stability.

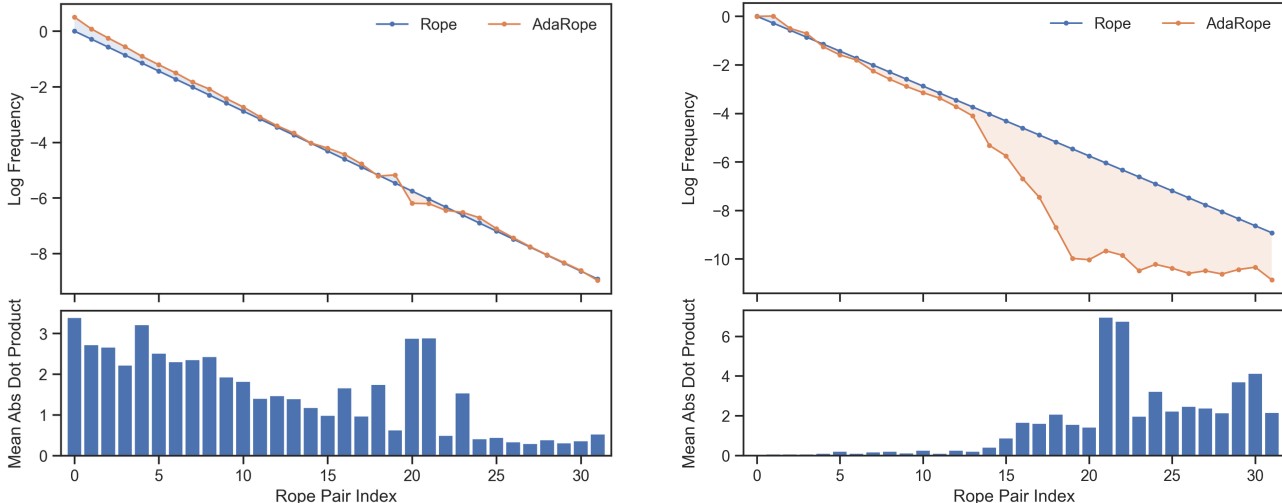

*Figure 9.* **Learned spectral adaptation in AdaRoPE.** Top plots show log-frequency; bottom plots show mean absolute dot product. **Left:** A high-frequency head (Layer 2, Head 2) learns to use even higher frequencies (orange) than standard RoPE (blue). **Right:** A low-frequency head (Layer 4, Head 3) learns to use even lower frequencies.

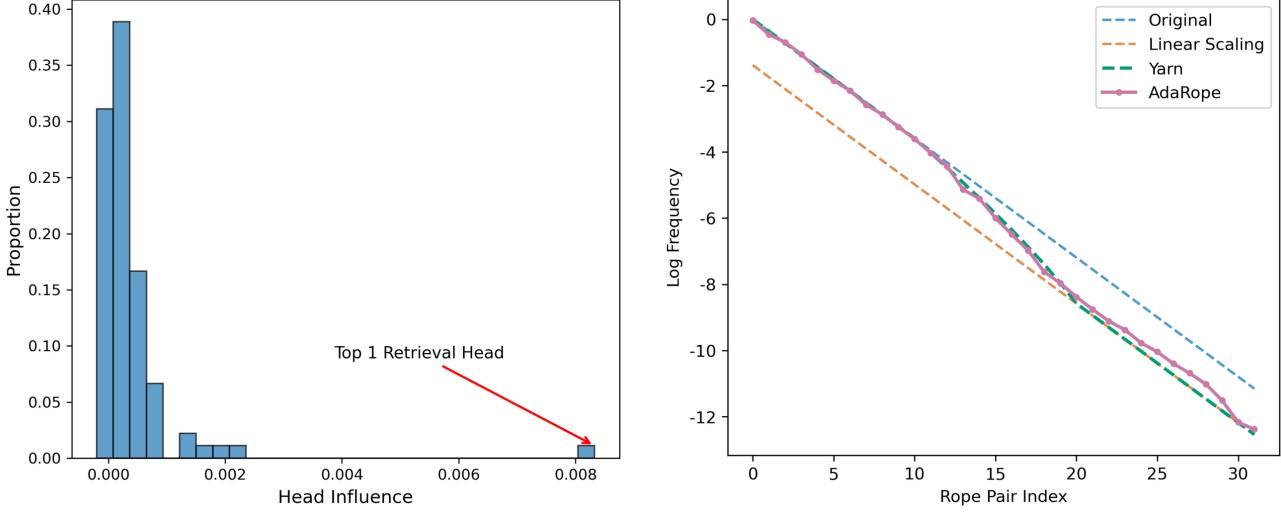

*Figure 10.* **AdaFreq ablations. Left:** Histogram of head influence $\Delta_h$ when reverting AdaFreq parameters one head at a time; most heads have negligible effect, but a small set forms a heavy tail that includes the top retrieval head. **Right:** Learned RoPE spectrum for the most influential head. AdaRoPE reshapes its frequencies non-uniformly, emphasizing mid–high-frequency bands beyond what global schemes like YaRN prescribe.

The learned distributions of $\tau^{(h)}$ and $\gamma^{(h)}$ concentrate around $\tau^{(h)} \approx 1$ with a mild right tail and predominantly positive $\gamma^{(h)}$. This pattern matches the theoretical prediction that most heads benefit from a slightly super-logarithmic sharpening of attention as the context grows, with a few heads learning more aggressive scaling to maintain very small effective attention length, while others keep $\gamma^{(h)}$ near zero and act as global-summary heads whose receptive field intentionally expands with $L$. Together, these ablations support the view that AdaScale 's per-head, length-aware temperatures are not merely a cosmetic modification of the logits: they enable a small set of specialized heads to counteract attention dilution at extreme lengths while allowing the rest of the network to preserve its original global behavior.

**Head-wise effective attention length.** To better understand how different context-extension schemes redistribute attention across heads, we measure the *effective attention length* $\mathcal{E}_2$ of each head before and after modification. For every head, we compute $\mathcal{E}_2$ under the base model and under a given method, and plot $\log \mathcal{E}_2^{\text{method}}$ against $\log \mathcal{E}_2^{\text{base}}$ (Figure 12). Ideally, a

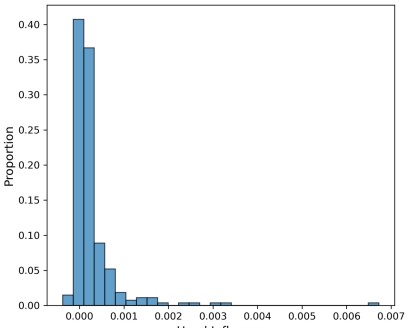 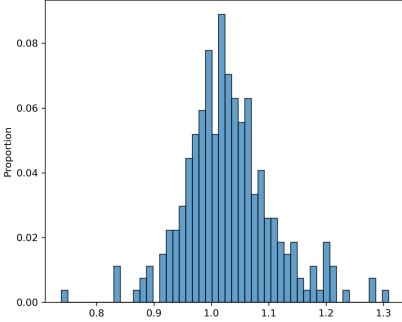 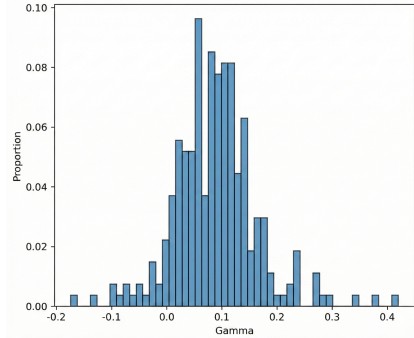

*Figure 11.* **AdaScale ablations. Left:** Histogram of head influence $\Delta_h$ when reverting AdaScale parameters $(\gamma^{(h)}, \tau^{(h)})$ for one head at a time, revealing a sparse set of critical temperature heads. **Middle:** Distribution of learned base temperatures $\tau^{(h)}$, centered near 1 with a right tail corresponding to heads that sharpen more aggressively with context length. **Right:** Distribution of learned growth coefficients $\gamma^{(h)}$, which are mostly positive, indicating that many heads benefit from slightly increasing sharpness as the sequence length grows.

method that preserves short-context behavior while extending long-range capacity should keep heads with intrinsically small $\mathcal{E}_2$ close to the diagonal ($y = x$), but *increase* $\mathcal{E}_2$ for heads that already operate at large ranges. YaRN without scaling (left) better matches the long-range regime at the cost of noticeably distorting many short-range heads, whereas globally scaled YaRN (middle) closely follows the diagonal for short-range heads but fails to sufficiently extend large-$\mathcal{E}_2$ heads. In contrast, AdaRoPE (right) stays near the diagonal for small $\mathcal{E}_2$ while selectively moving large-$\mathcal{E}_2$ heads upward, indicating that it preserves the effective length of local heads and scales up only those heads that need longer reach. This head-wise adaptive behavior is precisely what allows AdaRoPE to handle both short and extreme-length contexts without sacrificing either regime.

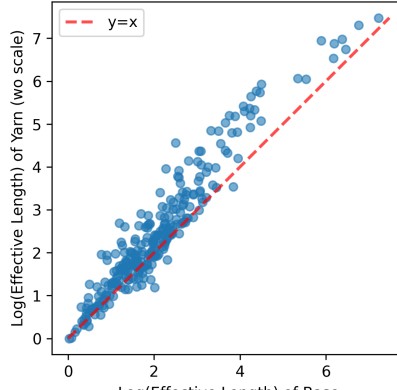 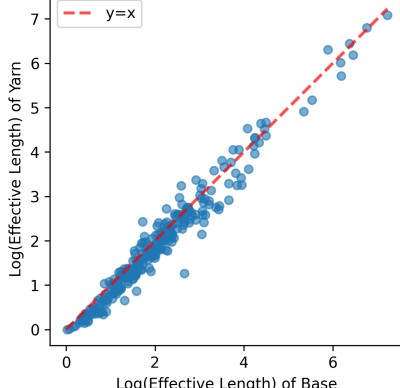 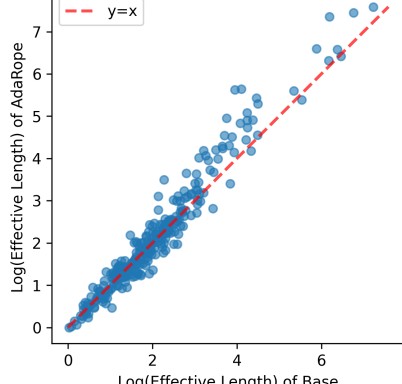

*Figure 12.* **Effective attention length per head.** Each point is an attention head; the horizontal axis is $\log(\mathcal{E}_2^{\text{base}})$ and the vertical axis is $\log(\mathcal{E}_2^{\text{method}})$, with the red dashed line indicating $y = x$. **Left:** YaRN without scaling; short-range heads are preserved but long-range heads are not sufficiently extended. **Middle:** YaRN with a global scale; long-range heads are better matched but many short-range heads are distorted. **Right:** AdaRoPE; small-$\mathcal{E}_2$ heads remain close to the diagonal while large-$\mathcal{E}_2$ heads are systematically pushed above it, showing that AdaRoPE maintains effective length for local heads and scales it up only when needed.

## C. Notation

**General Notation.** Let $\mathbb{R}$, $\mathbb{C}$, $\mathbb{Z}$, and $\mathbb{N} = \{0, 1, \dots\}$ denote the sets of real numbers, complex numbers, integers, and non-negative integers, respectively. We denote by $\mathbb{R}^d$ the $d$-dimensional Euclidean space, and use $\|\cdot\|_p$ for the $p$-norm, $1 \le p \le \infty$ (with $p = 2$ when unspecified). Let $\mathbf{0}_d, \mathbf{1}_d \in \mathbb{R}^d$ denote the vectors of all zeros and all ones, respectively. Let $\boldsymbol{I}_d$ denote the $d \times d$ identity matrix. We use $O(\cdot)$, $o(\cdot)$, $\Omega(\cdot)$, $\omega(\cdot)$, and $\Theta(\cdot)$ for standard asymptotic notation. Almost sure convergence, convergence in probability, and convergence in distribution are denoted by $\xrightarrow{\text{a.s.}}$, $\xrightarrow{\mathbb{P}}$, and $\xrightarrow{\mathcal{D}}$, respectively. We denote the multivariate normal distribution with mean vector $\boldsymbol{\mu}$ and covariance matrix $\boldsymbol{\Sigma}$ by $\mathcal{N}(\boldsymbol{\mu}, \boldsymbol{\Sigma})$. The indicator function is denoted by $\mathbb{1}\{\cdot\}$.

Now we briefly review the essential details of multi-head attention and rotary position embeddings, which form the basis of our analysis.

**Multi-Head Attention.** Let $\boldsymbol{x}_i \in \mathbb{R}^D$ be the $D$-dimensional embedding of the $i$-th token, where $i \in \{1, \dots, L\}$. Consider the input embedding sequence $\boldsymbol{X} = (\boldsymbol{x}_1, \dots, \boldsymbol{x}_L)^\mathsf{T} \in \mathbb{R}^{L \times D}$. Suppose the model uses $H$ attention heads. For each head $h \in \{1, \dots, H\}$, let $\boldsymbol{W}_Q^{(h)}, \boldsymbol{W}_K^{(h)}, \boldsymbol{W}_V^{(h)} \in \mathbb{R}^{D \times d}$ be the projection matrices, where $d = D/H \in \mathbb{N}$ is the hidden dimension of each head. The query, key, and value matrices for the $h$-th head are computed as

$$\boldsymbol{Q}^{(h)} = \boldsymbol{X}\boldsymbol{W}_Q^{(h)}, \qquad \boldsymbol{K}^{(h)} = \boldsymbol{X}\boldsymbol{W}_K^{(h)}, \qquad \boldsymbol{V}^{(h)} = \boldsymbol{X}\boldsymbol{W}_V^{(h)}. \tag{3}$$

The attention output of head $h$, denoted by $\boldsymbol{O}^{(h)} \in \mathbb{R}^{L \times d}$, is defined as

$$\boldsymbol{O}^{(h)} = \text{Softmax}\left(\frac{1}{\sqrt{d}}\boldsymbol{Q}^{(h)}(\boldsymbol{K}^{(h)})^\mathsf{T} + \boldsymbol{M}\right)\boldsymbol{V}^{(h)}, \tag{4}$$

where $\boldsymbol{M} \in \mathbb{R}^{L \times L}$ is the causal mask matrix. Let $\boldsymbol{q}_i^{(h)}, \boldsymbol{k}_i^{(h)}, \boldsymbol{v}_i^{(h)} \in \mathbb{R}^d$ denote the $i$-th rows of $\boldsymbol{Q}^{(h)}, \boldsymbol{K}^{(h)}$, and $\boldsymbol{V}^{(h)}$, respectively. For token $t \in \{1, \dots, L\}$, the output vector $\boldsymbol{o}_t^{(h)} \in \mathbb{R}^d$ is

$$\boldsymbol{o}_t^{(h)} = \sum_{i=1}^{t} \frac{\exp\left((\boldsymbol{q}_t^{(h)})^\mathsf{T}\boldsymbol{k}_i^{(h)}/\sqrt{d}\right)}{\sum_{j=1}^{t} \exp\left((\boldsymbol{q}_t^{(h)})^\mathsf{T}\boldsymbol{k}_j^{(h)}/\sqrt{d}\right)} \boldsymbol{v}_i^{(h)}. \tag{5}$$

We denote the attention logit between query $\boldsymbol{q}_i^{(h)}$ and key $\boldsymbol{k}_j^{(h)}$ by $s_{i,j}^{(h)} = \frac{1}{\sqrt{d}}(\boldsymbol{q}_i^{(h)})^\mathsf{T}\boldsymbol{k}_j^{(h)}$, and define $\alpha_{t,i}^{(h)} = \text{Softmax}_i(s_{t,1}^{(h)}, \dots, s_{t,t}^{(h)})$ as the attention weight of token $t$ attending to token $i$. Thus, $\boldsymbol{o}_t^{(h)} = \sum_{i=1}^{t} \alpha_{t,i}^{(h)}\boldsymbol{v}_i^{(h)}$.

Finally, the head outputs are concatenated along the feature dimension and linearly projected to produce the multi-head output $\boldsymbol{O} \in \mathbb{R}^{L \times D}$, which is

$$\boldsymbol{O} = \left[\boldsymbol{O}^{(1)}, \boldsymbol{O}^{(2)}, \dots, \boldsymbol{O}^{(H)}\right]\boldsymbol{W}_O, \tag{6}$$

where $\boldsymbol{W}_O \in \mathbb{R}^{D \times D}$ is the output projection matrix.

**Rotary Positional Embedding (RoPE).** Here we focus on a single attention head, and omit the superscript $\bullet^{(h)}$ for simplicity. Assuming $d \in 2\mathbb{N}$, we partition the $d$-dimensional space into $d/2$ pairs of elements. For each subspace index $f \in \{0, \dots, d/2 - 1\}$, we define a frequency $\theta_f = b^{-2f/d}$ with a constant $b > 1$ (typically $10,000$). We first define the family of rotation matrices $\boldsymbol{R}_m \in \mathbb{R}^{d \times d}$ for rotation step $m \in \mathbb{Z}$, which is block-diagonal and formed by $d/2$ rotation sub-matrices

$$\boldsymbol{R}_m = \text{diag}\left(\boldsymbol{G}_m(\theta_0), \dots, \boldsymbol{G}_m(\theta_{d/2-1})\right), \quad \text{where} \quad \boldsymbol{G}_m(\theta_f) = \begin{pmatrix} \cos(m\theta_f) & -\sin(m\theta_f) \\ \sin(m\theta_f) & \cos(m\theta_f) \end{pmatrix}. \tag{7}$$

Since $(\boldsymbol{G}_m)^\mathsf{T} = \boldsymbol{G}_{-m}$ and $\boldsymbol{G}_m \boldsymbol{G}_n = \boldsymbol{G}_{m+n}$, it follows that $\boldsymbol{R}_m = (\boldsymbol{R}_1)^m$. For the $i$-th token, we apply the rotation matrix $\boldsymbol{R}_{i-1}$ to its query and key vectors.[1] The attention logit is then

$$
\begin{aligned}
s_{i,j} &= \frac{1}{\sqrt{d}} \left(\boldsymbol{R}_{i-1}\boldsymbol{q}_i\right)^\mathsf{T} \left(\boldsymbol{R}_{j-1}\boldsymbol{k}_j\right) = \frac{1}{\sqrt{d}}\boldsymbol{q}_i^\mathsf{T}\boldsymbol{R}_{j-i}\boldsymbol{k}_j \\
&= \frac{1}{\sqrt{d}} \sum_{f=0}^{d/2-1} \begin{pmatrix} q_{i,2f+1} & q_{i,2f+2} \end{pmatrix} \boldsymbol{G}_{j-i}(\theta_f) \begin{pmatrix} k_{j,2f+1} \\ k_{j,2f+2} \end{pmatrix}.
\end{aligned}
\tag{8}
$$

Here, $q_{i,\bullet}, k_{j,\bullet} \in \mathbb{R}$ denote the $\bullet$-th coordinates of $\boldsymbol{q}_i$ and $\boldsymbol{k}_j$, respectively. For convenience, we write $\boldsymbol{R} := \boldsymbol{R}_1$ and $\boldsymbol{G} := \boldsymbol{G}_1$. If $\theta_f \equiv 0$ for all $f \in \{0, \dots, d/2-1\}$, then $\boldsymbol{R}$ reduces to the identity matrix, corresponding to No Positional Encoding (NoPE). In AdaRoPE, we parameterize $\theta_f = \exp(\xi_f)$ and treat $\xi_f$ as trainable parameters.

## D. Theoretical Analysis of Rotation Frequencies

In this section, we show the necessity of head-wise rotation frequencies by analyzing the Needle Retrieval from a Window (NRW) task. As defined in Section 3.1, the objective of NRW is to determine whether the NEEDLE token lies within a target window $[c - W/2, c + W/2]$, where $c \in \mathbb{N}$, $W \in 2\mathbb{N}$ are two integers. Given an input sequence of tokens $\boldsymbol{x}_1, \dots, \boldsymbol{x}_L \in \mathbb{R}^D$, the attention output at the position $L$ is used as an indicator.

**Problem Setup.** Fix a context length $L$. There are two token types: NEEDLE and BG. Let $t$ denote the position of the target NEEDLE token. The goal is to use a single-layer attention mechanism to determine whether $t \in [c - W/2, c + W/2]$. Let $\boldsymbol{q}$ be the query vector of the $L$-th token, and let $(\boldsymbol{k}_+, \boldsymbol{v}_+)$ and $(\boldsymbol{k}_-, \boldsymbol{v}_-)$ denote the key-value pairs for NEEDLE and BG, respectively. We introduce a buffer ratio $r > 0$ to separate the interior and exterior of the window, and assume $\|\boldsymbol{v}_+ - \boldsymbol{v}_-\| = 1$. The NRW problem can then be formulated as the following feasibility problem: for any $\varepsilon > 0$ and $t \in \{1, \dots, L\}$,

$$
\begin{aligned}
\|\boldsymbol{o}(t) - \boldsymbol{v}_+\| &\le \varepsilon, & \text{if } & |t - c| \le W/2, \\
\|\boldsymbol{o}(t) - \boldsymbol{v}_-\| &\le \varepsilon, & \text{if } & |t - c| > (1 + r)W/2,
\end{aligned}
\tag{9}
$$

where $\boldsymbol{o}(t)$ is the attention output at the $L$-th position. Note that $\boldsymbol{o}(t)$ depends on the input sequence, and hence on the target position $t$.

**Derivation of Frequency Mass.** We begin by expressing the variables in Eq. (9) as a weighted sum of rotational frequencies. Let $\alpha_i$ denote the attention weights. The attention output is

$$
\boldsymbol{o}_L = \sum_{i=1}^{L} \alpha_i \boldsymbol{v}_i = \sum_{i \ne t} \alpha_i \boldsymbol{v}_- + \alpha_t \boldsymbol{v}_+.
$$

Hence $\|\boldsymbol{o}(t) - \boldsymbol{v}_+\| = (1 - \alpha_t)\|\boldsymbol{v}_+ - \boldsymbol{v}_-\|$ and $\|\boldsymbol{o}(t) - \boldsymbol{v}_-\| = \alpha_t\|\boldsymbol{v}_+ - \boldsymbol{v}_-\|$. The constraints become

$$
\begin{aligned}
\alpha_t &\ge 1 - \varepsilon, & \text{if } & |t - c| \le W/2, \\
\alpha_t &\le \varepsilon, & \text{if } & |t - c| > (1 + r)W/2.
\end{aligned}
$$

Using the RoPE-equipped definition of attention weights,

$$
\alpha_t = \frac{\exp\left(\boldsymbol{q}^\mathsf{T}\boldsymbol{R}^{t-L}\boldsymbol{k}_t\right)}{\sum_{i=1}^{t} \exp\left(\boldsymbol{q}^\mathsf{T}\boldsymbol{R}^{i-L}\boldsymbol{k}_i\right)} = \frac{\exp\left(\boldsymbol{q}^\mathsf{T}\boldsymbol{R}^{t-L}\boldsymbol{k}_+\right)}{\exp\left(\boldsymbol{q}^\mathsf{T}\boldsymbol{R}^{t-L}\boldsymbol{k}_+\right) + \sum_{i \ne t} \exp\left(\boldsymbol{q}^\mathsf{T}\boldsymbol{R}^{i-L}\boldsymbol{k}_-\right)},
$$

where the factor $1/\sqrt{d}$ is absorbed into $\boldsymbol{q}$. Let

$$
s_t = \boldsymbol{q}^\mathsf{T}\boldsymbol{R}^{t-L}\boldsymbol{k}_+, \qquad s_{-t} = \ln\left(\sum_{i \ne t} \exp\left(\boldsymbol{q}^\mathsf{T}\boldsymbol{R}^{i-L}\boldsymbol{k}_-\right)\right).
$$

---

[1] Because token indices in our theoretical analysis start at 1, the rotation step is shifted by $-1$, so the first token receives the identity matrix $\boldsymbol{R}_0 = \boldsymbol{I}_d$ (Su et al., 2021).

Then,

$$\alpha_t = \frac{e^{s_t}}{e^{s_t} + e^{s_{-t}}} = \text{Sigmoid}(s_t - s_{-t}), \quad \text{where} \quad \text{Sigmoid}(z) := \frac{1}{1 + e^{-z}}.$$

By the monotonicity of the sigmoid function, the constraints are equivalent to

$$s_t \geq s_{-t} + \ln\left(\frac{1-\varepsilon}{\varepsilon}\right), \qquad \text{if} \quad |t-c| \leq W/2,$$

$$s_t \leq s_{-t} + \ln\left(\frac{\varepsilon}{1-\varepsilon}\right), \qquad \text{if} \quad |t-c| > (1+r)W/2.$$

Let $\hat{q}_f := q_{2f+1} + iq_{2f+2}$ and $\hat{k}_f := k_{+,2f+1} + ik_{+,2f+2} \in \mathbb{C}$. Then,

$$s_t = \sum_{f=0}^{d/2-1} \begin{pmatrix} q_{2f+1} & q_{2f+2} \end{pmatrix} \boldsymbol{G}_{t-L}(\theta_f) \begin{pmatrix} k_{+,2f+1} \\ k_{+,2f+2} \end{pmatrix} = \sum_{f=0}^{d/2-1} \text{Re}\left(\hat{q}_f \overline{\hat{k}}_f e^{-i(t-L)\theta_f}\right)$$

$$= \sum_{f=0}^{d/2-1} A_f \cos((t-L)\theta_f + \varphi'_f) = \sum_{f=0}^{d/2-1} A_f \cos(t\theta_f + \varphi_f).$$

where $A_f = \left|\hat{q}_f \overline{\hat{k}}_f\right| = |\hat{q}_f| \cdot |\hat{k}_f| \geq 0$, $\varphi'_f = -\arg\left(\hat{q}_f \overline{\hat{k}}_f\right) = \arg(\hat{k}_f) - \arg(\hat{q}_f)$, and $\varphi_f = \varphi'_f - L\theta_f$. Let

$$A_\star := \sum_{f=0}^{d/2-1} A_f = \sum_{f=0}^{d/2-1} \sqrt{q_{2f+1}^2 + q_{2f+2}^2} \cdot \sqrt{k_{+,2f+1}^2 + k_{+,2f+2}^2} > 0.$$

Dividing by $A_\star$ yields normalized weights $a_f := A_f/A_\star$, which satisfy $a_f \geq 0$ and $\sum_{f=0}^{d/2-1} a_f = 1$. Define the finite Fourier sum $T(x)$ and the *frequency mass* $\mathcal{M}$ by

$$T(x) = \sum_{f=0}^{d/2-1} a_f \cos(\theta_f x + \varphi_f), \qquad \mathcal{M} = \sum_{f=0}^{d/2-1} a_f \theta_f. \tag{10}$$

In terms of $T(x)$, the feasibility constraints in Eq. (9) become

$$T(t) \geq \frac{1}{A_\star}\left(s_{-t} + \ln\left(\frac{1-\varepsilon}{\varepsilon}\right)\right) =: T_L(t), \qquad \text{if} \quad |t-c| \leq W/2,$$

$$T(t) \leq \frac{1}{A_\star}\left(s_{-t} + \ln\left(\frac{\varepsilon}{1-\varepsilon}\right)\right) =: T_U(t), \qquad \text{if} \quad |t-c| \geq (1+r)W/2. \tag{11}$$

As shown in Theorem 1, the frequency mass $\mathcal{M}$ depends on the RoPE frequencies and the NRW parameters, and characterizes the feasibility of the attention weights.

**Theorem 1** (Necessity of head-wise rotation frequencies). *Suppose* $0 < \theta_f \leq \pi$ *for all* $f \in \{0, \ldots, d/2-1\}$*, and* $\|\boldsymbol{q}\|, \|\boldsymbol{k}_+\| > 0$*. For any* $\varepsilon > 0$ *with* $\varepsilon < (\exp(\|\boldsymbol{q}\| \cdot \|\boldsymbol{k}_-\|) + 1)^{-1}$*, every feasible solution of Eq. (9) satisfies*

$$\frac{C_1}{rW/2+1} \leq \mathcal{M} \leq \frac{\pi}{W+1} + C_2 \cdot \mathbb{1}\left\{\max_{0 \leq f \leq d/2-1} \theta_f > \frac{\pi}{W+1}\right\}, \tag{12}$$

*where* $C_1, C_2 > 0$ *are constants that depend only on* $\varepsilon$*,* $L$*,* $\boldsymbol{q}$*,* $\boldsymbol{k}_+$ *and* $\boldsymbol{k}_-$*, and are independent of the window center* $c$*, width* $W$*, and the buffer ratio* $r$*.*

**Proof.** We defer the proof to Section F.1. $\qquad \square$

**Remark 1** (NoPE fails on NRW tasks). For NoPE, $\theta_f \equiv 0$ for all $f \in \{0, \ldots, d/2-1\}$, and therefore $\mathcal{M} = \sum_{f=0}^{d/2-1} a_f \theta_f = 0$. This contradicts the lower bound $\frac{C_1}{rW/2+1}$ in Eq. (12). Thus, a single-layer Transformer without positional encoding cannot solve NRW tasks, consistent with the experimental results in Section 3.1.

**Remark 2** (Feasible frequencies depend on the window width $W$). Let $\theta_{\min}$ and $\theta_{\max}$ denote the minimum and maximum frequencies. Since $\theta_{\min} \leq \mathcal{M} \leq \theta_{\max}$, Eq. (12) implies

$$\theta_{\max} \geq \frac{C_1}{rW/2 + 1}, \qquad \theta_{\min} \leq \frac{\pi}{W+1} + C_2 \cdot \mathbb{1}\left\{\theta_{\max} > \frac{\pi}{W+1}\right\}.$$

For large $W$, the condition $\theta_{\max} > \frac{\pi}{W+1}$ becomes less restrictive, allowing a wider range of feasible frequencies. For small $W$, $\theta_{\max}$ must be sufficiently large for the constraints to be feasible. As shown in Figure 5, high frequencies are necessary when $W$ is small; as $W$ increases, the contribution of high frequencies decreases, and the activation is mainly concentrated in low frequencies.

**Remark 3** (Implication for AdaFreq). Because NRW tasks are sensitive to frequency choices, a simple way to increase the expressivity of RoPE is to make these frequencies learnable. In AdaFreq, for each head $h$, we define a head-specific rotation matrix

$$\boldsymbol{R}^{(h)} = \mathrm{diag}\left(\boldsymbol{G}(\theta_0^{(h)}), \ldots, \boldsymbol{G}(\theta_{d/2-1}^{(h)})\right),$$

where the frequency of each dimension $f \in \{0, \ldots, d/2 - 1\}$ is parameterized as $\theta_f^{(h)} = \exp\left(\xi_f^{(h)}\right)$.

# E. Theoretical Analysis of Scaling Factor

In this section, we first define the *effective sequence length* (Section E.1) and then explain (i) why temperature scaling is necessary in attention mechanisms (Sections E.2 and E.3), and (ii) which forms of temperature schedules are appropriate (Section E.4).

### E.1. Notation and Properties of Effective Sequence Length

**Definition 1** (Generalized effective sample size (Huggins & Roy, 2019; Martino et al., 2017)). For normalized weights $\boldsymbol{w} = (w_1, \ldots, w_n)$ with $w_i \geq 0$ and $\sum_{i=1}^{n} w_i = 1$, the generalized effective sample size (ESS) of positive order $\beta \in (0, 1) \cup (1, \infty)$ is defined as

$$\mathcal{E}_\beta(\boldsymbol{w}) = \left(\sum_{i=1}^{n} w_i^\beta\right)^{\frac{1}{1-\beta}}. \tag{13}$$

For $\beta = 2$, this coincides with the classical ESS estimator used in importance sampling (Kong, 1992; Kong et al., 1994). As $\beta$ tends to 1, the generalized ESS tends to the perplexity (Cappé et al., 2008):

$$\lim_{\beta \to 1} \mathcal{E}_\beta(\boldsymbol{w}) = \mathcal{E}_1(\boldsymbol{w}) := \exp\left(-\sum_{i=1}^{n} w_i \ln w_i\right). \tag{14}$$

**Definition 2** (Rényi divergence and Rényi entropy (Rényi, 1961)). For a positive order $\beta \neq 1$, the Rényi divergence of a discrete probability distribution $\boldsymbol{p} = (p_1, \ldots, p_n)$ from $\boldsymbol{q} = (q_1, \ldots, q_n)$ is defined as

$$D_\beta(\boldsymbol{p} \parallel \boldsymbol{q}) = \frac{1}{\beta - 1} \ln\left(\sum_{i=1}^{n} p_i^\beta q_i^{1-\beta}\right). \tag{15}$$

The Rényi entropy is defined as

$$H_\beta(\boldsymbol{w}) = \frac{1}{1-\beta} \ln\left(\sum_{i=1}^{n} w_i^\beta\right) = \ln\left(\mathcal{E}_\beta(\boldsymbol{w})\right). \tag{16}$$

As $\beta$ tends to 1, the Rényi divergence converges to the Kullback–Leibler (KL) divergence, and the Rényi entropy converges to the Shannon entropy.

Here we state, without proof, several facts concerning the generalized ESS and the Rényi divergence (Huggins & Roy, 2019; Van Erven & Harremoës, 2014; Martino et al., 2017):

*Fact* 1 (Bounds of ESS). For any normalized weights $\boldsymbol{w} = (w_1, \ldots, w_n)$ and $\beta \in (0, \infty)$, we have $1 \leq \mathcal{E}_\beta(\boldsymbol{w}) \leq n$. The lower bound is achieved if $\boldsymbol{w} = (\ldots, 0, 1, 0, \ldots)$ is a one-hot vector; and the upper bound is achieved if $w_i = 1/n, \ \forall i \in \{1, \ldots, n\}$.

*Fact* 2 (Monotonicity of ESS). For any normalized weights $\boldsymbol{w} = (w_1, \ldots, w_n)$ and $0 < p \leq q < \infty$, we have $\mathcal{E}_q(\boldsymbol{w}) \leq \mathcal{E}_p(\boldsymbol{w})$.

*Fact* 3 (Relationship between Rényi divergence and entropy). The Rényi entropy can be expressed in terms of the Rényi divergence, $H_\beta(\boldsymbol{w}) = \ln n - D_\beta(\boldsymbol{w} \parallel \boldsymbol{u})$, where $\boldsymbol{u} = (1/n, \ldots, 1/n)$ is the uniform distribution.

*Fact* 4 (Relationship between ESS and Rényi divergence). $\mathcal{E}_\beta$ can be expressed in terms of the Rényi divergence,

$$\mathcal{E}_\beta(\boldsymbol{w}) = \exp\left(H_\beta(\boldsymbol{w})\right) = \exp\left(\ln n - D_\beta(\boldsymbol{w} \parallel \boldsymbol{u})\right) = n \exp\left(-D_\beta(\boldsymbol{w} \parallel \boldsymbol{u})\right)$$

where $\boldsymbol{u} = (1/n, \ldots, 1/n)$ is the uniform distribution.

In this paper, we use the generalized ESS to quantify the effective sequence length induced by the attention mechanism for an input sequence of length $L$. Let $\lambda > 0$ denote the *scaling factor* (or *inverse temperature*). When $\lambda$ is applied to the attention logits, the attention weight assigned by token $t \in \{1, \ldots, L\}$ to token $i$, as defined in Section C, becomes

$$\alpha_{t,i} = \text{Softmax}_i\left(\lambda s_{t,1}, \ldots, \lambda s_{t,t}\right) = \frac{e^{\lambda s_{t,i}}}{\sum_{j=1}^t e^{\lambda s_{t,j}}}. \tag{17}$$

We now define the effective sequence length for single-layer attention.

**Definition 3** (Effective sequence length). Let $\lambda > 0$ be the scaling factor, and let $\boldsymbol{s} = (s_1, \ldots, s_L)$ denote the logits for the last token $\boldsymbol{x}_L$. The attention weights $\boldsymbol{\alpha} = (\alpha_1, \ldots, \alpha_L)$ are given by $\alpha_i = e^{\lambda s_i}/Z(\boldsymbol{s}; \lambda)$, where $Z(\boldsymbol{s}; \lambda) := \sum_{i=1}^L e^{\lambda s_i}$ is the partition function. For $\beta \in (0, 1) \cup (1, \infty)$, the effective sequence length of single-layer attention is defined as

$$\mathcal{E}_\beta(\boldsymbol{\alpha}) = \left(\sum_{i=1}^L \left(\frac{e^{\lambda s_i}}{\sum_{j=1}^L e^{\lambda s_j}}\right)^\beta\right)^{\frac{1}{1-\beta}} = \frac{\left(\sum_{i=1}^L e^{\lambda s_i}\right)^{\frac{\beta}{\beta-1}}}{\left(\sum_{i=1}^L e^{\beta \lambda s_i}\right)^{\frac{1}{\beta-1}}} = \frac{Z(\boldsymbol{s}; \lambda)^{\frac{\beta}{\beta-1}}}{Z(\boldsymbol{s}; \beta\lambda)^{\frac{1}{\beta-1}}}. \tag{18}$$

Define $\mathcal{E}_1(\boldsymbol{\alpha}) := \lim_{\beta \to 1} \mathcal{E}_\beta(\boldsymbol{\alpha})$.

In Sections E.2 and E.3, we first assume $\lambda = 1$ (i.e., the scaling is absorbed into $\boldsymbol{s}$) and that the logits are deterministic. We then show that attention heads performing different tasks exhibit different scaling behavior in effective sequence length $\mathcal{E}_\beta$. Specifically, a retrieval head requires $\mathcal{E}_\beta$ to remain constant as $L$ increases (Theorem 2), whereas a global aggregation head requires $\mathcal{E}_\beta$ to scale linearly with $L$ (Theorem 3). In Section E.4, we further discuss how the scaling factor $\lambda = \lambda(L)$, as a function of the context length, is determined by the scaling behavior of $\mathcal{E}_\beta$ (Theorems 4 and 5).

## E.2. Retrieval Heads

A retrieval head is an attention head that implements a specific key–value matching rule and operates on the local context. Its output remains stable as long as the target is included in the context; increasing the context length therefore does not substantially affect the output.

**Problem Setup.** Consider a retrieval task on a sequence of length $n$, and extend the sequence to length $n + m$. Assume that the query vector for the last token remains unchanged. It then suffices to consider the logits $s_i$ and value vectors $\boldsymbol{v}_i$ for $i \in \{1, \ldots, n+m\}$. The original attention output $\boldsymbol{o}_n$ is computed from the *signal* logit–value pairs $(s_i, \boldsymbol{v}_i)$ for $i \in \{1, \ldots, n\}$. The extended output $\boldsymbol{o}_{n+m}$ additionally includes the *noise* pairs $(s_{n+j}, \boldsymbol{v}_{n+j})$ for $j \in \{1, \ldots, m\}$. Our goal is to define a metric that quantifies the difference between $\boldsymbol{o}_{n+m}$ and $\boldsymbol{o}_n$; if this difference is small, the head's retrieval performance is preserved.

Let $\mathcal{E}_{\beta,n}$ and $\mathcal{E}_{\beta,n+m}$ denote the effective sequence lengths of the logits $(s_1, \ldots, s_n)$ and $(s_1, \ldots, s_{n+m})$, respectively, and define the relative change in $\mathcal{E}_\beta$ as

$$\Delta_\beta := \frac{\mathcal{E}_{\beta,n+m}}{\mathcal{E}_{\beta,n}} - 1. \tag{19}$$

The following Theorem 2 shows that $\Delta_\beta$ provides a metric that bounds the difference $\|\boldsymbol{o}_{n+m} - \boldsymbol{o}_n\|$. Therefore, if $\mathcal{E}_\beta$ remains approximately constant across context lengths for some $\beta$, the behavior of single-layer attention under length generalization is approximately stable.

**Theorem 2** ($\mathcal{E}_\beta$ as robustness metric). *Assume $\Delta_\beta > 0$ and that the vectors $v_1, \ldots, v_{n+m}$ are uniformly bounded by an absolute constant. Then, for any $\beta > 0$, the attention output varies continuously with $\Delta_\beta$. Precisely, for any $\varepsilon > 0$, there exists $\delta > 0$ such that $\Delta_\beta < \delta$ implies $\|o_{n+m} - o_n\| < \varepsilon$. Moreover, there exist constants $\delta_\beta, C_\beta > 0$ depending only on $n, m, \beta$ such that if $\Delta_\beta < \delta_\beta$, then*

(i) *If $\beta > 1$,*

$$\|o_{n+m} - o_n\| \leq C_\beta \Delta_\beta. \tag{20}$$

(ii) *If $\beta = 1$,*

$$\|o_{n+m} - o_n\| \leq \frac{C_\beta \Delta_\beta}{\ln(1/\Delta_\beta)}. \tag{21}$$

(iii) *If $0 < \beta < 1$,*

$$\|o_{n+m} - o_n\| \leq C_\beta \Delta_\beta^{\frac{1}{\beta}}. \tag{22}$$

**Proof.** We defer the proof to Section F.2. □

Although the assumption $\Delta_\beta > 0$ may seem abstract, it is naturally satisfied in typical retrieval tasks where a small number of signal logits dominate. In this setting, $\mathcal{E}_{\beta,n}(= O(1)) \leq \tilde{\mathcal{E}}_{\beta,m}(= \Theta(m))$, where $\tilde{\mathcal{E}}_{\beta,m}$ denotes the effective sequence length of the additional noise logits $(s_{n+1}, \ldots, s_{n+m})$. We formalize this in Corollary 1.

**Corollary 1** (Robustness of retrieval heads with dominant signals). *Consider a retrieval task in which the signal is sufficiently dominant such that $\mathcal{E}_{\beta,n} \leq \tilde{\mathcal{E}}_{\beta,m}$. Then $\Delta_\beta > 0$. Consequently, the output difference $\|o_{n+m} - o_n\|$ is bounded in terms of $\Delta_\beta$ as in Theorem 2.*

**Proof.** We defer the proof to Section F.2. □

### E.3. Global Aggregation Heads

An aggregation head is an attention head that extracts global statistics or aggregates information across the full context. It must therefore attend broadly to all tokens and adapt to changes in context length.

**Problem Setup.** Consider a value-aggregation task on a sequence of length $n$. The aggregation rule depends on the positions of the values and is specified by a strictly positive probability density $p : [0, 1] \to (0, \infty)$ (e.g., $p \equiv 1$ corresponds to uniform averaging). The target attention output is the scale-invariant weighted average of the value vectors $v_1, \ldots, v_n$:

$$o_n^* = \sum_{i=1}^n v_i \hat{\pi}_{n,i} \approx \int_0^1 v_{\lceil nx \rceil} p(x) \, dx, \tag{23}$$

where $\hat{\pi}_{n,i} = p(i/n) / \sum_{j=1}^n p(j/n)$ are the weights of the discrete distribution induced by $p$. For each $n$, our goal is to approximate $o_n^*$ using the single-layer attention output at the last token, $o_n$. Equivalently, the attention weights should approximate the target weight vector $(\hat{\pi}_{n,1}, \ldots, \hat{\pi}_{n,n})$.

The following Theorem 3 shows that, to approximate global aggregation tasks within an $\varepsilon$ tolerance, $\mathcal{E}_\beta(\alpha)$ must grow linearly with the sequence length.

**Theorem 3** ($\mathcal{E}_\beta$ scales in aggregation tasks). *Suppose the target density function $p : [0, 1] \to (0, \infty)$ is Riemann integrable. Let $\pi = (\pi_1, \ldots, \pi_n)$ be the target distribution induced by $p$, and let $\alpha = (\alpha_1, \ldots, \alpha_n)$ be the attention weight distribution for sequence length $n$. For any $\beta > 0$ and $\varepsilon > 0$, if $D_\beta(\alpha \| \pi) \leq \varepsilon$, then $\mathcal{E}_\beta(\alpha) = \Theta(n)$. Moreover, there exists a constant $C > 0$ depending only on $p$ such that*

$$Ce^{-\varepsilon} n \leq \mathcal{E}_\beta(\alpha) \leq n. \tag{24}$$

**Proof.** We defer the proof to Section F.3. □

### E.4. Theoretical Foundations of AdaScale

Motivated by Theorems 2 and 3, we assume that each attention head performs a specific task (e.g., retrieval or aggregation) and is characterized by a head-specific $\beta > 0$. The task determines the desired behavior of $\mathcal{E}_\beta(\boldsymbol{\alpha})$ as the context length $L$ increases: $\mathcal{E}_\beta(\boldsymbol{\alpha})$ should remain approximately constant for a retrieval head (Theorem 2) and grow as $\Theta(L)$ for a global aggregation head (Theorem 3). By Eq. (18), different scaling factors $\lambda$ can induce different asymptotic behaviors of $\mathcal{E}_\beta(\boldsymbol{\alpha})$. Consider $\beta = 2$ as an example. Suppose that $\boldsymbol{s} = (s_1, \ldots, s_L)$ is a one-hot vector. Then

$$\mathcal{E}_2(\boldsymbol{\alpha}) = \frac{\left(e^\lambda + L - 1\right)^2}{e^{2\lambda} + L - 1} = \frac{1 + 2(L-1)e^{-\lambda} + (L-1)^2 e^{-2\lambda}}{1 + (L-1)e^{-2\lambda}} \leq 1 + 2Le^{-\lambda} + L^2 e^{-2\lambda}.$$

If $e^\lambda = \Omega(L)$, then $\mathcal{E}_2(\boldsymbol{\alpha}) = O(1)$; in contrast, if $e^\lambda = O(1)$, then $\mathcal{E}_2(\boldsymbol{\alpha}) = \Theta(L)$. In this subsection, we consider a more general setting in which $\boldsymbol{s}$ is a random vector. Under this setting, we identify scaling factors $\lambda(L)$ for which $\mathcal{E}_\beta(\boldsymbol{\alpha})$ exhibits the prescribed asymptotic behavior (Theorems 4 and 5). This analysis leads to the per-head AdaScale inverse temperature used in Section 3.3.

**Problem Setup.** We consider a simplified model in which query and key vectors are correlated Gaussian distributed. Fix a context length $L$. Let $\boldsymbol{q} = (q_1, \ldots, q_d)$ denote the query vector at position $L$, and assume $\boldsymbol{q} \sim \mathcal{N}(\boldsymbol{0}_d, \boldsymbol{I}_d)$. For each $i \in \{1, \ldots, L\}$, the key vectors $\boldsymbol{k}_i = (k_{i,1}, \ldots, k_{i,d})$ are conditionally i.i.d. given $\boldsymbol{q}$, with $\boldsymbol{k}_i = \frac{1}{\sqrt{d}}\rho\boldsymbol{q} + \sigma\boldsymbol{z}_i$, where $\rho, \sigma > 0$ are constants and $\boldsymbol{z}_i \overset{\text{iid}}{\sim} \mathcal{N}(\boldsymbol{0}_d, \boldsymbol{I}_d)$.

In the following analysis, we investigate the relationship between the scaling factor $\lambda(L)$ and the effective sequence length $\mathcal{E}_\beta(\boldsymbol{\alpha})$. However, by Eq. (18), $\mathcal{E}_\beta(\boldsymbol{\alpha})$ is a random variable (since the query and key vectors are random) and it would be more convenient to study its limiting behavior (as $d$ and $L$ approach infinity).

Note that our correlated Gaussian query–key setup corresponds to the Gaussian logits assumption in (Anson et al., 2025) as $d \to \infty$, and generalizes the independent query–key model that yields zero-mean logits (Barbero et al., 2025), thereby capturing effects beyond the initialization regime. Under this model, we first show in Proposition 1 that the attention logits jointly converge to a multivariate normal distribution with diagonal covariance. In the NoPE case the logits are asymptotically i.i.d.; in the RoPE case the limiting mean depends on the token position.

**Proposition 1** (Infinite-width limit of logits). *Let $\theta_f$, $f \in \{0, \ldots, d/2 - 1\}$ denote the rotation frequencies. Then*

$$(s_1, \ldots, s_L) \overset{\mathcal{D}}{\longrightarrow} \mathcal{N}\left((\mu_1, \ldots, \mu_L), \sigma^2 \boldsymbol{I}_L\right), \quad \text{as } d \to \infty.$$

*where*

$$\mu_i = \rho \lim_{d \to \infty} \frac{2}{d} \sum_{f=0}^{d/2-1} \cos\left((i - L)\theta_f\right), \quad \forall i \in \{1, \ldots, L\}. \tag{25}$$

*Moreover, in the NoPE case, where $\theta_f = 0$ for all $f \in \{0, \ldots, d/2 - 1\}$,*

$$\mu_i = \rho, \quad \forall i \in \{1, \ldots, L\}. \tag{26}$$

*In the RoPE case, where $\theta_f = b^{-2f/d}$ for $f \in \{0, \ldots, d/2 - 1\}$,*

$$\mu_i = \rho \int_0^1 \cos\left((i - L)b^{-x}\right) dx, \quad \forall i \in \{1, \ldots, L\}. \tag{27}$$

**Proof.** We defer the proof to Section F.4. □

On the other hand, assuming the logits are independently Gaussian-distributed, Proposition 2 characterizes the conditions under which the law of large numbers holds as the sequence length $L \to \infty$. The effect of the rotation frequencies appears in the limiting behavior of $\mathcal{E}_\beta(\boldsymbol{\alpha})$. For convenience, we introduce the following notation. Let $\mathring{\boldsymbol{\alpha}} = (\mathring{\alpha}_1, \ldots, \mathring{\alpha}_L)$ denote the limit of $\boldsymbol{\alpha}$ as $\sigma \to 0$, where

$$\mathring{\alpha}_i = \frac{e^{\lambda\mu_i}}{\sum_{j=1}^L e^{\lambda\mu_j}} = \frac{e^{\lambda\mu_i}}{Z(\boldsymbol{\mu}; \lambda)}. \tag{28}$$

Then, from Eq. (18), we have

$$
\mathcal{E}_\beta(\mathring{\boldsymbol{\alpha}}) = \left( \sum_{i=1}^{L} \left( \frac{e^{\lambda \mu_i}}{\sum_{j=1}^{L} e^{\lambda \mu_j}} \right)^\beta \right)^{\frac{1}{1-\beta}} = \frac{\left( \sum_{i=1}^{L} e^{\lambda \mu_i} \right)^{\frac{\beta}{\beta-1}}}{\left( \sum_{i=1}^{L} e^{\beta \lambda \mu_i} \right)^{\frac{1}{\beta-1}}} = \frac{Z(\boldsymbol{\mu}; \lambda)^{\frac{\beta}{\beta-1}}}{Z(\boldsymbol{\mu}; \beta \lambda)^{\frac{1}{\beta-1}}}.
\tag{29}
$$

Accordingly, we define a law-of-large-numbers proxy for the effective sequence length $\mathcal{E}_\beta(\boldsymbol{\alpha})$ by

$$
\hat{\mathcal{E}}_\beta(L) := \frac{(\mathbb{E}\, Z(\boldsymbol{s}; \lambda))^{\frac{\beta}{\beta-1}}}{(\mathbb{E}\, Z(\boldsymbol{s}; \beta \lambda))^{\frac{1}{\beta-1}}} = \frac{\left( \sum_{i=1}^{L} \mathbb{E}\, e^{\lambda \mu_i} \right)^{\frac{\beta}{\beta-1}}}{\left( \sum_{i=1}^{L} \mathbb{E}\, e^{\beta \lambda \mu_i} \right)^{\frac{1}{\beta-1}}} = \frac{\left( \sum_{i=1}^{L} e^{\lambda \mu_i + \frac{1}{2} \lambda^2 \sigma^2} \right)^{\frac{\beta}{\beta-1}}}{\left( \sum_{i=1}^{L} e^{\beta \lambda \mu_i + \frac{1}{2} \beta^2 \lambda^2 \sigma^2} \right)^{\frac{1}{\beta-1}}}
$$

$$
= e^{-\frac{\beta}{2} \lambda^2 \sigma^2} \frac{Z(\boldsymbol{\mu}; \lambda)^{\frac{\beta}{\beta-1}}}{Z(\boldsymbol{\mu}; \beta \lambda)^{\frac{1}{\beta-1}}} = e^{-\frac{\beta}{2} \lambda^2 \sigma^2} \mathcal{E}_\beta(\mathring{\boldsymbol{\alpha}}),
\tag{30}
$$

**Proposition 2** (Infinite-length limit of $\mathcal{E}_\beta$). *Let the logits $s_1, \ldots, s_L$ be independent Gaussian random variables with common variance $\sigma^2$, i.e., $s_i \sim \mathcal{N}(\mu_i, \sigma^2)$. Write $\boldsymbol{\mu} := (\mu_1, \ldots, \mu_L)$ and assume $\|\boldsymbol{\mu}\|_\infty := \max_{1 \le i \le L} |\mu_i| < \infty$. Let the inverse temperature $\lambda = \lambda(L)$ be a deterministic function of $L$. Define the scaling parameter*

$$
\Lambda := \limsup_{L \to \infty} \frac{\lambda(L)\sigma}{\sqrt{\ln L}}.
\tag{31}
$$

*Then the following hold:*

(i) *If $\Lambda < \sqrt{2}\min\{1/\beta, 1\}$, then*

$$
\frac{\mathcal{E}_\beta(\boldsymbol{\alpha})}{\hat{\mathcal{E}}_\beta(L)} \xrightarrow{\mathbb{P}} 1 \quad \text{as } L \to \infty.
\tag{32}
$$

(ii) *If $\Lambda = \sqrt{2}\min\{1/\beta, 1\}$ and $\mathcal{E}_2(\mathring{\boldsymbol{\alpha}}) = \omega(L/\sqrt{\ln L})$, then*

$$
\frac{\mathcal{E}_\beta(\boldsymbol{\alpha})}{\hat{\mathcal{E}}_\beta(L)} \xrightarrow{\mathbb{P}} 2^{\frac{1}{\max\{\beta, \beta^{-1}\}-1}} \quad \text{as } L \to \infty.
\tag{33}
$$

(iii) *If $\Lambda > \sqrt{2}/\beta$, then*

$$
\liminf_{L \to \infty} \hat{\mathcal{E}}_\beta(L) = 0.
\tag{34}
$$

*Moreover, if $\Lambda = \infty$, then*

$$
\mathcal{E}_\beta(\boldsymbol{\alpha}) \xrightarrow{\mathbb{P}} 1 \quad \text{as } L \to \infty.
\tag{35}
$$

**Proof.** We defer the proof to Section F.5. $\qquad\square$

Define $\mathcal{E}_\beta^*(L)$ as the asymptotic limit of $\mathcal{E}_\beta(\boldsymbol{\alpha})$ in Proposition 2 under the law of large numbers, so that $\mathcal{E}_\beta(\boldsymbol{\alpha})/\mathcal{E}_\beta^*(L) \xrightarrow{\mathbb{P}} 1$ as $L \to \infty$.[2] Moreover, if the asymptotic forms of $Z(\boldsymbol{\mu}; \lambda)$ and $Z(\boldsymbol{\mu}; \beta \lambda)$ are known, the scaling factor $\lambda(L)$ can be recovered from Proposition 2. We apply this approach to NoPE and RoPE in Theorems 4 and 5, respectively.

**Theorem 4** (Scaling factor of NoPE). *Suppose $\theta_f = 0$ for all $f \in \{0, \ldots, d/2 - 1\}$, and let $\boldsymbol{\alpha} = (\alpha_1, \ldots, \alpha_L)$ denote the attention weights. Let $\mathcal{E}_\beta^*(L)$ denote the head-specific asymptotic limit of $\mathcal{E}_\beta(\boldsymbol{\alpha})$ under the law of large numbers. If one of the following conditions holds*

---

[2]If the law of large numbers does not hold, the limit of $\mathcal{E}_\beta(\boldsymbol{\alpha})$ may remain random and therefore cannot be used in the AdaScale criterion, which requires a deterministic limit of $\mathcal{E}_\beta$.

(i) $\liminf\limits_{L\to\infty} \dfrac{\ln \mathcal{E}_\beta^*(L)}{\ln L} > 1 - \min\left\{\beta, \beta^{-1}\right\}$ *and*

$$\lambda(L) = \sqrt{\frac{2}{\beta\sigma^2} \ln\left(\frac{L}{\mathcal{E}_\beta^*(L)}\right)};$$ (36)

(ii) $\lim\limits_{L\to\infty} \dfrac{\ln \mathcal{E}_\beta^*(L)}{\ln L} = 1 - \min\left\{\beta, \beta^{-1}\right\}$ *and*

$$\lambda(L) = \sqrt{\frac{2}{\beta\sigma^2} \ln\left(\frac{2^{\frac{1}{\max\{\beta,\beta^{-1}\}-1}} L}{\mathcal{E}_\beta^*(L)}\right)};$$ (37)

(iii) $\lim\limits_{L\to\infty} \mathcal{E}_\beta^*(L) = 1$ *and*

$$\lambda(L) = \omega(\sqrt{\ln L}) \quad \text{as } L \to \infty;$$ (38)

*then*

$$\frac{\mathcal{E}_\beta(\boldsymbol{\alpha})}{\mathcal{E}_\beta^*(L)} \xrightarrow{\mathbb{P}} 1 \quad \text{as } d, L \to \infty.$$

**Proof.** We defer the proof to Section F.6. $\qquad\square$

**Remark 4.** By Eq. (30), the square-root dependence in Eqs. (36)–(38) arises from the logarithm of the moment generating function (MGF) of a Gaussian distribution. Thus, the derivation can be extended to other assumptions on the logit distribution, such as Laplace or Lévy $\alpha$-stable distributions with $\alpha \in [1, 2)$, which would yield different forms of $\lambda(L)$.

**Theorem 5** (Scaling factor of RoPE). *Suppose* $\theta_f = b^{-2f/d}$ *for all* $f \in \{0, \ldots, d/2 - 1\}$, *and let* $\boldsymbol{\alpha} = (\alpha_1, \ldots, \alpha_L)$ *denote the attention weights. Let* $\mathcal{E}_\beta^*(L)$ *denote the head-specific asymptotic limit of* $\mathcal{E}_\beta(\boldsymbol{\alpha})$ *under the law of large numbers. If* $\lambda(L) \to \infty$ *and* $\lambda(L) \ln L = o(e^{\rho \min\{1,\beta\}\lambda(L)})$ *as* $L \to \infty$, *and one of the following conditions holds*

(i) $\liminf\limits_{L\to\infty} \dfrac{\ln \mathcal{E}_\beta^*(L)}{\ln L} > 1 - \min\left\{\beta, \beta^{-1}\right\}$ *and*

$$\lambda(L) = \sqrt{\frac{2}{\beta\sigma^2} \ln\left(\frac{1}{\mathcal{E}_\beta^*(L)}\left(L + O\left(e^{\rho\max\{1,\beta\}\lambda(L)}\right)\right)\right)} \quad \text{as } L \to \infty;$$ (39)

(ii) $\lim\limits_{L\to\infty} \dfrac{\ln \mathcal{E}_\beta^*(L)}{\ln L} = 1 - \min\left\{\beta, \beta^{-1}\right\}$ *and*

$$\lambda(L) = \sqrt{\frac{2}{\beta\sigma^2} \ln\left(\frac{2^{\frac{1}{\max\{\beta,\beta^{-1}\}-1}}}{\mathcal{E}_\beta^*(L)}\left(L + O\left(e^{\rho\max\{1,\beta\}\lambda(L)}\right)\right)\right)} \quad \text{as } L \to \infty;$$ (40)

(iii) $\lim\limits_{L\to\infty} \mathcal{E}_\beta^*(L) = 1$ *and*

$$\lambda(L) = \omega(\sqrt{\ln L}) \quad \text{as } L \to \infty;$$ (41)

*then*

$$\frac{\mathcal{E}_\beta(\boldsymbol{\alpha})}{\mathcal{E}_\beta^*(L)} \xrightarrow{\mathbb{P}} 1 \quad \text{as } d, L \to \infty.$$

**Proof.** We defer the proof to Section F.7. $\qquad\square$

**Remark 5.** Unlike the NoPE result in Theorem 4, we do not derive an explicit expression for $\lambda(L)$ under RoPE. Instead, we obtain an asymptotic relation analogous to the NoPE case, in which $\lambda(L)$ appears on both sides. For suitable choices of $\lambda(L)$, this relation can be viewed as a small correction to the NoPE expression; for example, if $\lambda(L)$ is chosen such that $e^{\rho \max\{1,\beta\}\lambda(L)} = o(L)$, then the correction term is negligible.

**Remark 6** (Implication for AdaScale). Because the scaling factor $\lambda(L)$ depends on the head-specific $\mathcal{E}_\beta^*(L)$, a simple way to adapt each head to its prescribed task is to introduce a learnable scale. In AdaScale, for each head $h$ we parametrize the inverse temperature as

$$\lambda^{(h)}(L) = \frac{1}{\tau^{(h)}} \left[ \ln \left( 1 + \frac{\max\{L, L_{\text{ref}}\}}{L_{\text{ref}}} \right) \right]^{\gamma^{(h)}},$$

where $L_{\text{ref}}$ is a hyperparameter, and $\tau^{(h)}$ and $\gamma^{(h)}$ are trainable parameters. Note that:

- the additive 1 inside the logarithm ensures a positive argument and avoids numerical instability;

- when $L \leq L_{\text{ref}}$, $\lambda^{(h)}(L)$ is constant, which prevents instability on short sequences;

- when $L > L_{\text{ref}}$, $\lambda^{(h)}(L)$ varies with $L$, allowing different heads to specialize according to their tasks. Since the head-specific $\beta$ and $\hat{\mathcal{E}}_\beta(L)$ forms are unknown a priori, we absorb those degrees of freedom into the factor $\tau^{(h)}$ and the exponent $\gamma^{(h)}$.

# F. Proofs

## F.1. Proof of Theorem 1

**Theorem 1** (Necessity of head-wise rotation frequencies). *Suppose $0 < \theta_f \leq \pi$ for all $f \in \{0, \ldots, d/2 - 1\}$, and $\|\boldsymbol{q}\|, \|\boldsymbol{k}_+\| > 0$. For any $\varepsilon > 0$ with $\varepsilon < (\exp(\|\boldsymbol{q}\| \cdot \|\boldsymbol{k}_-\|) + 1)^{-1}$, every feasible solution of Eq. (9) satisfies*

$$\frac{C_1}{rW/2 + 1} \leq \mathcal{M} \leq \frac{\pi}{W + 1} + C_2 \cdot \mathbb{1}\left\{ \max_{0 \leq f \leq d/2 - 1} \theta_f > \frac{\pi}{W + 1} \right\}, \tag{12}$$

*where $C_1, C_2 > 0$ are constants that depend only on $\varepsilon$, $L$, $\boldsymbol{q}$, $\boldsymbol{k}_+$ and $\boldsymbol{k}_-$, and are independent of the window center $c$, width $W$, and the buffer ratio $r$.*

**Proof.** By the Cauchy–Schwarz inequality,

$$-\|\boldsymbol{q}\| \cdot \|\boldsymbol{k}_-\| \leq \boldsymbol{q}^\mathsf{T} \boldsymbol{R}^{i-L} \boldsymbol{k}_- \leq \|\boldsymbol{q}\| \cdot \|\boldsymbol{k}_-\|.$$

Hence for any $t \in [1, L]$,

$$\ln(L - 1) - \|\boldsymbol{q}\| \cdot \|\boldsymbol{k}_-\| \leq s_{-t} \leq \ln(L - 1) + \|\boldsymbol{q}\| \cdot \|\boldsymbol{k}_-\|.$$

Therefore,

$$T_L(t) \geq T_L^* := \frac{1}{A_\star} \left( \ln(L - 1) - \|\boldsymbol{q}\| \cdot \|\boldsymbol{k}_-\| + \ln \left( \frac{1 - \varepsilon}{\varepsilon} \right) \right),$$

$$T_U(t) \leq T_U^* := \frac{1}{A_\star} \left( \ln(L - 1) + \|\boldsymbol{q}\| \cdot \|\boldsymbol{k}_-\| + \ln \left( \frac{\varepsilon}{1 - \varepsilon} \right) \right). \tag{42}$$

**1. Lower bound of $\mathcal{M}$.** Note that

$$|T'(x)| = \left| \sum_f (-a_f \theta_f) \sin(\theta_f x + \varphi_f) \right| \leq \sum_f a_f \theta_f = \mathcal{M},$$

so $T$ is $\mathcal{M}$-Lipschitz. At the window boundary, the Lipschitz property together with Eqs. (11) and (42) implies

$$\left\lceil \frac{r}{2}W \right\rceil \mathcal{M} \geq T\left(c - \frac{W}{2}\right) - T\left(c - \left\lceil (1+r)\frac{W}{2} \right\rceil\right)$$

$$\geq T_L\left(c - \frac{W}{2}\right) - T_U\left(c - \left\lceil (1+r)\frac{W}{2} \right\rceil\right)$$

$$\geq T_L^* - T_U^*$$

$$= \frac{2}{A_\star}\left(\ln\left(\frac{1-\varepsilon}{\varepsilon}\right) - \|\boldsymbol{q}\| \cdot \|\boldsymbol{k}_-\|\right).$$

The assumption $\varepsilon < (\exp(\|\boldsymbol{q}\| \cdot \|\boldsymbol{k}_-\|) + 1)^{-1}$ guarantees that the above lower bound is positive. Therefore,

$$\mathcal{M} \geq \frac{1}{rW/2 + 1} \cdot \frac{2}{A_\star}\left(\ln\left(\frac{1-\varepsilon}{\varepsilon}\right) - \|\boldsymbol{q}\| \cdot \|\boldsymbol{k}_-\|\right). \tag{43}$$

**2. Upper bound of $\mathcal{M}$.** Define the average of $T(x)$ over $[c - W/2, c + W/2]$ as

$$(\mathcal{A}_W T)(c) = \frac{1}{W+1} \sum_{t=c-W/2}^{c+W/2} T(t).$$

Then Eq. (11) implies

$$(\mathcal{A}_W T_L)(c) \leq (\mathcal{A}_W T)(c). \tag{44}$$

By Lagrange's trigonometric identity,

$$(\mathcal{A}_W T)(c) = \frac{1}{W+1} \sum_{t=c-W/2}^{c+W/2} \sum_{f=0}^{d/2-1} a_f \cos(\theta_f t + \varphi_f)$$

$$= \frac{1}{W+1} \sum_{f=0}^{d/2-1} a_f \sum_{k=-W/2}^{W/2} \cos(\theta_f(c+k) + \varphi_f)$$

$$= \frac{1}{W+1} \sum_{f=0}^{d/2-1} a_f \sum_{k=-W/2}^{W/2} \left[\cos(\theta_f k)\cos(\theta_f c + \varphi_f) - \cancel{\sin(\theta_f k)\sin(\theta_f c + \varphi_f)}\right]$$

$$= \sum_{f=0}^{d/2-1} a_f \cos(\theta_f c + \varphi_f)\left[\frac{1}{W+1}\sum_{k=-W/2}^{W/2}\cos(\theta_f k)\right]$$

$$= \sum_{f=0}^{d/2-1} a_f \cos(\theta_f c + \varphi_f)\mathcal{D}_W(\theta_f),$$

where

$$\mathcal{D}_W(\theta_f) := \frac{1}{W+1}\sum_{k=-W/2}^{W/2}\cos(\theta_f k) = \frac{\sin\left((W+1)\theta_f/2\right)}{(W+1)\sin\left(\theta_f/2\right)}.$$

Since $|\mathcal{D}_W(\theta_f)| \leq 1$ and $\sin z \geq \frac{2}{\pi}z$ for $0 \leq z \leq \pi/2$, we obtain

$$|\mathcal{D}_W(\theta_f)| \leq \min\left\{1, \frac{\pi}{(W+1)\theta_f}\right\}, \quad \text{if } 0 < \theta_f \leq \pi.$$

Hence

$$(\mathcal{A}_W T)(c) \leq \sum_{f=0}^{d/2-1} a_f|\mathcal{D}_W(\theta_f)| \leq \sum_{f=0}^{d/2-1} a_f \min\left\{1, \frac{\pi}{(W+1)\theta_f}\right\}.$$

By Eq. (44) this yields $(\mathcal{A}_W T_L)(c) \leq 1$. If $\theta_{\max} := \max_{0 \leq f \leq d/2-1} \theta_f \leq \frac{\pi}{W+1}$, then

$$\mathcal{M} = \sum_{f=0}^{d/2-1} a_f \theta_f \leq \theta_{\max} \leq \frac{\pi}{W+1}. \tag{45}$$

Otherwise, by the concavity of $1/z$, the function $\min\left\{1, \frac{\pi}{(W+1)\theta_f}\right\}$ is bounded above by the line through $\left(\frac{\pi}{W+1}, 1\right)$ and $\left(\theta_{\max}, \frac{\pi}{(W+1)\theta_{\max}}\right)$, so

$$\min\left\{1, \frac{\pi}{(W+1)\theta_f}\right\} \leq 1 + \left(\theta_f - \frac{\pi}{W+1}\right) \frac{\frac{\pi}{(W+1)\theta_{\max}} - 1}{\theta_{\max} - \frac{\pi}{W+1}} = 1 - \frac{1}{\theta_{\max}} \left(\theta_f - \frac{\pi}{W+1}\right).$$

Therefore,

$$(\mathcal{A}_W T)(c) \leq \sum_{f=0}^{d/2-1} a_f \left(1 - \frac{1}{\theta_{\max}} \left(\theta_f - \frac{\pi}{W+1}\right)\right) = 1 + \frac{\pi}{(W+1)\theta_{\max}} - \frac{\mathcal{M}}{\theta_{\max}}.$$

Combining this with Eq. (44), $(\mathcal{A}_W T_L)(c) \leq 1$, and $\theta_{\max} \leq \pi$ gives

$$\mathcal{M} \leq \frac{\pi}{W+1} + (1 - (\mathcal{A}_W T_L)(c)) \theta_{\max} \leq \frac{\pi}{W+1} + (1 - (\mathcal{A}_W T_L)(c)) \pi.$$

Finally, using Eq. (42), which gives $(\mathcal{A}_W T_L)(c) \geq T_L^*$, we obtain

$$\mathcal{M} \leq \frac{\pi}{W+1} + \pi \left[1 - \frac{1}{A_\star} \left(\ln(L-1) + \ln\left(\frac{1-\varepsilon}{\varepsilon}\right) - \|\boldsymbol{q}\| \cdot \|\boldsymbol{k}_-\|\right)\right]. \tag{46}$$

3. Combining Eqs. (43), (45) and (46) yields

$$\begin{aligned}
C_1 &= \frac{2}{A_\star} \left(\ln\left(\frac{1-\varepsilon}{\varepsilon}\right) - \|\boldsymbol{q}\| \cdot \|\boldsymbol{k}_-\|\right), \\
C_2 &= \pi \left[1 - \frac{1}{A_\star} \left(\ln(L-1) + \ln\left(\frac{1-\varepsilon}{\varepsilon}\right) - \|\boldsymbol{q}\| \cdot \|\boldsymbol{k}_-\|\right)\right],
\end{aligned} \tag{47}$$

which completes the proof. $\qquad\square$

### F.2. Proofs of Theorem 2 and Corollary 1

We first introduce some additional notation. Define the unnormalized attention weights as $a_i = \exp(s_i)$ for $i \in \{1, \ldots, n+m\}$. Let

$$Z_{\beta,n} = \sum_{i=1}^{n} a_i^\beta, \qquad \tilde{Z}_{\beta,m} = \sum_{j=1}^{m} a_{n+j}^\beta, \qquad Z_{\beta,n+m} = \sum_{i=1}^{n+m} a_i^\beta.$$

Thus $\alpha_i = \frac{a_i}{Z_{1,n+m}}$. The attention outputs for the signal, noise, and combined sequences are

$$\boldsymbol{o}_n = \sum_{i=1}^{n} \frac{a_i}{Z_{1,n}} \boldsymbol{v}_i, \qquad \tilde{\boldsymbol{o}}_m = \sum_{j=1}^{m} \frac{a_{n+j}}{\tilde{Z}_{1,m}} \boldsymbol{v}_{n+j}, \qquad \boldsymbol{o}_{n+m} = \sum_{i=1}^{n+m} \frac{a_i}{Z_{1,n+m}} \boldsymbol{v}_i.$$

By Eq. (18), the corresponding effective sequence lengths for $\beta \neq 1$ are

$$\mathcal{E}_{\beta,n} = \frac{Z_{1,n}^{\frac{\beta}{\beta-1}}}{Z_{\beta,n}^{\frac{1}{\beta-1}}}, \qquad \tilde{\mathcal{E}}_{\beta,m} = \frac{\tilde{Z}_{1,m}^{\frac{\beta}{\beta-1}}}{\tilde{Z}_{\beta,m}^{\frac{1}{\beta-1}}}, \qquad \mathcal{E}_{\beta,n+m} = \frac{Z_{1,n+m}^{\frac{\beta}{\beta-1}}}{Z_{\beta,n+m}^{\frac{1}{\beta-1}}}.$$

Since $Z_{\beta,n+m} = Z_{\beta,n} + \tilde{Z}_{\beta,m}$ for all $\beta > 0$, we can decompose

$$\boldsymbol{o}_{n+m} = \sum_{i=1}^{n} \frac{a_i}{Z_{1,n} + \tilde{Z}_{1,m}} \boldsymbol{v}_i + \sum_{j=1}^{m} \frac{a_{n+j}}{Z_{1,n} + \tilde{Z}_{1,m}} \boldsymbol{v}_{n+j}$$

$$= \frac{Z_{1,n}}{Z_{1,n} + \tilde{Z}_{1,m}} \sum_{i=1}^{n} \frac{a_i}{Z_{1,n}} \boldsymbol{v}_i + \frac{\tilde{Z}_{1,m}}{Z_{1,n} + \tilde{Z}_{1,m}} \sum_{j=1}^{m} \frac{a_{n+j}}{\tilde{Z}_{1,m}} \boldsymbol{v}_{n+j}$$

Let $\eta := \frac{\tilde{Z}_{1,m}}{Z_{1,n} + \tilde{Z}_{1,m}} = 1 - \frac{Z_{1,n}}{Z_{1,n} + \tilde{Z}_{1,m}} \in (0,1)$. Then

$$\boldsymbol{o}_{n+m} - \boldsymbol{o}_n = \eta \left( \sum_{j=1}^{m} \frac{a_{n+j}}{\tilde{Z}_{1,m}} \boldsymbol{v}_{n+j} - \sum_{i=1}^{n} \frac{a_i}{Z_{1,n}} \boldsymbol{v}_i \right) = \eta(\tilde{\boldsymbol{o}}_m - \boldsymbol{o}_n).$$

Therefore,

$$\|\boldsymbol{o}_{n+m} - \boldsymbol{o}_n\| \le \eta \|\tilde{\boldsymbol{o}}_m - \boldsymbol{o}_n\|.$$

Since $\tilde{\boldsymbol{o}}_m$ and $\boldsymbol{o}_n$ are convex combinations of the corresponding value vectors, we have

$$\|\tilde{\boldsymbol{o}}_m - \boldsymbol{o}_n\| \le \|\tilde{\boldsymbol{o}}_m\| + \|\boldsymbol{o}_n\| \le \max_{1 \le i \le n} \|\boldsymbol{v}_i\| + \max_{1 \le j \le m} \|\boldsymbol{v}_{n+j}\| < \infty.$$

Thus, it remains to bound $\eta$ in terms of $\Delta_\beta$. Moreover,

$$\mathcal{E}_{\beta,n+m}^{1-\beta} = \frac{Z_{\beta,n+m}}{Z_{1,n+m}^{\beta}} = \frac{Z_{\beta,n} + \tilde{Z}_{\beta,m}}{\left( Z_{1,n} + \tilde{Z}_{1,m} \right)^{\beta}}$$

$$= \left( \frac{Z_{1,n}}{Z_{1,n} + \tilde{Z}_{1,m}} \right)^{\beta} \frac{Z_{\beta,n}}{Z_{1,n}^{\beta}} + \left( \frac{\tilde{Z}_{1,m}}{Z_{1,n} + \tilde{Z}_{1,m}} \right)^{\beta} \frac{\tilde{Z}_{\beta,m}}{\tilde{Z}_{1,m}^{\beta}}$$

$$= (1-\eta)^{\beta} \mathcal{E}_{\beta,n}^{1-\beta} + \eta^{\beta} \tilde{\mathcal{E}}_{\beta,m}^{1-\beta},$$

Dividing by $\mathcal{E}_{\beta,n}^{1-\beta}$ gives

$$(1 + \Delta_\beta)^{1-\beta} = (1-\eta)^{\beta} + \eta^{\beta} \left( \frac{\tilde{\mathcal{E}}_{\beta,m}}{\mathcal{E}_{\beta,n}} \right)^{1-\beta} = (1-\eta)^{\beta} + \kappa_\beta \eta^{\beta}, \tag{48}$$

where $\kappa_\beta := \left( \tilde{\mathcal{E}}_{\beta,m} / \mathcal{E}_{\beta,n} \right)^{1-\beta} > 0$ for $\beta \ne 1$, since the generalized ESS satisfies $\mathcal{E}_{\beta,\bullet} \ge 1$. Since $\Delta_\beta > 0$, Bernoulli's inequality yields

$$(1 + \Delta_\beta)^{1-\beta} \ge 1 + (1-\beta)\Delta_\beta, \qquad \text{if } \beta > 1,$$
$$(1 + \Delta_\beta)^{1-\beta} \le 1 + (1-\beta)\Delta_\beta, \qquad \text{if } 0 < \beta < 1.$$

For the right-hand side, since $\eta \in (0,1)$, Taylor's theorem gives

$$(1-\eta)^{\beta} = 1 - \beta\eta + \frac{\beta(\beta-1)}{2}(1-\xi)^{\beta-2}\eta^2,$$

for some $\xi \in (0,\eta) \subset (0,1)$. Thus, for the three ranges of $\beta$, we obtain

$$(1-\eta)^{\beta} \le 1 - \beta\eta + \frac{\beta(\beta-1)}{2}\eta^2, \qquad \text{if } \beta \ge 2,$$
$$(1-\eta)^{\beta} \le 1 - \eta, \qquad \text{if } 1 < \beta < 2,$$
$$(1-\eta)^{\beta} \ge 1 - \eta, \qquad \text{if } 0 < \beta < 1.$$

Substituting these bounds into Eq. (48) yields

$$\eta \le \frac{\beta - 1}{\beta}\Delta_\beta + \left(\frac{\beta - 1}{2} + \frac{\kappa_\beta}{\beta}\right)\eta^2, \qquad \text{if} \quad \beta \ge 2,$$

$$\eta \le (\beta - 1)\Delta_\beta + \kappa_\beta \eta^\beta, \qquad \text{if} \quad 1 < \beta < 2,$$

$$\eta^\beta \le \frac{1 - \beta}{\kappa_\beta}\Delta_\beta + \frac{1}{\kappa_\beta}\eta, \qquad \text{if} \quad 0 < \beta < 1.$$

The following Lemma 1 unifies these three cases and provides the basis for the proof of Theorem 2.

**Lemma 1.** *For constants $\gamma > 1$ and $A, B > 0$, suppose $x \in [0,1]$ satisfies $x \le A + Bx^\gamma$. If $A < (\gamma B2^\gamma)^{\frac{1}{1-\gamma}}$, then $0 \le x \le 2A$.*

**Proof.** Let $\Psi(x,t) = x - Bx^\gamma - tA$, so $\partial_x\Psi(x,t) = 1 - \gamma Bx^{\gamma-1}$. Since $\Psi(0,0) = 0$ and $\partial_x\Psi(0,0) = 1 \ne 0$, the implicit function theorem yields a unique $C^1$ function $x(t)$ defined in a neighborhood of 0 with $\Psi(x(t),t) = 0$ and $x(0) = 0$. Define

$$I = \{\tau \in [0,1] : \forall t \in [0,\tau], \ x(t) \text{ exists, is continuous, and } x(t) \le 2tA\}.$$

Clearly $0 \in I$, so $I \ne \varnothing$. We show $I = [0,1]$ by the continuity method.

**1. Monotonicity of $x(t)$.** For any $\tau \in I$ and $t \in [0,\tau]$, differentiate $\Psi(x(t),t) = 0$ to obtain

$$x'(t)\left(1 - \gamma Bx(t)^{\gamma-1}\right) = A.$$

Since $x(t) \le 2tA \le 2A$ and by the hypothesis $0 < A < (\gamma B2^\gamma)^{\frac{1}{1-\gamma}}$ we have

$$1 - \gamma Bx(t)^{\gamma-1} \ge 1 - \gamma B(2A)^{\gamma-1} > 1 - \gamma B(2\gamma B)^{-1} = \frac{1}{2} > 0,$$

hence $x'(t) > 0$. Thus $x(t)$ is strictly increasing on $[0,\tau]$.

**2. Closedness of $I$.** Let $\tau_n \in I$ with $\tau_n \uparrow \tau^*$. Then $x(\tau_n)$ is increasing and bounded by $2A$, so by the monotone convergence theorem the limit $x^* := \lim_{n\to\infty} x(\tau_n)$ exists. Clearly $x^* \le 2\tau^*A$. Hence

$$\partial_x\Psi(x^*, \tau^*) = 1 - \gamma B(x^*)^{\gamma-1} \ge 1 - \gamma B(2A)^{\gamma-1} > 0.$$

By the implicit function theorem the solution extends uniquely to $x(\tau^*) := x^*$. Therefore the property holds at $\tau^*$, and $I$ is closed.

**3. Openness of $I$.** For any $t \in I$, substituting $x(t) \le 2tA$ into $x \le A + Bx^\gamma$ gives

$$\begin{aligned}
x(t) &= tA + B(x(t))^\gamma \\
&\le tA + B(2tA)^\gamma \\
&= tA\left(1 + B2^\gamma t^{\gamma-1}A^{\gamma-1}\right) \\
&\le tA\left(1 + \gamma B2^\gamma A^{\gamma-1}\right) \\
&< 2tA,
\end{aligned}$$

where the second inequality uses $\gamma > 1$ and $t \in [0,1]$, and the last inequality follows from $0 < A < (\gamma B2^\gamma)^{\frac{1}{1-\gamma}}$. By the continuity of $x(t)$, $t$ is an interior point of $I$, so $I$ is open.

**4.** Since $I$ is nonempty, open and closed in the connected set $[0,1]$, we have $I = [0,1]$. In particular at $t = 1$ there is a solution $x(1)$ with $x(1) \le 2A$, completing the proof. $\qquad\square$

**Theorem 2** ($\mathcal{E}_\beta$ as robustness metric). *Assume $\Delta_\beta > 0$ and that the vectors $v_1, \ldots, v_{n+m}$ are uniformly bounded by an absolute constant. Then, for any $\beta > 0$, the attention output varies continuously with $\Delta_\beta$. Precisely, for any $\varepsilon > 0$, there exists $\delta > 0$ such that $\Delta_\beta < \delta$ implies $\|o_{n+m} - o_n\| < \varepsilon$. Moreover, there exist constants $\delta_\beta, C_\beta > 0$ depending only on $n, m, \beta$ such that if $\Delta_\beta < \delta_\beta$, then*

(i) *If $\beta > 1$,*

$$\|\boldsymbol{o}_{n+m} - \boldsymbol{o}_n\| \leq C_\beta \Delta_\beta. \tag{20}$$

(ii) *If $\beta = 1$,*

$$\|\boldsymbol{o}_{n+m} - \boldsymbol{o}_n\| \leq \frac{C_\beta \Delta_\beta}{\ln(1/\Delta_\beta)}. \tag{21}$$

(iii) *If $0 < \beta < 1$,*

$$\|\boldsymbol{o}_{n+m} - \boldsymbol{o}_n\| \leq C_\beta \Delta_\beta^{\frac{1}{\beta}}. \tag{22}$$

**Proof.** Let $B := \max_{1 \leq i \leq n+m} \|\boldsymbol{v}_i\| < \infty$. Then the change in the output is bounded by

$$\|\boldsymbol{o}_{n+m} - \boldsymbol{o}_n\| \leq \eta \|\tilde{\boldsymbol{o}}_m - \boldsymbol{o}_n\| \leq 2B\eta. \tag{49}$$

Consider four regimes depending on the range of $\beta$.

1. If $\beta \geq 2$, applying Lemma 1 with $\gamma = 2$ to

$$\eta \leq \frac{\beta - 1}{\beta}\Delta_\beta + \left( \frac{\beta - 1}{2} + \frac{\kappa_\beta}{\beta} \right)\eta^2$$

gives

$$\eta \leq \frac{2(\beta - 1)}{\beta}\Delta_\beta \quad \text{if } \Delta_\beta < \frac{\beta}{\beta - 1}\left( 8\left( \frac{\beta - 1}{2} + \frac{\kappa_\beta}{\beta} \right) \right)^{-1}. \tag{50}$$

Combined with Eq. (49), the change in the output satisfies

$$\|\boldsymbol{o}_{n+m} - \boldsymbol{o}_n\| \lesssim \Delta_\beta.$$

2. If $1 < \beta < 2$, applying Lemma 1 with $\gamma = \beta$ to

$$\eta \leq (\beta - 1)\Delta_\beta + \kappa_\beta \eta^\beta,$$

we obtain

$$\eta \leq 2(\beta - 1)\Delta_\beta \quad \text{if } \Delta_\beta < \frac{1}{\beta - 1}\left( \kappa_\beta \beta 2^\beta \right)^{\frac{1}{1-\beta}}. \tag{51}$$

Combined with Eq. (49), this implies

$$\|\boldsymbol{o}_{n+m} - \boldsymbol{o}_n\| \lesssim \Delta_\beta.$$

3. If $0 < \beta < 1$, applying Lemma 1 with $\gamma = 1/\beta$ to

$$\eta^\beta \leq \frac{1 - \beta}{\kappa_\beta}\Delta_\beta + \frac{1}{\kappa_\beta}\left( \eta^\beta \right)^{\frac{1}{\beta}}$$

yields

$$\eta \leq (2\Delta_\beta)^{\frac{1}{\beta}} \quad \text{if } \Delta_\beta < \frac{\kappa_\beta}{1 - \beta}\left( \frac{2^{\frac{1}{\beta}}}{\kappa_\beta \beta} \right)^{\frac{\beta}{\beta-1}}. \tag{52}$$

Combining this with Eq. (49) gives

$$\|\boldsymbol{o}_{n+m} - \boldsymbol{o}_n\| \lesssim \Delta_\beta^{\frac{1}{\beta}}.$$

4. If $\beta = 1$, note that $\mathcal{E}_1 = \exp(H_1)$, where $H_1$ is the Shannon entropy. Define

$$H_{1,n} = -\sum_{i=1}^{n} \frac{a_i}{Z_{1,n}} \ln \frac{a_i}{Z_{1,n}}, \qquad \tilde{H}_{1,m} = -\sum_{j=1}^{m} \frac{a_{n+j}}{\tilde{Z}_{1,m}} \ln \frac{a_{n+j}}{\tilde{Z}_{1,m}},$$

$$H_{1,n+m} = -\sum_{i=1}^{n+m} \frac{a_i}{Z_{1,n+m}} \ln \frac{a_i}{Z_{1,n+m}}.$$

By direct decomposition,

$$H_{1,n+m} = -\sum_{i=1}^{n} \frac{a_i}{Z_{1,n+m}} \ln \frac{a_i}{Z_{1,n+m}} - \sum_{j=1}^{m} \frac{a_{n+j}}{Z_{1,n+m}} \ln \frac{a_{n+j}}{Z_{1,n+m}}$$

$$= -\sum_{i=1}^{n} \frac{(1-\eta)a_i}{Z_{1,n}} \ln \frac{(1-\eta)a_i}{Z_{1,n}} - \sum_{j=1}^{m} \frac{\eta a_{n+j}}{\tilde{Z}_{1,m}} \ln \frac{\eta a_{n+j}}{\tilde{Z}_{1,m}}$$

$$= -(1-\eta)\sum_{i=1}^{n} \frac{a_i}{Z_{1,n}} \ln \frac{a_i}{Z_{1,n}} - \eta \sum_{j=1}^{m} \frac{a_{n+j}}{\tilde{Z}_{1,m}} \ln \frac{a_{n+j}}{\tilde{Z}_{1,m}} - (1-\eta)\ln(1-\eta) - \eta \ln \eta$$

$$\geq (1-\eta)H_{1,n} + \eta \tilde{H}_{1,m} - \eta \ln \eta,$$

where the last inequality uses $(1-\eta)\ln(1-\eta) \leq 0$. Hence

$$\ln(\mathcal{E}_{1,n+m}) \geq (1-\eta)\ln(\mathcal{E}_{1,n}) + \eta \ln(\tilde{\mathcal{E}}_{1,m}) - \eta \ln \eta.$$

Subtracting $\ln(\mathcal{E}_{1,n})$ gives

$$\ln(1 + \Delta_1) \geq \eta \ln\left(\frac{\tilde{\mathcal{E}}_{1,m}}{\mathcal{E}_{1,n}} \cdot \frac{1}{\eta}\right).$$

Let $\kappa_1 := \tilde{\mathcal{E}}_{1,m}/\mathcal{E}_{1,n} > 0$. In analogy with the proof of Lemma 1, set $A := \ln(1+\Delta_1)$ and define $\Psi(x,t) = x\ln(\kappa_\beta/x) - tA$. By the implicit function theorem there exists a unique continuous solution $x(t)$ in a neighborhood of $0$ with $x(0) = 0$. Define

$$I = \left\{ \tau \in [0,1] : \forall t \in [0,\tau], \; x(t) \text{ exists, is continuous, and } x(t) \leq \frac{tA}{\ln(\kappa_1/tA)} \right\}.$$

Assume $A < \kappa_1/e$. Then for any $t \in I \subseteq [0,1]$,

$$\partial_x \Psi(x(t), t) = \ln \frac{\kappa_1}{x(t)} - 1 \geq \ln\left(\frac{\kappa_1}{tA}\ln\frac{\kappa_1}{tA}\right) - 1 > \ln\left(\frac{e}{t}\ln\frac{e}{t}\right) - 1 \geq 0,$$

and

$$x(t) = \frac{tA}{\ln(\kappa_1/x(t))} \leq \frac{tA}{\ln\left(\frac{\kappa_1}{tA}\ln\frac{\kappa_1}{tA}\right)} < \frac{tA}{\ln\left(\frac{\kappa_1}{tA}\ln\frac{e}{t}\right)} \leq \frac{tA}{\ln(\kappa_1/tA)}.$$

The same connectivity argument as in Lemma 1 implies $I = [0,1]$. Hence,

$$\eta \leq \frac{\ln(1+\Delta_1)}{\ln(\kappa_1/\ln(1+\Delta_1))} \quad \text{if } \Delta_\beta < e^{\kappa_1/e} - 1. \tag{53}$$

Combined with Eq. (49), the change in the output is bounded by

$$\|o_{n+m} - o_n\| \lesssim \frac{\Delta_1}{\ln(1/\Delta_1)}.$$

5. Finally, since $1 \leq \mathcal{E}_{\beta,n} \leq n$ and $1 \leq \tilde{\mathcal{E}}_{\beta,m} \leq m$, we obtain

$$n^{\frac{1}{\beta-1}} \leq \kappa_\beta \leq n^{\frac{1}{\beta-1}}, \qquad \text{if } \beta > 1,$$

$$\frac{1}{n} \leq \kappa_1 \leq m, \qquad \text{if } \beta = 1,$$

$$n^{\frac{1}{\beta-1}} \leq \kappa_\beta \leq n^{\frac{1}{\beta-1}}, \qquad \text{if } 0 < \beta < 1.$$

The proof is completed by substituting these bounds into Eqs. (50)–(53). □

**Corollary 1** (Robustness of retrieval heads with dominant signals). *Consider a retrieval task in which the signal is sufficiently dominant such that $\mathcal{E}_{\beta,n} \leq \tilde{\mathcal{E}}_{\beta,m}$. Then $\Delta_\beta > 0$. Consequently, the output difference $\|o_{n+m} - o_n\|$ is bounded in terms of $\Delta_\beta$ as in Theorem 2.*

**Proof.** For $\beta > 1$, define

$$R_1 := \frac{\tilde{Z}_{1,m}}{Z_{1,n}} = \frac{\sum_{j=1}^{m} e^{s_{n+j}}}{\sum_{i=1}^{n} e^{s_i}}, \qquad R_\beta := \frac{\tilde{Z}_{\beta,m}}{Z_{\beta,n}} = \frac{\sum_{j=1}^{m} e^{\beta s_{n+j}}}{\sum_{i=1}^{n} e^{\beta s_i}}.$$

By $\tilde{\mathcal{E}}_{\beta,m} \geq \mathcal{E}_{\beta,n}$ and Eq. (18), we have

$$1 \leq \left(\frac{\tilde{\mathcal{E}}_{\beta,m}}{\mathcal{E}_{\beta,n}}\right)^{\beta-1} = \left(\frac{\sum_{j=1}^{m} e^{s_{n+j}}}{\sum_{i=1}^{n} e^{s_i}}\right)^{\beta} \Bigg/ \left(\frac{\sum_{j=1}^{m} e^{\beta s_{n+j}}}{\sum_{i=1}^{n} e^{\beta s_i}}\right) = \frac{R_1^{\beta}}{R_\beta}.$$

Thus,

$$(1 + \Delta_\beta)^{\beta-1} = \left(\frac{\mathcal{E}_{\beta,n+m}}{\mathcal{E}_{\beta,n}}\right)^{\beta-1} = \left(\frac{\sum_{i=1}^{n+m} e^{s_i}}{\sum_{i=1}^{n} e^{s_i}}\right)^{\beta} \Bigg/ \left(\frac{\sum_{i=1}^{n+m} e^{\beta s_i}}{\sum_{i=1}^{n} e^{\beta s_i}}\right)$$

$$= \frac{(1 + R_1)^{\beta}}{1 + R_\beta} \geq \frac{(1 + R_1)^{\beta}}{1 + R_1^{\beta}} > \frac{1 + R_1^{\beta}}{1 + R_1^{\beta}} = 1,$$

which implies $\Delta_\beta > 0$. The cases $0 < \beta < 1$ and $\beta = 1$ follow analogously. $\square$

## F.3. Proof of Theorem 3

**Theorem 3** ($\mathcal{E}_\beta$ scales in aggregation tasks). *Suppose the target density function $p : [0,1] \to (0, \infty)$ is Riemann integrable. Let $\boldsymbol{\pi} = (\pi_1, \ldots, \pi_n)$ be the target distribution induced by $p$, and let $\boldsymbol{\alpha} = (\alpha_1, \ldots, \alpha_n)$ be the attention weight distribution for sequence length $n$. For any $\beta > 0$ and $\varepsilon > 0$, if $D_\beta(\boldsymbol{\alpha} \parallel \boldsymbol{\pi}) \leq \varepsilon$, then $\mathcal{E}_\beta(\boldsymbol{\alpha}) = \Theta(n)$. Moreover, there exists a constant $C > 0$ depending only on $p$ such that*

$$C e^{-\varepsilon} n \leq \mathcal{E}_\beta(\boldsymbol{\alpha}) \leq n. \tag{24}$$

**Proof.** Since $p$ is Riemann integrable and $\int_0^1 p(x)\,\mathrm{d}x = 1$, the Riemann sums $R_n := \frac{1}{n}\sum_{i=1}^{n} p(i/n)$ converge to 1 as $n \to \infty$. Hence there exists $N \in \mathbb{N}$ such that for all $n > N$ we have $R_n \geq 1/2$, and therefore

$$R_n \geq \min\{R_1, \ldots, R_N, 1/2\} := K > 0.$$

For every $n$ and $1 \leq i \leq n$,

$$\frac{p(i/n)}{\sum_{j=1}^{n} p(j/n)} = \frac{p(i/n)}{n R_n} \leq \frac{\sup_{x \in [0,1]} p(x)}{n K}.$$

Since a Riemann integrable function on $[0,1]$ is bounded, set $C := \sup_{x \in [0,1]} p(x)/K > 0$. Then the delocalization condition

$$\max_{1 \leq i \leq n} \pi_i \leq \frac{C}{n} \tag{54}$$

holds uniformly in $n$. Because $1 \leq \mathcal{E}_\beta(\boldsymbol{\alpha}) \leq n$, it suffices to prove $\mathcal{E}_\beta(\boldsymbol{\alpha}) = \Omega(n)$. We consider three cases depending on the value of $\beta$.

1. If $\beta > 1$, the condition $D_\beta(\boldsymbol{\alpha} \parallel \boldsymbol{\pi}) \leq \varepsilon$ implies

$$\frac{1}{\beta - 1} \ln\left(\sum_{i=1}^{n} \alpha_i^{\beta} \pi_i^{1-\beta}\right) \leq \varepsilon,$$

and since $\beta - 1 > 0$, exponentiating both sides gives

$$\sum_{i=1}^{n} \alpha_i^{\beta} \pi_i^{1-\beta} \leq e^{\varepsilon(\beta-1)}.$$

By Eq. (54), $\pi_i^{1-\beta} \geq (C/n)^{1-\beta}$ for all $i \in \{1, \ldots, n\}$. Substituting yields

$$C^{1-\beta} n^{\beta-1} \sum_{i=1}^{n} \alpha_i^{\beta} \leq \sum_{i=1}^{n} \alpha_i^{\beta} \pi_i^{1-\beta} \leq e^{\varepsilon(\beta-1)}.$$

Hence,

$$\sum_{i=1}^{n} \alpha_i^{\beta} \le e^{\varepsilon(\beta-1)} C^{\beta-1} n^{1-\beta}.$$

By the definition of $\mathcal{E}_{\beta}$,

$$\mathcal{E}_{\beta}(\boldsymbol{\alpha}) = \left(\sum_{i=1}^{n} \alpha_i^{\beta}\right)^{\frac{1}{1-\beta}} \ge \left(e^{\varepsilon(\beta-1)} C^{\beta-1} n^{1-\beta}\right)^{\frac{1}{1-\beta}} = \frac{n}{C e^{\varepsilon}}.$$

2. If $0 < \beta < 1$, then from $D_{\beta}(\boldsymbol{\alpha} \parallel \boldsymbol{\pi}) \le \varepsilon$ we obtain

$$\sum_{i=1}^{n} \alpha_i^{\beta} \pi_i^{1-\beta} \ge e^{\varepsilon(\beta-1)}.$$

By Eq. (54), $\pi_i^{1-\beta} \le (C/n)^{1-\beta}$ for all $i \in \{1, \dots, n\}$. Substituting this bound gives

$$C^{1-\beta} n^{\beta-1} \sum_{i=1}^{n} \alpha_i^{\beta} \ge \sum_{i=1}^{n} \alpha_i^{\beta} \pi_i^{1-\beta} \ge e^{\varepsilon(\beta-1)}.$$

Thus,

$$\sum_{i=1}^{n} \alpha_i^{\beta} \ge e^{\varepsilon(\beta-1)} C^{\beta-1} n^{1-\beta}.$$

Raising both sides to the power $1/(1-\beta)$ preserves the inequality, so

$$\mathcal{E}_{\beta}(\boldsymbol{\alpha}) \ge \left(e^{\varepsilon(\beta-1)} C^{\beta-1} n^{1-\beta}\right)^{\frac{1}{1-\beta}} = \frac{n}{C e^{\varepsilon}}.$$

3. For $\beta = 1$, the Rényi divergence equals the KL divergence,

$$D_1(\boldsymbol{\alpha} \parallel \boldsymbol{\pi}) = \sum_{i=1}^{n} \alpha_i \ln \frac{\alpha_i}{\pi_i} \le \varepsilon.$$

With the Shannon entropy $H_1(\boldsymbol{\alpha}) = -\sum_i \alpha_i \ln \alpha_i$, expanding the KL divergence gives

$$H_1(\boldsymbol{\alpha}) \ge -\sum_{i=1}^{n} \alpha_i \ln \pi_i - \varepsilon.$$

By Eq. (54), $-\ln \pi_i \ge \ln n - \ln C$. Therefore,

$$H_1(\boldsymbol{\alpha}) \ge \sum_{i=1}^{n} \alpha_i (\ln n - \ln C) - \varepsilon = \ln n - \ln C - \varepsilon.$$

Exponentiating both sides yields

$$\mathcal{E}_1(\boldsymbol{\alpha}) = \exp(H_1(\boldsymbol{\alpha})) \ge \exp(\ln n - \ln C - \varepsilon) = \frac{n}{C e^{\varepsilon}}.$$

$\square$

### F.4. Proof of Proposition 1

**Proposition 1** (Infinite-width limit of logits). *Let $\theta_f$, $f \in \{0, \ldots, d/2 - 1\}$ denote the rotation frequencies. Then*

$$(s_1, \ldots, s_L) \xrightarrow{\mathcal{D}} \mathcal{N}\left((\mu_1, \ldots, \mu_L), \sigma^2 \boldsymbol{I}_L\right), \quad \text{as } d \to \infty.$$

*where*

$$\mu_i = \rho \lim_{d \to \infty} \frac{2}{d} \sum_{f=0}^{d/2-1} \cos\left((i - L)\theta_f\right), \quad \forall i \in \{1, \ldots, L\}. \tag{25}$$

*Moreover, in the NoPE case, where $\theta_f = 0$ for all $f \in \{0, \ldots, d/2 - 1\}$,*

$$\mu_i = \rho, \quad \forall i \in \{1, \ldots, L\}. \tag{26}$$

*In the RoPE case, where $\theta_f = b^{-2f/d}$ for $f \in \{0, \ldots, d/2 - 1\}$,*

$$\mu_i = \rho \int_0^1 \cos\left((i - L)b^{-x}\right) \mathrm{d}x, \quad \forall i \in \{1, \ldots, L\}. \tag{27}$$

**Proof.** Given the rotation matrix $\boldsymbol{R}$, the attention logit between $\boldsymbol{q}$ and $\boldsymbol{k}_i$ for $i \in \{1, \ldots, L\}$ is

$$s_i = \frac{1}{\sqrt{d}} \boldsymbol{q}^\mathsf{T} \boldsymbol{R}_{i-L} \boldsymbol{k}_i = \frac{1}{\sqrt{d}} \boldsymbol{q}^\mathsf{T} \boldsymbol{R}_{i-L} \left(\frac{1}{\sqrt{d}} \rho \boldsymbol{q} + \sigma \boldsymbol{z}_i\right) = \frac{\rho}{d} \boldsymbol{q}^\mathsf{T} \boldsymbol{R}_{i-L} \boldsymbol{q} + \frac{\sigma}{\sqrt{d}} \boldsymbol{q}^\mathsf{T} \boldsymbol{R}_{i-L} \boldsymbol{z}_i.$$

Since $\boldsymbol{q} \sim \mathcal{N}(\boldsymbol{0}_d, \boldsymbol{I}_d)$, define

$$Y_d := \frac{1}{d} \boldsymbol{q}^\mathsf{T} \boldsymbol{R}_{i-L} \boldsymbol{q}, \qquad \mathbb{E} Y_d = \frac{1}{d} \operatorname{Tr}(\boldsymbol{R}_{i-L}) = \frac{2}{d} \sum_{f=0}^{d/2-1} \cos((i - L)\theta_f).$$

Since $\boldsymbol{R}$ is orthogonal,

$$\|\boldsymbol{R}\|_F^2 = \operatorname{Tr}(\boldsymbol{R}^\mathsf{T} \boldsymbol{R}) = \operatorname{Tr}(\boldsymbol{I}_d) = d, \qquad \|\boldsymbol{R}\| = \sqrt{\lambda_{\max}(\boldsymbol{R}^\mathsf{T} \boldsymbol{R})} = \sqrt{\lambda_{\max}(\boldsymbol{I}_d)} = 1.$$

By the Hanson–Wright inequality (see, e.g., Vershynin (2026, Section 6.2)), there exists a constant $c > 0$ such that, for any $\varepsilon \in (0, 1)$,

$$\mathbb{P}\left\{|Y_d - \mathbb{E} Y_d| > \varepsilon\right\} \leq 2 \exp\left(-c \min\left\{\varepsilon^2 d, \varepsilon d\right\}\right) \leq 2 \exp\left(-c\varepsilon^2 d\right).$$

Therefore,

$$\sum_{d=1}^{\infty} \mathbb{P}\left\{|Y_d - \mathbb{E} Y_d| > \varepsilon\right\} \leq \sum_{d=1}^{\infty} 2 \exp\left(-c\varepsilon^2 d\right) = 2 \sum_{d=1}^{\infty} \left(e^{-c\varepsilon^2}\right)^d < \infty.$$

By the Borel–Cantelli lemma, as $d \to \infty$,

$$Y_d \xrightarrow{\text{a.s.}} \lim_{d \to \infty} \mathbb{E} Y_d = \lim_{d \to \infty} \frac{2}{d} \sum_{f=0}^{d/2-1} \cos\left((i - L)\theta_f\right).$$

Let $\boldsymbol{b} = (b_1, \ldots, b_L)^\mathsf{T}$, where $b_i := \frac{\sigma}{\sqrt{d}} \boldsymbol{q}^\mathsf{T} \boldsymbol{R}_{i-L} \boldsymbol{z}_i$. Then

$$\boldsymbol{b} = \frac{\sigma}{\sqrt{d}} \begin{pmatrix} \vdots \\ \boldsymbol{q}^\mathsf{T} \boldsymbol{R}_{i-L} \boldsymbol{z}_i \\ \vdots \end{pmatrix} = \frac{\sigma}{\sqrt{d}} \begin{pmatrix} \vdots \\ \sum_{j=1}^d q_j (\boldsymbol{R}_{i-L} \boldsymbol{z}_i)_j \\ \vdots \end{pmatrix} = \frac{\sigma}{\sqrt{d}} \sum_{j=1}^d q_j \begin{pmatrix} \vdots \\ (\boldsymbol{R}_{i-L} \boldsymbol{z}_i)_j \\ \vdots \end{pmatrix} := \frac{1}{\sqrt{d}} \sum_{j=1}^d \boldsymbol{c}_j.$$

Since $\tilde{\boldsymbol{z}}_i := \boldsymbol{R}_{i-L} \boldsymbol{z}_i \overset{\text{iid}}{\sim} \mathcal{N}(\boldsymbol{0}, \boldsymbol{I}_d)$, the vectors $\boldsymbol{c}_j \in \mathbb{R}^L$ are i.i.d. and satisfy

$$\mathbb{E}(\boldsymbol{c}_1) = \boldsymbol{0}_L, \qquad \operatorname{Cov}(\boldsymbol{c}_1) = \mathbb{E}\left(\sigma^2 q_1^2 \left(\tilde{z}_{i1} \tilde{z}_{j1}\right)_{i,j}\right) = \sigma^2 \boldsymbol{I}_L.$$

By the multivariate central limit theorem,

$$b \xrightarrow{\mathcal{D}} \mathcal{N}(\mathbf{0}_L, \sigma^2 \boldsymbol{I}_L) \quad \text{as } d \to \infty.$$

By Slutsky's theorem,

$$(s_1, \ldots, s_L) \xrightarrow{\mathcal{D}} \mathcal{N}\left((\mu_1, \ldots, \mu_L), \sigma^2 \boldsymbol{I}_L\right) \quad \text{as } d \to \infty,$$

where

$$\mu_i = \rho \lim_{d \to \infty} \frac{2}{d} \sum_{f=0}^{d/2-1} \cos\left((i-L)\theta_f\right), \quad \forall i \in \{1, \ldots, L\}.$$

For NoPE, $\boldsymbol{R} = \boldsymbol{I}_d$, so $\theta_f = 0$ for all $f \in \{0, \ldots, d/2 - 1\}$. Hence

$$\mu_i = \rho, \quad \forall i \in \{1, \ldots, L\}.$$

For RoPE, $\theta_f = b^{-2f/d}$ for $f \in \{0, \ldots, d/2 - 1\}$. Thus

$$\mu_i = \rho \lim_{d \to \infty} \frac{2}{d} \sum_{f=0}^{d/2-1} \cos\left((i-L)b^{-2f/d}\right) = \rho \int_0^1 \cos\left((i-L)b^{-x}\right) dx, \quad \forall i \in \{1, \ldots, L\}.$$

$\square$

### F.5. Proof of Proposition 2

We first analyze the limiting behavior of the partition functions $Z(\boldsymbol{s}; \lambda)$ and $Z(\boldsymbol{s}; \beta\lambda)$, as stated in Lemma 2.

**Lemma 2.** *Let the logits $s_1, \ldots, s_L$ be independent Gaussian random variables with common variance $\sigma^2$, i.e., $s_i \sim \mathcal{N}(\mu_i, \sigma^2)$. Write $\boldsymbol{\mu} := (\mu_1, \ldots, \mu_L)$ and assume $\|\boldsymbol{\mu}\|_\infty := \max_{1 \le i \le L} |\mu_i| < \infty$. Let the inverse temperature $\tau = \tau(L)$ possibly depend on $L$ (e.g., $\tau \in \{\lambda, \beta\lambda\}$), and define the scaling parameter*

$$\Lambda := \limsup_{L \to \infty} \frac{\tau(L)\sigma}{\sqrt{\ln L}}.$$

*Let*

$$S_L(\tau) = \sum_{i=1}^L e^{\tau s_i}.$$

*Then:*

(i) *If $0 \le \Lambda < \sqrt{2}$, then*

$$\frac{S_L(\tau)}{\mathbb{E}\, S_L(\tau)} \xrightarrow{\mathbb{P}} 1 \quad \text{as } L \to \infty.$$

(ii) *If $\Lambda = \sqrt{2}$ and*

$$\lim_{L \to \infty} \frac{\mathcal{E}_2(\mathring{\boldsymbol{\alpha}})}{L/\sqrt{\ln L}} \to \infty, \quad \text{where} \quad \mathcal{E}_2(\mathring{\boldsymbol{\alpha}}) := \frac{\left(\sum_{i=1}^L e^{\tau \mu_i}\right)^2}{\sum_{i=1}^L e^{2\tau \mu_i}},$$

   *then*

$$\frac{S_L(\tau)}{\mathbb{E}\, S_L(\tau)} \xrightarrow{\mathbb{P}} \frac{1}{2} \quad \text{as } L \to \infty.$$

(iii) *If $\sqrt{2} < \Lambda < \infty$, the law of large numbers fails: the normalized sum $S_L(\tau)/\mathbb{E}\, S_L(\tau)$ can converge to a nondegenerate, unbounded random variable on $[0, \infty)$.*

(iv) *If $\Lambda = \infty$, then*

$$\frac{S_L(\tau)}{M_L(\tau)} \xrightarrow{\mathbb{P}} 1 \quad \text{as } L \to \infty,$$

   *where $M_L(\tau) := \max_{1 \le i \le L} e^{\tau s_i}$.*

**Proof.** For each $s_i \sim \mathcal{N}(\mu_i, \sigma^2)$, $e^{\tau s_i}$ is log-normal with expectation $\mathbb{E}\, e^{\tau s_i} = e^{\tau \mu_i + \frac{1}{2}\tau^2 \sigma^2}$.

**(i)** If $0 \leq \Lambda < \sqrt{2}$, let

$$\bar{S}_L(\tau) := \frac{S_L(\tau)}{\mathbb{E}\, S_L(\tau)}.$$

It suffices to show that for some $r > 1$, $\lim_{L \to \infty} \mathbb{E}\, |\bar{S}_L(\tau) - 1|^r = 0$, which implies $\bar{S}_L(\tau) \xrightarrow{\mathbb{P}} 1$ as $L \to \infty$. By the von Bahr–Esseen inequality (von Bahr & Esseen, 1965, Theorem 2), for any $r \in [1, 2]$,

$$\mathbb{E}\, |\bar{S}_L(\tau) - 1|^r = \mathbb{E}\, \left| \sum_{i=1}^{L} \frac{e^{\tau s_i} - \mathbb{E}\, e^{\tau s_i}}{\mathbb{E}\, S_L(\tau)} \right|^r \leq 2 \sum_{i=1}^{L} \mathbb{E}\, \left| \frac{e^{\tau s_i} - \mathbb{E}\, e^{\tau s_i}}{\mathbb{E}\, S_L(\tau)} \right|^r = 2 \frac{\sum_{i=1}^{L} \mathbb{E}\, |e^{\tau s_i} - \mathbb{E}\, e^{\tau s_i}|^r}{\left| \sum_{i=1}^{L} \mathbb{E}\, e^{\tau s_i} \right|^r}.$$

For the denominator, since $\mu_i \geq -\|\boldsymbol{\mu}\|_\infty$,

$$\sum_{i=1}^{L} \mathbb{E}\, e^{\tau s_i} = \sum_{i=1}^{L} e^{\tau \mu_i + \frac{1}{2}\tau^2 \sigma^2} \geq L e^{-\tau \|\boldsymbol{\mu}\|_\infty + \frac{1}{2}\tau^2 \sigma^2}.$$

For the numerator, using the power mean inequality $(x + y)^r \leq 2^{r-1}(x^r + y^r)$ for $r \geq 1$ and $x, y \geq 0$,

$$\sum_{i=1}^{L} \mathbb{E}\, |e^{\tau s_i} - \mathbb{E}\, e^{\tau s_i}|^r \leq \sum_{i=1}^{L} \mathbb{E}(e^{\tau s_i} + \mathbb{E}\, e^{\tau s_i})^r$$

$$\leq 2^{r-1} \sum_{i=1}^{L} [\mathbb{E}\, e^{r\tau s_i} + (\mathbb{E}\, e^{\tau s_i})^r]$$

$$\leq 2^{r-1} L (e^{r\tau \|\boldsymbol{\mu}\|_\infty + \frac{1}{2}r^2 \tau^2 \sigma^2} + e^{r\tau \|\boldsymbol{\mu}\|_\infty + \frac{1}{2}r\tau^2 \sigma^2}).$$

Hence,

$$\mathbb{E}\, |\bar{S}_L(\tau) - 1|^r \leq 2^r L^{1-r} \frac{e^{r\tau \|\boldsymbol{\mu}\|_\infty + \frac{1}{2}r^2 \tau^2 \sigma^2} + e^{r\tau \|\boldsymbol{\mu}\|_\infty + \frac{1}{2}r\tau^2 \sigma^2}}{e^{-r\tau \|\boldsymbol{\mu}\|_\infty + \frac{1}{2}r\tau^2 \sigma^2}}$$

$$= 2^r L^{1-r} \left( e^{2r\tau \|\boldsymbol{\mu}\|_\infty + \frac{1}{2}r(r-1)\tau^2 \sigma^2} + e^{2r\tau \|\boldsymbol{\mu}\|_\infty} \right)$$

$$= 2^r \exp\left( -(r-1)\ln L + 2r\tau \|\boldsymbol{\mu}\|_\infty + \frac{1}{2}r(r-1)\tau^2 \sigma^2 \right) (1 + o(1)).$$

Thus $\lim_{L \to \infty} \mathbb{E}\, |\bar{S}_L(\tau) - 1|^r = 0$ provided the exponent

$$\frac{1}{2}r(r-1)\tau^2 \sigma^2 - (r-1)\ln L = \frac{r(r-1)}{2} \ln L \cdot \left( \frac{\tau^2 \sigma^2}{\ln L} - \frac{2}{r} \right) \to -\infty \quad \text{as } L \to \infty.$$

Since $\Lambda < \sqrt{2}$, there exists $\varepsilon > 0$ such that for sufficiently large $L$, $0 \leq \tau^2 \sigma^2 / \ln L < 2 - \varepsilon$. Choose any $r \in (1, 2/(2 - \varepsilon)) \subset (1, 2)$; then for large $L$ we have $\tau^2 \sigma^2 / \ln L < 2/r$, so the exponent tends to $-\infty$. Hence some $r > 1$ satisfies $\lim_{L \to \infty} \mathbb{E}\, |\bar{S}_L(\tau) - 1|^r = 0$, which completes the proof.

**(ii)** If $\Lambda = \sqrt{2}$, write $s_i = \mu_i + \sigma z_i$ with $z_i \overset{\text{iid}}{\sim} \mathcal{N}(0, 1)$. Decompose $S_L(\tau)$ as $S_L(\tau) = S_L^{\leq}(\tau) + S_L^{>}(\tau)$, where

$$S_L^{\leq}(\tau) = \sum_{i=1}^{L} e^{\tau s_i} \mathbb{1}_{\{z_i \leq \tau\sigma\}}, \qquad S_L^{>}(\tau) = \sum_{i=1}^{L} e^{\tau s_i} \mathbb{1}_{\{z_i > \tau\sigma\}}.$$

**1. Bounding $S_L^{>}(\tau)$.** For any $x > 0$, the Mills' ratio gives $\Phi(-x) \leq \phi(x)/x$, where $\Phi$ and $\phi$ denote the CDF and PDF of the standard normal distribution, respectively (see, e.g., Vershynin (2026, Proposition 2.1.2)). By the union bound,

$$\mathbb{P}\left( \max_{1 \leq i \leq L} z_i > \tau\sigma \right) \leq \sum_{i=1}^{L} \mathbb{P}(z_i > \tau\sigma) = L\Phi(-\tau\sigma) \leq L \cdot \frac{1}{\tau\sigma\sqrt{2\pi}} e^{-\frac{1}{2}\tau^2 \sigma^2}.$$

Substituting $\tau\sigma = \sqrt{2\ln L}$, the right-hand side equals $1/(2\sqrt{\pi \ln L}) \to 0$ as $L \to \infty$. It follows that

$$\mathbb{P}\left(S_L^>(\tau) = 0\right) \geq \mathbb{P}\left(\max_{1 \leq i \leq L} z_i \leq \tau\sigma\right) = 1 - \mathbb{P}\left(\max_{1 \leq i \leq L} z_i > \tau\sigma\right) \to 1 \quad \text{as } L \to \infty.$$

Therefore $S_L^>(\tau) \xrightarrow{\mathbb{P}} 0$ as $L \to \infty$.

**2. Bounding $S_L^{\leq}(\tau)$.** Set $Y_i := e^{\tau\sigma z_i} \mathbb{1}_{\{z_i \leq \tau\sigma\}}$. Then

$$\mathbb{E}\, Y_i = \int_{-\infty}^{\tau\sigma} e^{\tau\sigma x} \frac{1}{\sqrt{2\pi}} e^{-\frac{x^2}{2}} \, dx = e^{\frac{1}{2}\tau^2\sigma^2} \int_{-\infty}^{\tau\sigma} \frac{1}{\sqrt{2\pi}} e^{-\frac{(x-\tau\sigma)^2}{2}} \, dx = \frac{1}{2} e^{\frac{1}{2}\tau^2\sigma^2}.$$

Hence

$$\mathbb{E}\, S_L^{\leq}(\tau) = \sum_{i=1}^{L} e^{\tau\mu_i} \cdot \frac{1}{2} \mathbb{E}\, e^{\tau\sigma z_i} = \frac{1}{2} \sum_{i=1}^{L} \mathbb{E}\, e^{\tau\sigma s_i} = \frac{1}{2} \mathbb{E}\, S_L(\tau).$$

Moreover,

$$\mathbb{E}\, Y_i^2 = \int_{-\infty}^{\tau\sigma} e^{2\tau\sigma x} \frac{1}{\sqrt{2\pi}} e^{-\frac{x^2}{2}} \, dx = e^{2\tau^2\sigma^2} \int_{-\infty}^{\tau\sigma} \frac{1}{\sqrt{2\pi}} e^{-\frac{(x-2\tau\sigma)^2}{2}} \, dx = e^{2\tau^2\sigma^2} \Phi(-\tau\sigma),$$

By Mills' ratio,

$$\mathbb{E}\, Y_i^2 \leq e^{2\tau^2\sigma^2} \frac{1}{\tau\sigma\sqrt{2\pi}} e^{-\frac{\tau^2\sigma^2}{2}} = \frac{1}{\tau\sigma\sqrt{2\pi}} e^{\frac{3}{2}\tau^2\sigma^2}$$

Therefore,

$$\frac{\text{Var}(S_L^{\leq}(\tau))}{\mathbb{E}\, S_L^{\leq}(\tau)^2} = \frac{\sum_{i=1}^{L} e^{2\tau\mu_i} \text{Var}(Y_i)}{(\frac{1}{2} \sum_{i=1}^{L} e^{\tau\mu_i + \frac{1}{2}\tau^2\sigma^2})^2} \leq \frac{\sum_{i=1}^{L} e^{2\tau\mu_i} \mathbb{E}(Y_i^2)}{(\frac{1}{2} \sum_{i=1}^{L} e^{\tau\mu_i + \frac{1}{2}\tau^2\sigma^2})^2}$$

$$= \frac{4}{\tau\sigma\sqrt{2\pi}} e^{\frac{1}{2}\tau^2\sigma^2} \frac{\sum_{i=1}^{L} e^{2\tau\mu_i}}{(\sum_{i=1}^{L} e^{\tau\mu_i})^2} = \frac{2L}{\sqrt{\pi\ln L}} \cdot \frac{1}{\mathcal{E}_2(\mathring{\boldsymbol{\alpha}})} \to 0 \quad \text{as } L \to \infty,$$

where we used $\tau\sigma = \sqrt{2\ln L}$ in the last step and $\mathcal{E}_2(\mathring{\boldsymbol{\alpha}})$ denotes the effective sample size defined by $(\sum_{i=1}^{L} e^{\tau\mu_i})^2 / (\sum_{i=1}^{L} e^{2\tau\mu_i})$. By Chebyshev's inequality, for any $\delta > 0$,

$$\mathbb{P}\left(\left|\frac{S_L^{\leq}(\tau)}{\mathbb{E}\, S_L^{\leq}(\tau)} - 1\right| > \delta\right) \leq \frac{1}{\delta^2} \frac{\text{Var}(S_L^{\leq}(\tau))}{\mathbb{E}\, S_L^{\leq}(\tau)^2} \to 0 \quad \text{as } L \to \infty.$$

Finally,

$$\frac{S_L(\tau)}{\mathbb{E}\, S_L(\tau)} = \frac{S_L^{\leq}(\tau)}{\mathbb{E}\, S_L(\tau)} + \frac{S_L^>(\tau)}{\mathbb{E}\, S_L(\tau)} = \frac{S_L^{\leq}(\tau)}{2\,\mathbb{E}\, S_L^{\leq}(\tau)} + \frac{S_L^>(\tau)}{\mathbb{E}\, S_L(\tau)} \xrightarrow{\mathbb{P}} \frac{1}{2} \quad \text{as } L \to \infty.$$

**(iii)** Consider the case $\sqrt{2} < \Lambda < \infty$. In the simplest setting, where the $s_i$ are i.i.d. standard Gaussian (i.e., $\mu_i \equiv 0$ and $\sigma = 1$), $S_L(\tau)$ converges in distribution to a nondegenerate stable law on $[0, \infty)$ (see, e.g., Molchanov & Panov (2019), Proposition 3.1) and Ben Arous et al. (2005, Theorem 3)). Hence the law of large numbers fails.

**(iv)** If $\Lambda = \infty$, define $\zeta_L := \frac{S_L(\tau) - M_L(\tau)}{M_L(\tau)}$; it suffices to show $\zeta_L \xrightarrow{\mathbb{P}} 0$ as $L \to \infty$. Let $s_{(L)} := \max_{1 \leq i \leq L} s_i$ denote the largest logit, so that $M_L(\tau) = \max_{1 \leq i \leq L} e^{\tau s_i} = e^{\tau s_{(L)}}$. Denote by $F_i(z)$ and $f_i(z)$ the CDF and PDF of $s_i$, respectively.

**1. Bounding $s_{(L)}$.** Let $F_{(L)}(z) := \prod_{i=1}^{L} F_i(z)$ be the CDF of $s_{(L)}$. Since $F_{(L)}(\cdot)$ is non-decreasing, choose $\gamma_L := \min\left\{\ln L, \left(\tau/\sqrt{\ln L}\right)^{1/2}\right\}$ and define $s_* := \inf\left\{z : F_{(L)}(z) = e^{-\gamma_L}\right\}$. Then $\mathbb{P}(s_{(L)} < s_*) = e^{-\gamma_L} \to 0$ as $L \to \infty$. Moreover,

$$\gamma_L = -\sum_{i=1}^{L} \ln F_i(s_*) = -\sum_{i=1}^{L} \ln \Phi\left(\frac{s_* - \mu_i}{\sigma}\right) \geq -L \ln \Phi\left(\frac{s_* + \|\boldsymbol{\mu}\|_\infty}{\sigma}\right).$$

Hence

$$-\frac{\gamma_L}{L} \le \ln \Phi \left( \frac{s_* + \|\boldsymbol{\mu}\|_\infty}{\sigma} \right) \le 0.$$

Letting $L \to \infty$ implies $\Phi \left( \frac{s_* + \|\boldsymbol{\mu}\|_\infty}{\sigma} \right) \to 1$, and therefore $s_* \to \infty$. On the other hand,

$$\gamma_L = -\sum_{i=1}^{L} \ln \Phi \left( \frac{s_* - \mu_i}{\sigma} \right) \le -L \ln \Phi \left( \frac{s_* - \|\boldsymbol{\mu}\|_\infty}{\sigma} \right).$$

If $\frac{s_* - \|\boldsymbol{\mu}\|_\infty}{\sigma} \ge 1$, then $\Phi \left( \frac{s_* - \|\boldsymbol{\mu}\|_\infty}{\sigma} \right) \ge \frac{1}{2}$. For $x \in [\frac{1}{2}, 1)$, we have $\ln x \ge 1 - \frac{1}{x} = -\frac{1-x}{x} \ge -2(1-x)$. Thus, by Mills' ratio,

$$\ln \Phi \left( \frac{s_* - \|\boldsymbol{\mu}\|_\infty}{\sigma} \right) \ge -2 \left( 1 - \Phi \left( \frac{s_* - \|\boldsymbol{\mu}\|_\infty}{\sigma} \right) \right)$$

$$\ge -2 \frac{1}{\sqrt{2\pi} \left( \frac{s_* - \|\boldsymbol{\mu}\|_\infty}{\sigma} \right)} \exp \left( -\frac{1}{2} \left( \frac{s_* - \|\boldsymbol{\mu}\|_\infty}{\sigma} \right)^2 \right)$$

$$\ge - \exp \left( -\frac{1}{2} \left( \frac{s_* - \|\boldsymbol{\mu}\|_\infty}{\sigma} \right)^2 \right).$$

Therefore, for $L \ge 3$,

$$1 \le \gamma_L \le L \exp \left( -\frac{1}{2} \left( \frac{s_* - \|\boldsymbol{\mu}\|_\infty}{\sigma} \right)^2 \right),$$

which implies $s_* \le \sigma \sqrt{2 \ln L} + \|\boldsymbol{\mu}\|_\infty$. To obtain an upper bound for $s_{(L)}$, set $s^* := 2\sigma \sqrt{\ln L} + \|\boldsymbol{\mu}\|_\infty$. By the union bound,

$$\mathbb{P}(s_{(L)} > s^*) = \mathbb{P} \left( \bigcup_{i=1}^{L} \{s_i > s^*\} \right) \le \sum_{i=1}^{L} \mathbb{P}(s_i > s^*)$$

$$= \sum_{i=1}^{L} \Phi \left( \frac{\mu_i - s^*}{\sigma} \right) \le L \Phi \left( \frac{\|\boldsymbol{\mu}\|_\infty - s^*}{\sigma} \right)$$

$$= L \Phi \left( -2 \sqrt{\ln L} \right) \le L e^{-2 \ln L} = \frac{1}{L} \to 0 \quad \text{as } L \to \infty,$$

where the last inequality follows from $\Phi(-x) \le \frac{1}{2} e^{-x^2/2} < e^{-x^2/2}$ for $x > 0$. Combining the two bounds yields

$$\mathbb{P} \left( s_{(L)} \notin [s_*, s^*] \right) \le \mathbb{P}(s_{(L)} < s_*) + \mathbb{P}(s_{(L)} > s^*) \to 0 \quad \text{as } L \to \infty. \tag{55}$$

2.  Suppose $z \in [s_*, s^*]$. Since $s_* \to \infty$ as $L \to \infty$, for sufficiently large $L$ we have $s_* > \|\boldsymbol{\mu}\|_\infty$. For each $i \in \{1, \ldots, L\}$, define

$$R_i(z) := e^{-\tau z} \, \mathbb{E} \left[ e^{\tau s_i} \mid s_i \le z \right]$$

$$= \frac{e^{-\tau z}}{\Phi \left( \frac{z - \mu_i}{\sigma} \right)} \int_{-\infty}^{z} e^{\tau u} \frac{1}{\sqrt{2\pi}\sigma} e^{-\frac{(u - \mu_i)^2}{2\sigma^2}} \, du$$

$$\overset{v = \frac{u - \mu_i}{\sigma}}{=\!=\!=\!=\!=} \frac{e^{-\tau(z - \mu_i)}}{\Phi \left( \frac{z - \mu_i}{\sigma} \right)} \int_{-\infty}^{\frac{z - \mu_i}{\sigma}} e^{\tau \sigma v} \frac{1}{\sqrt{2\pi}} e^{-\frac{v^2}{2}} \, dv$$

$$= \exp \left( \frac{1}{2} \tau^2 \sigma^2 - \tau(z - \mu_i) \right) \frac{\Phi \left( \frac{z - \mu_i}{\sigma} - \tau\sigma \right)}{\Phi \left( \frac{z - \mu_i}{\sigma} \right)}.$$

Since $s^* = 2\sigma \sqrt{\ln L} + \|\boldsymbol{\mu}\|_\infty$ and $\tau = \omega(\sqrt{\ln L})$, for sufficiently large $L$ we have $z \le s^* < -\|\boldsymbol{\mu}\|_\infty + \tau\sigma^2$, and hence $\frac{z - \mu_i}{\sigma} - \tau\sigma < 0$ for every $i \in \{1, \ldots, L\}$. Applying Mills' ratio yields

$$\Phi \left( \frac{z - \mu_i}{\sigma} - \tau\sigma \right) \le \frac{1}{\sqrt{2\pi} \left( \tau\sigma - \frac{z - \mu_i}{\sigma} \right)} \exp \left( -\frac{1}{2} \tau^2 \sigma^2 + \tau(z - \mu_i) - \frac{(z - \mu_i)^2}{2\sigma^2} \right).$$

Therefore,

$$R_i(z) \le \frac{1}{\left(\tau\sigma - \frac{z-\mu_i}{\sigma}\right)\Phi\left(\frac{z-\mu_i}{\sigma}\right)} \cdot \frac{1}{\sqrt{2\pi}} e^{-\frac{(z-\mu_i)^2}{2\sigma^2}} = \frac{\sigma^2 f_i(z)}{(\tau\sigma^2 - (z - \mu_i))\, F_i(z)} =: \tilde{R}_i(z).$$

Since $z - \mu_i \ge s_* - \mu_i \ge s_* - \|\boldsymbol{\mu}\|_\infty > 0$,

$$\begin{aligned}
\frac{\mathrm{d}}{\mathrm{d}z}\ln\tilde{R}_i(z) &= \frac{f_i'(z)}{f_i(z)} - \frac{f_i(z)}{F_i(z)} + \frac{1}{\tau\sigma^2 - (z - \mu_i)} \\
&= -\frac{z - \mu_i}{\sigma^2} - \frac{f_i(z)}{F_i(z)} + \frac{1}{\tau\sigma^2 - (z - \mu_i)} \\
&\le -\frac{s_* - \mu_i}{\sigma^2} + \frac{1}{\tau\sigma^2 - (s_* - \mu_i)}.
\end{aligned}$$

Since $s_* \le \sigma\sqrt{2\ln L} + \|\boldsymbol{\mu}\|_\infty$ and $\tau = \omega(\sqrt{\ln L})$, for sufficiently large $L$ we have $\tau\sigma^2 \gg s^* - \mu_i > s_* - \mu_i$. Thus, $\tilde{R}_i'(z) < 0$, so the maximum of $\tilde{R}_i(z)$ is attained at $s_*$. Moreover, since $F_i(s_*) \ge \frac{1}{2}$ and $\tau\sigma^2 \ge 2(s^* - \mu_i) \ge 2(z - \mu_i)$, we have

$$R_i(z) \le \tilde{R}_i(z) \le \tilde{R}_i(s_*) \le \frac{4}{\tau} f_i(s_*).$$

For any $x \ge 1$, Mills' ratio gives $1 - \Phi(x) \ge \frac{x}{x^2+1}\phi(x) \ge \frac{1}{2x}\phi(x)$, where $\phi$ is the standard normal density. Since $s_* \ge \|\boldsymbol{\mu}\|_\infty$, for sufficiently large $L$,

$$f_i(s_*) = \frac{1}{\sigma}\phi\left(\frac{s_* - \mu_i}{\sigma}\right) \le \frac{2}{\sigma}\left(\frac{s_* - \mu_i}{\sigma}\right)\left(1 - \Phi\left(\frac{s_* - \mu_i}{\sigma}\right)\right) \le \frac{4s_*}{\sigma^2}(1 - F_i(s_*)).$$

Combining these inequalities yields

$$\sup_{z\in[s_*,s^*]} R_i(z) \le \frac{16 s_*}{\tau\sigma^2}(1 - F_i(s_*)), \quad \forall i \in \{1, \dots, L\}. \tag{56}$$

Summing Eq. (56) over $i \in \{1, \dots, L\}$, using the definition of $s_*$, and applying $\ln x \le x - 1$, we obtain, for any $z \in [s_*, s^*]$,

$$\sum_{i=1}^{L} R_i(z) \le \frac{16 s_*}{\tau\sigma^2}\sum_{i=1}^{L}(1 - F_i(s_*)) \le -\frac{16 s_*}{\tau\sigma^2}\sum_{i=1}^{L}\ln F_i(s_*) = \frac{16 s_* \gamma_L}{\tau\sigma^2}.$$

Using $s_* \le \sigma\sqrt{2\ln L} + \|\boldsymbol{\mu}\|_\infty$ and $\gamma_L \le \left(\tau/\sqrt{\ln L}\right)^{1/2}$, we have

$$\sup_{z\in[s_*,s^*]} \sum_{i=1}^{L} R_i(z) \le \frac{16}{\tau\sigma^2}\left(\sigma\sqrt{2\ln L} + \|\boldsymbol{\mu}\|_\infty\right)\left(\frac{\tau}{\sqrt{\ln L}}\right)^{1/2} \lesssim \left(\frac{\sqrt{\ln L}}{\tau}\right)^{1/2} \to 0 \quad \text{as } L \to \infty.$$

**3. Bounding $\zeta_L$.** For any $\delta > 0$, write

$$\mathbb{P}\left(\zeta_L > \delta\right) \le \mathbb{P}(\zeta_L > \delta, s_{(L)} \in [s_*, s^*]) + \mathbb{P}(s_{(L)} \notin [s_*, s^*]). \tag{57}$$

By the law of total probability,

$$\mathbb{P}(\zeta_L > \delta, s_{(L)} \in [s_*, s^*]) = \int_{s_*}^{s^*} \mathbb{P}\left(\zeta_L > \delta \mid s_{(L)} = z\right) f_{(L)}(z)\, \mathrm{d}z,$$

where $f_{(L)}(z)$ is the PDF of $s_{(L)}$. By Markov's inequality,

$$\mathbb{P}\left(\zeta_L > \delta \mid s_{(L)} = z\right) \le \frac{1}{\delta}\mathbb{E}\left[\zeta_L \mid s_{(L)} = z\right].$$

Since $s_1, \ldots, s_L$ are independent continuous random variables, the events $\{s_j = z; \; s_i < z, \forall i \neq j\}$ partition the event $\{s_{(L)} = z\}$. Thus,

$$
\begin{aligned}
\mathbb{E}\left[\zeta_L \mid s_{(L)} = z\right] &= \sum_{j=1}^{L} \mathbb{E}\left[\zeta_L \cdot \mathbb{1}\left\{s_j = z; \; s_i < z, \forall i \neq j\right\} \mid s_{(L)} = z\right] \\
&= \sum_{j=1}^{L} \pi_j(z)\, \mathbb{E}\left[\zeta_L \mid s_j = z; \; s_i < z, \forall i \neq j\right] \\
&= \sum_{j=1}^{L} \pi_j(z) \sum_{i \neq j} \mathbb{E}\left[e^{\tau(s_i - z)} \mid s_j = z; \; s_i < z, \forall i \neq j\right] \\
&= \sum_{j=1}^{L} \pi_j(z) \sum_{i \neq j} \mathbb{E}\left[e^{\tau(s_i - z)} \mid s_i < z\right] \\
&= \sum_{j=1}^{L} \pi_j(z) \sum_{i \neq j} R_i(z) \leq \sum_{j=1}^{L} \pi_j(z) \sum_{i=1}^{L} R_i(z) = \sum_{i=1}^{L} R_i(z),
\end{aligned}
$$

where $\pi_j(z) := \mathbb{P}\left(s_j = z; \; s_i < z, \forall i \neq j \mid s_{(L)} = z\right)$ satisfies $\sum_{j=1}^{L} \pi_j(z) = 1$. Therefore,

$$
\begin{aligned}
\mathbb{P}(\zeta_L > \delta, s_{(L)} \in [s_*, s^*]) &\leq \frac{1}{\delta} \sup_{z \in [s_*, s^*]} \sum_{i=1}^{L} R_i(z) \left(\int_{s_*}^{s^*} f_{(L)}(z)\, \mathrm{d}z\right) \\
&\leq \frac{1}{\delta} \sup_{z \in [s_*, s^*]} \sum_{i=1}^{L} R_i(z) \to 0 \quad \text{as } L \to \infty.
\end{aligned}
$$

Combining this with Eq. (55) and substituting into Eq. (57) yields $\zeta_L \xrightarrow{\mathbb{P}} 0$, which completes the proof. $\qquad\square$

Using Lemma 2, we prove the law of large numbers in Proposition 2 for the three cases.

**Proposition 2** (Infinite-length limit of $\mathcal{E}_\beta$)**.** *Let the logits $s_1, \ldots, s_L$ be independent Gaussian random variables with common variance $\sigma^2$, i.e., $s_i \sim \mathcal{N}(\mu_i, \sigma^2)$. Write $\boldsymbol{\mu} := (\mu_1, \ldots, \mu_L)$ and assume $\|\boldsymbol{\mu}\|_\infty := \max_{1 \leq i \leq L} |\mu_i| < \infty$. Let the inverse temperature $\lambda = \lambda(L)$ be a deterministic function of $L$. Define the scaling parameter*

$$
\Lambda := \limsup_{L \to \infty} \frac{\lambda(L)\sigma}{\sqrt{\ln L}}. \tag{31}
$$

*Then the following hold:*

(i) *If $\Lambda < \sqrt{2}\min\{1/\beta, 1\}$, then*

$$
\frac{\mathcal{E}_\beta(\boldsymbol{\alpha})}{\hat{\mathcal{E}}_\beta(L)} \xrightarrow{\mathbb{P}} 1 \quad \text{as } L \to \infty. \tag{32}
$$

(ii) *If $\Lambda = \sqrt{2}\min\{1/\beta, 1\}$ and $\mathcal{E}_2(\mathring{\boldsymbol{\alpha}}) = \omega(L/\sqrt{\ln L})$, then*

$$
\frac{\mathcal{E}_\beta(\boldsymbol{\alpha})}{\hat{\mathcal{E}}_\beta(L)} \xrightarrow{\mathbb{P}} 2^{\frac{1}{\max\{\beta, \beta^{-1}\} - 1}} \quad \text{as } L \to \infty. \tag{33}
$$

(iii) *If $\Lambda > \sqrt{2}/\beta$, then*

$$
\liminf_{L \to \infty} \hat{\mathcal{E}}_\beta(L) = 0. \tag{34}
$$

*Moreover, if $\Lambda = \infty$, then*

$$
\mathcal{E}_\beta(\boldsymbol{\alpha}) \xrightarrow{\mathbb{P}} 1 \quad \text{as } L \to \infty. \tag{35}
$$

**Proof.** For any $\beta \neq 1$,

$$\frac{\mathcal{E}_\beta(\boldsymbol{\alpha})}{\hat{\mathcal{E}}_\beta(L)} = \left(\frac{Z(\boldsymbol{s};\lambda)}{\mathbb{E}\, Z(\boldsymbol{s};\lambda)}\right)^{\frac{\beta}{\beta-1}} \Bigg/ \left(\frac{Z(\boldsymbol{s};\beta\lambda)}{\mathbb{E}\, Z(\boldsymbol{s};\beta\lambda)}\right)^{\frac{1}{\beta-1}}.$$

**(i)** If $\Lambda < \sqrt{2}\min\{1/\beta, 1\}$, then $\Lambda < \sqrt{2}$ and $\beta\Lambda < \sqrt{2}$. By Lemma 2(i),

$$\frac{Z(\boldsymbol{s};\lambda)}{\mathbb{E}\, Z(\boldsymbol{s};\lambda)} \xrightarrow{\mathbb{P}} 1, \quad \frac{Z(\boldsymbol{s};\beta\lambda)}{\mathbb{E}\, Z(\boldsymbol{s};\beta\lambda)} \xrightarrow{\mathbb{P}} 1, \quad \text{as } L \to \infty.$$

Since $g(x, y) = x^{\frac{\beta}{\beta-1}}/y^{\frac{1}{\beta-1}}$ is continuous at $(1, 1)$, the continuous mapping theorem gives

$$\frac{\mathcal{E}_\beta(\boldsymbol{\alpha})}{\hat{\mathcal{E}}_\beta(L)} = g\left(\frac{Z(\boldsymbol{s};\lambda)}{\mathbb{E}\, Z(\boldsymbol{s};\lambda)}, \frac{Z(\boldsymbol{s};\beta\lambda)}{\mathbb{E}\, Z(\boldsymbol{s};\beta\lambda)}\right) \xrightarrow{\mathbb{P}} g(1, 1) = 1 \quad \text{as } L \to \infty.$$

**(ii)** If $\Lambda = \sqrt{2}\min\{1/\beta, 1\}$, then by Lemma 2(i) and (ii), there are two cases. If $\Lambda = \sqrt{2}$ and $\beta\Lambda < \sqrt{2}$, so that $0 < \beta < 1$, then

$$\frac{\mathcal{E}_\beta(\boldsymbol{\alpha})}{\hat{\mathcal{E}}_\beta(L)} \xrightarrow{\mathbb{P}} g\left(\frac{1}{2}, 1\right) = 2^{\frac{\beta}{1-\beta}} \quad \text{as } L \to \infty.$$

If $\Lambda < \sqrt{2}$ and $\beta\Lambda = \sqrt{2}$, so that $\beta > 1$, then

$$\frac{\mathcal{E}_\beta(\boldsymbol{\alpha})}{\hat{\mathcal{E}}_\beta(L)} \xrightarrow{\mathbb{P}} g\left(1, \frac{1}{2}\right) = 2^{\frac{1}{\beta-1}} \quad \text{as } L \to \infty.$$

Combining these two cases, for $\beta \neq 1$,

$$\frac{\mathcal{E}_\beta(\boldsymbol{\alpha})}{\hat{\mathcal{E}}_\beta(L)} \xrightarrow{\mathbb{P}} 2^{\frac{1}{\max\{\beta,\beta^{-1}\}-1}} \quad \text{as } L \to \infty.$$

**(iii)** If $\Lambda > \sqrt{2}/\beta$, then, by the definition of $\Lambda$, there exist $\varepsilon > 0$ and a subsequence $\{L_n\} \subset \mathbb{N}$ such that

$$\frac{\lambda^2(L_n)\sigma^2}{\ln L_n} > \frac{2}{\beta} + \varepsilon, \quad \forall n \in \mathbb{N}.$$

Since $\mathcal{E}_\beta(\mathring{\boldsymbol{\alpha}}) \leq L$, we obtain

$$\hat{\mathcal{E}}_\beta(L_n) \leq e^{-\frac{\beta}{2}\lambda^2(L_n)\sigma^2} L_n < e^{-\frac{\beta}{2}\left(\frac{2}{\beta}+\varepsilon\right)\ln L_n} L_n = L_n^{-\frac{\beta}{2}\varepsilon} \to 0 \quad \text{as } n \to \infty.$$

Therefore, $\liminf_{L \to \infty} \hat{\mathcal{E}}_\beta(L) = 0$. Moreover, if $\Lambda = \infty$, then by Lemma 2(iv),

$$\mathcal{E}_\beta(\boldsymbol{\alpha}) = \frac{\left(\sum_{i=1}^{L} e^{\lambda s_i}\right)^{\frac{\beta}{\beta-1}}}{\left(\sum_{i=1}^{L} e^{\beta\lambda s_i}\right)^{\frac{1}{\beta-1}}} = \left(\frac{\sum_{i=1}^{L} e^{\lambda s_i}}{\max_{1\leq i\leq L} e^{\lambda s_i}}\right)^{\frac{\beta}{\beta-1}} \Bigg/ \left(\frac{\sum_{i=1}^{L} e^{\beta\lambda s_i}}{\max_{1\leq i\leq L} e^{\beta\lambda s_i}}\right)^{\frac{1}{\beta-1}}$$

$$= g\left(\frac{\sum_{i=1}^{L} e^{\lambda s_i}}{\max_{1\leq i\leq L} e^{\lambda s_i}}, \frac{\sum_{i=1}^{L} e^{\beta\lambda s_i}}{\max_{1\leq i\leq L} e^{\beta\lambda s_i}}\right) \xrightarrow{\mathbb{P}} g(1, 1) = 1 \quad \text{as } L \to \infty.$$

$\square$

### F.6. Proof of Theorem 4

**Theorem 4** (Scaling factor of NoPE). *Suppose $\theta_f = 0$ for all $f \in \{0, \ldots, d/2 - 1\}$, and let $\boldsymbol{\alpha} = (\alpha_1, \ldots, \alpha_L)$ denote the attention weights. Let $\mathcal{E}_\beta^*(L)$ denote the head-specific asymptotic limit of $\mathcal{E}_\beta(\boldsymbol{\alpha})$ under the law of large numbers. If one of the following conditions holds*

(i) $\displaystyle \liminf_{L \to \infty} \frac{\ln \mathcal{E}_\beta^*(L)}{\ln L} > 1 - \min\{\beta, \beta^{-1}\}$ *and*

$$\lambda(L) = \sqrt{\frac{2}{\beta \sigma^2} \ln\left(\frac{L}{\mathcal{E}_\beta^*(L)}\right)}; \tag{36}$$

(ii) $\displaystyle \lim_{L \to \infty} \frac{\ln \mathcal{E}_\beta^*(L)}{\ln L} = 1 - \min\{\beta, \beta^{-1}\}$ *and*

$$\lambda(L) = \sqrt{\frac{2}{\beta \sigma^2} \ln\left(\frac{2^{\frac{1}{\max\{\beta, \beta^{-1}\} - 1}} L}{\mathcal{E}_\beta^*(L)}\right)}; \tag{37}$$

(iii) $\displaystyle \lim_{L \to \infty} \mathcal{E}_\beta^*(L) = 1$ *and*

$$\lambda(L) = \omega(\sqrt{\ln L}) \quad \text{as } L \to \infty; \tag{38}$$

*then*

$$\frac{\mathcal{E}_\beta(\boldsymbol{\alpha})}{\mathcal{E}_\beta^*(L)} \xrightarrow{\mathbb{P}} 1 \quad \text{as } d, L \to \infty.$$

**Proof.** By Proposition 1, as $d \to \infty$, the limiting distribution of $\boldsymbol{\alpha}$ is $\boldsymbol{g} \sim \mathcal{N}(\boldsymbol{\mu}, \sigma^2 \boldsymbol{I}_d)$. For NoPE, Eq. (26) gives $\boldsymbol{\mu} = \rho \boldsymbol{1}_d$, so $Z(\boldsymbol{\mu}; \lambda) = Le^\rho$ and $Z(\boldsymbol{\mu}; \beta\lambda) = Le^{\rho\beta}$. Therefore, by Eqs. (29) and (30),

$$\mathcal{E}_\beta(\mathring{\boldsymbol{g}}) = \frac{Z(\boldsymbol{\mu}; \lambda)^{\frac{\beta}{\beta-1}}}{Z(\boldsymbol{\mu}; \beta\lambda)^{\frac{1}{\beta-1}}} = L, \qquad \hat{\mathcal{E}}_\beta(L) = e^{-\frac{\beta}{2}\lambda(L)^2 \sigma^2} L.$$

By Slutsky's theorem,

$$\frac{\mathcal{E}_\beta(\boldsymbol{\alpha})}{\mathcal{E}_\beta^*(L)} = \frac{\mathcal{E}_\beta(\boldsymbol{\alpha})}{\mathcal{E}_\beta(\boldsymbol{g})} \cdot \frac{\mathcal{E}_\beta(\boldsymbol{g})}{\mathcal{E}_\beta^*(L)} \xrightarrow{\mathbb{P}} 1. \quad \text{as } d, L \to \infty.$$

By Proposition 2, the form of $\mathcal{E}_\beta^*(L)$ and the scaling factor $\lambda(L)$ depend on the range of $\Lambda$.

**(i)** If $\Lambda < \sqrt{2} \min\{1/\beta, 1\}$, then Proposition 2(i) gives

$$\mathcal{E}_\beta^*(L) = \hat{\mathcal{E}}_\beta(L) = e^{-\frac{\beta}{2}\lambda(L)^2 \sigma^2} L.$$

Thus,

$$\lambda(L) = \sqrt{\frac{2}{\beta \sigma^2} \ln\left(\frac{L}{\mathcal{E}_\beta^*(L)}\right)}.$$

Hence, the condition is equivalent to

$$\Lambda = \limsup_{L \to \infty} \sqrt{\frac{2}{\beta} \cdot \frac{\ln(L/\mathcal{E}_\beta^*(L))}{\ln L}} = \sqrt{\frac{2}{\beta}\left(1 - \liminf_{L \to \infty} \frac{\ln \mathcal{E}_\beta^*(L)}{\ln L}\right)} < \sqrt{2} \min\{1/\beta, 1\},$$

which implies

$$\liminf_{L \to \infty} \frac{\ln \mathcal{E}_\beta^*(L)}{\ln L} > 1 - \min\{\beta, \beta^{-1}\}.$$

**(ii)** If $\Lambda = \sqrt{2}\min\{1/\beta, 1\}$, then Proposition 2(ii) gives

$$\mathcal{E}_\beta^*(L) = 2^{\overline{\max\{\beta,\beta^{-1}\}-1}}\hat{\mathcal{E}}_\beta(L) = e^{-\frac{\beta}{2}\lambda(L)^2\sigma^2}2^{\overline{\max\{\beta,\beta^{-1}\}-1}}L.$$

Thus,

$$\lambda(L) = \sqrt{\frac{2}{\beta\sigma^2}\ln\left(\frac{2^{\frac{1}{\max\{\beta,\beta^{-1}\}-1}}L}{\mathcal{E}_\beta^*(L)}\right)}.$$

Hence, the condition is equivalent to

$$\Lambda = \sqrt{\frac{2}{\beta}\left(1 - \lim_{L\to\infty}\frac{\ln\mathcal{E}_\beta^*(L)}{\ln L}\right)} = \sqrt{2}\min\{1/\beta, 1\},$$

which gives

$$\lim_{L\to\infty}\frac{\ln\mathcal{E}_\beta^*(L)}{\ln L} = 1 - \min\{\beta, \beta^{-1}\}.$$

**(iii)** If $\Lambda = \infty$, then Proposition 2(iii) gives $\mathcal{E}_\beta^*(L) = 1$ and

$$\Lambda = \sigma\lim_{L\to\infty}\frac{\lambda(L)}{\sqrt{\ln L}} = \infty.$$

Therefore, the condition is automatically satisfied. □

### F.7. Proof of Theorem 5

**Theorem 5** (Scaling factor of RoPE). *Suppose $\theta_f = b^{-2f/d}$ for all $f \in \{0, \ldots, d/2 - 1\}$, and let $\boldsymbol{\alpha} = (\alpha_1, \ldots, \alpha_L)$ denote the attention weights. Let $\mathcal{E}_\beta^*(L)$ denote the head-specific asymptotic limit of $\mathcal{E}_\beta(\boldsymbol{\alpha})$ under the law of large numbers. If $\lambda(L) \to \infty$ and $\lambda(L)\ln L = o(e^{\rho\min\{1,\beta\}\lambda(L)})$ as $L \to \infty$, and one of the following conditions holds*

**(i)** $\liminf_{L\to\infty}\dfrac{\ln\mathcal{E}_\beta^*(L)}{\ln L} > 1 - \min\{\beta, \beta^{-1}\}$ *and*

$$\lambda(L) = \sqrt{\frac{2}{\beta\sigma^2}\ln\left(\frac{1}{\mathcal{E}_\beta^*(L)}\left(L + O\left(e^{\rho\max\{1,\beta\}\lambda(L)}\right)\right)\right)} \quad \text{as } L \to \infty; \tag{39}$$

**(ii)** $\lim_{L\to\infty}\dfrac{\ln\mathcal{E}_\beta^*(L)}{\ln L} = 1 - \min\{\beta, \beta^{-1}\}$ *and*

$$\lambda(L) = \sqrt{\frac{2}{\beta\sigma^2}\ln\left(\frac{2^{\frac{1}{\max\{\beta,\beta^{-1}\}-1}}}{\mathcal{E}_\beta^*(L)}\left(L + O\left(e^{\rho\max\{1,\beta\}\lambda(L)}\right)\right)\right)} \quad \text{as } L \to \infty; \tag{40}$$

**(iii)** $\lim_{L\to\infty}\mathcal{E}_\beta^*(L) = 1$ *and*

$$\lambda(L) = \omega(\sqrt{\ln L}) \quad \text{as } L \to \infty; \tag{41}$$

*then*

$$\frac{\mathcal{E}_\beta(\boldsymbol{\alpha})}{\mathcal{E}_\beta^*(L)} \xrightarrow{\mathbb{P}} 1 \quad \text{as } d, L \to \infty.$$

**Proof.** By Proposition 1, as $d \to \infty$, the limiting distribution of $\boldsymbol{\alpha}$ is $\boldsymbol{g} \sim \mathcal{N}(\boldsymbol{\mu}, \sigma^2 \boldsymbol{I}_d)$. For RoPE, Eq. (27) gives $\mu_i = \rho \int_0^1 \cos\left((i - L)b^{-x}\right) \mathrm{d}x$ for all $i \in \{1, \dots, L\}$. By Slutsky's theorem,

$$\frac{\mathcal{E}_\beta(\boldsymbol{\alpha})}{\mathcal{E}_\beta^*(L)} = \frac{\mathcal{E}_\beta(\boldsymbol{\alpha})}{\mathcal{E}_\beta(\boldsymbol{g})} \cdot \frac{\mathcal{E}_\beta(\boldsymbol{g})}{\mathcal{E}_\beta^*(L)} \xrightarrow{\mathbb{P}} 1. \quad \text{as } d, L \to \infty.$$

When $\Lambda = \infty$ (case (iii)), the proof is identical to that of Theorem 4(iii). It remains to consider $\Lambda \leq \sqrt{2} \min \{1/\beta, 1\}$ (cases (i) and (ii)). As in the proof of Theorem 4(i) and (ii), by Proposition 2, it suffices to show that for any $\beta \neq 1$,

$$\mathcal{E}_\beta(\mathring{\boldsymbol{g}}) = \frac{Z(\boldsymbol{\mu}; \lambda)^{\frac{\beta}{\beta-1}}}{Z(\boldsymbol{\mu}; \beta\lambda)^{\frac{1}{\beta-1}}} = L + O\left(e^{\rho \max\{1,\beta\}\lambda}\right) \quad \text{as } L \to \infty. \tag{58}$$

Note that $\mathcal{E}_2(\mathring{\boldsymbol{g}}) = L + O(e^{2\lambda}) = \omega(L/\sqrt{\ln L})$, which satisfies the condition required by Proposition 2(ii). Let $I_k := \int_0^1 \cos(kb^{-x}) \mathrm{d}x$. Then $\mu_i = \rho I_{i-L} = \rho I_{L-i}$. Moreover,

$$I_k = \int_0^1 \cos(kb^{-x}) \mathrm{d}x \xlongequal{u=b^{-x}} \int_0^{1/b} \cos(ku) \frac{-\mathrm{d}u}{u \ln b} = \frac{1}{\ln b} \int_{1/b}^1 \frac{\cos(ku)}{u} \mathrm{d}u.$$

Define

$$S_L(\tau) := Z(\boldsymbol{\mu}; \tau/\rho) = \sum_{i=1}^L e^{(\tau/\rho)\mu_i} = \sum_{k=0}^{L-1} e^{\tau I_k},$$

where $\tau \in \{\rho\lambda, \rho\beta\lambda\}$ corresponds to the inverse temperature. We prove Eq. (58) by showing that $S_L(\tau) = L + e^\tau + o(e^\tau)$ as $L \to \infty$.

**1. Bounding $I_k$.** We first show that, for any $k \geq b$, $|I_k| \leq (b + 1)/(k \ln b)$. We estimate $I_k$ using the cosine integral function $\mathrm{Ci}(z)$, defined by

$$\mathrm{Ci}(z) = -\int_z^\infty \frac{\cos t}{t} \mathrm{d}t. \tag{59}$$

We first prove that, for any $z \in [1, \infty)$, $|\mathrm{Ci}(z)| \leq 1/z$. For any $t > 0$, we have $t^{-1} = \int_0^\infty e^{-ut} \mathrm{d}u$. Applying Fubini's theorem on truncated intervals and then taking the limit gives

$$\begin{aligned}
\mathrm{Ci}(z) &= -\lim_{K \to \infty} \int_z^K \frac{\cos t}{t} \mathrm{d}t \\
&= -\lim_{K \to \infty} \int_z^K \left(\int_0^\infty e^{-ut} \cos t \, \mathrm{d}u\right) \mathrm{d}t \\
&\overset{(a)}{=} -\lim_{K \to \infty} \int_0^\infty \left(\int_z^K e^{-ut} \cos t \, \mathrm{d}t\right) \mathrm{d}u \\
&= -\lim_{K \to \infty} \int_0^\infty \left(\left.\frac{e^{-ut}(\sin t - u \cos t)}{1 + u^2}\right|_{t=z}^K\right) \mathrm{d}u \\
&= \int_0^\infty \frac{e^{-uz}(\sin z - u \cos z)}{1 + u^2} \mathrm{d}u - \lim_{K \to \infty} \int_0^\infty \frac{e^{-uK}(\sin K - u \cos K)}{1 + u^2} \mathrm{d}u \\
&\overset{(b)}{=} \int_0^\infty e^{-uz} \frac{\sin z - u \cos z}{1 + u^2} \mathrm{d}u.
\end{aligned}$$

Here, step (a) follows from Fubini's theorem on $[z, K] \times [0, \infty)$, and step (b) follows from the dominated convergence theorem since the integrand in the second term is bounded by the integrable function $e^{-uz}$. By the Cauchy–Schwarz inequality, $|\sin z - u \cos z| \leq \sqrt{1 + u^2}$, and hence

$$|\mathrm{Ci}(z)| \leq \int_0^\infty e^{-uz} \frac{\sqrt{1 + u^2}}{1 + u^2} \mathrm{d}u = \int_0^\infty \frac{e^{-uz}}{\sqrt{1 + u^2}} \mathrm{d}u \leq \int_0^\infty e^{-uz} \mathrm{d}u = \frac{1}{z}.$$

Since $\mathrm{Ci}'(z) = \cos(z)/z$, the chain rule gives $\frac{\mathrm{d}}{\mathrm{d}z}\,\mathrm{Ci}(kz) = \cos(kz)/z$. Therefore,

$$I_k = \frac{1}{\ln b}\int_{1/b}^{1}\frac{\cos(ku)}{u}\,\mathrm{d}u = \frac{\mathrm{Ci}(ku)}{\ln b}\bigg|_{u=1/b}^{1} = \frac{1}{\ln b}\left[\mathrm{Ci}(k) - \mathrm{Ci}\left(\frac{k}{b}\right)\right].$$

Since $k \geq b > 1$, applying $|\mathrm{Ci}(z)| \leq 1/z$ yields

$$|I_k| \leq \frac{1}{\ln b}\left(|\mathrm{Ci}(k)| + \left|\mathrm{Ci}\left(\frac{k}{b}\right)\right|\right) \leq \frac{1}{\ln b}\left(\frac{1}{k} + \frac{b}{k}\right) = \frac{b+1}{k\ln b}.$$

**2. Bounding $S_L(\tau)$.** By assumption, $\lambda(L) \to \infty$ and $\lambda(L)\ln L = o(e^{\rho\min\{1,\beta\}\lambda(L)})$ as $L \to \infty$. Hence $\tau \to \infty$ and $\tau\ln L = o(e^{\tau})$ for both $\tau \in \{\rho\lambda, \rho\beta\lambda\}$. Decompose the partial sum as

$$S_L(\tau) = \sum_{k=0}^{L-1}1 + \sum_{k=0}^{L-1}\left(e^{\tau I_k} - 1\right) = L + (e^{\tau} - 1) + \sum_{k=1}^{L-1}\left(e^{\tau I_k} - 1\right).$$

We bound the remaining sum by splitting the index set at $k = \lfloor\tau\rfloor$. Recall that, for any $k \in \mathbb{N}$,

$$I_k = \frac{1}{\ln b}\int_{1/b}^{1}\frac{\cos(ku)}{u}\,\mathrm{d}u.$$

Thus $I_0 = 1$ and $|I_k| < 1$ for all $k \geq 1$. Since $I_k \to 0$ as $k \to \infty$, there exists

$$c := \max\left\{0,\ \sup_{k\geq 1}I_k\right\} \in [0,1)$$

such that $I_k \leq c$ for all $k \geq 1$. For $1 \leq k \leq \lfloor\tau\rfloor$, we have

$$\sum_{k=1}^{\lfloor\tau\rfloor}|e^{\tau I_k} - 1| \leq \sum_{k=1}^{\lfloor\tau\rfloor}\max\left\{e^{\tau I_k}, 1\right\} \leq \tau e^{\tau c} = o(e^{\tau}) \quad \text{as } L \to \infty.$$

For $\lfloor\tau\rfloor + 1 \leq k \leq L - 1$, the bound on $I_k$ gives

$$|\tau I_k| \leq \frac{\tau(b+1)}{k\ln b} \leq \frac{b+1}{\ln b}.$$

Using $|e^x - 1| \leq |x|e^{|x|}$ for all $x \in \mathbb{R}$, we obtain

$$|e^{\tau I_k} - 1| = O(|\tau I_k|) = O\left(\frac{\tau}{k}\right) \quad \text{as } L \to \infty.$$

Therefore,

$$\sum_{k=\lfloor\tau\rfloor+1}^{L-1}|e^{\tau I_k} - 1| \leq \sum_{k=\lfloor\tau\rfloor+1}^{L-1}O\left(\frac{\tau}{k}\right) = O\left(\tau\ln\left(\frac{L}{\tau}\right)\right) \leq O(\tau\ln L) = o(e^{\tau}) \quad \text{as } L \to \infty.$$

Combining these estimates yields $S_L(\tau) = L + e^{\tau} + o(e^{\tau})$ as $L \to \infty$.

**3.** It remains to verify Eq. (58). Using $Z(\boldsymbol{\mu};\lambda) = S_L(\rho\lambda) = L + e^{\rho\lambda} + o(e^{\rho\lambda})$ and $Z(\boldsymbol{\mu};\beta\lambda) = S_L(\rho\beta\lambda) = $

$L + e^{\rho\beta\lambda} + o(e^{\rho\beta\lambda})$ as $L \to \infty$, we obtain

$$
\begin{aligned}
\mathcal{E}_\beta(\mathring{\boldsymbol{g}}) &= \frac{Z(\boldsymbol{\mu};\lambda)^{\frac{\beta}{\beta-1}}}{Z(\boldsymbol{\mu};\beta\lambda)^{\frac{1}{\beta-1}}} \\
&= \left( \frac{(L + e^{\rho\lambda} + o(e^{\rho\lambda}))^\beta}{L + e^{\rho\beta\lambda} + o(e^{\rho\beta\lambda})} \right)^{\frac{1}{\beta-1}} \\
&= L \left( \frac{(1 + e^{\rho\lambda}/L + o(e^{\rho\lambda}/L))^\beta}{1 + e^{\rho\beta\lambda}/L + o(e^{\rho\beta\lambda}/L)} \right)^{\frac{1}{\beta-1}} \\
&= L \left( \left( 1 + \beta\frac{e^{\rho\lambda}}{L} + o\left(\frac{e^{\rho\lambda}}{L}\right) \right) \left( 1 - \frac{e^{\rho\beta\lambda}}{L} + o\left(\frac{e^{\rho\beta\lambda}}{L}\right) \right) \right)^{\frac{1}{\beta-1}} \\
&= L \left( 1 + \frac{\beta e^{\rho\lambda} - e^{\rho\beta\lambda}}{L} + o\left(\frac{e^{\rho\max\{1,\beta\}\lambda}}{L}\right) \right)^{\frac{1}{\beta-1}} \\
&= L \left( 1 + \frac{\beta e^{\rho\lambda} - e^{\rho\beta\lambda}}{(\beta-1)L} + o\left(\frac{e^{\rho\max\{1,\beta\}\lambda}}{L}\right) \right) \\
&= L + O\left( e^{\rho\max\{1,\beta\}\lambda} \right) \quad \text{as } L \to \infty,
\end{aligned}
$$

which completes the proof. $\qquad\square$

# G. Detailed Setting

## G.1. Pretraining

**Hyperparameters.** Key hyperparameters are as follows:

- **Tokenizer:** LLaMA-3 tokenizer.

- **Data:** FineWeb-Edu-100B training split.

- **Training:** sequence/context length 4096; batch size 128. We train for 100B tokens, which corresponds to roughly 195k steps.

- **Optimizer:** AdamW with $\epsilon = 10^{-8}, \beta_1 = 0.8, \beta_2 = 0.95$ and base learning rate $3 \times 10^{-4}$. For the AdaFreq module, we use a higher learning rate of $3 \times 10^{-3}$ due to its larger parameter magnitudes.

- **Scheduler:** Warmup-Stable-Decay (WSD) with linear decay, using 500 warmup steps and applying linear decay over the final $10\%$ of training; the minimum learning-rate ratio is set to 0.1.

- We initialize AdaRoPE with default RoPE base b = 10000. We set $L_{\text{ref}} = 64$ in AdaScale.

**Model Configuration** We test four backbones with the following core architectural settings:

- **LLaMA-430M:** 24 layers, hidden size 1024, 16 attention heads (MHA), MLP size 2816, RMSNorm $\epsilon = 10^{-6}$, RoPE base $\theta = 10,000$, vocab size 128,256.

- **LLaMA-1.3B:** 24 layers, hidden size 2048, 32 attention heads with 16 KV heads (GQA), MLP size 5632, RMSNorm $\epsilon = 10^{-6}$, RoPE base $\theta = 10,000$, vocab size 128,256.

- **OLMoE-2.7B (0.4B activated):** 24 layers, hidden size 1024, 16 attention heads (MHA), MoE MLP with 32 experts and top-4 routing, RMSNorm $\epsilon = 10^{-5}$, RoPE base $\theta = 10,000$, vocab size 128,256; we use auxiliary router losses with coefficient 0.01 and $z$-loss coefficient 0.001.

- **Qwen3 Gated Attention-430M:** 24 layers, hidden size 1024 with 8 attention heads (MHA), gated attention outputs with QK normalization enabled, RMSNorm $\epsilon = 10^{-6}$, RoPE base $\theta = 10,000$, vocab size 128,256.

## G.2. Context Extension

**Hyperparameters.** Key hyperparameters are as follows:

- **Training:** sequence/context length 65536; batch size 8.

- **Optimizer:** AdamW with $\epsilon = 10^{-8}, \beta_1 = 0.8, \beta_2 = 0.95$. For the AdaFreq module, we use a learning rate of $3 \times 10^{-2}$. For the AdaScale module, we use a learning rate of $3 \times 10^{-3}$. For others, we use a learning rate of $3 \times 10^{-4}$

- **Scheduler:** WSD with linear decay, using 20 warmup steps and applying linear decay over the final $10\%$ of training; the minimum learning-rate ratio is set to 0.1. For Stage 1, we train for 5 epochs. For Stage 2, we train for 2 epochs.

- We initialize AdaRoPE with YaRN. We set $L_{\text{ref}} = 8192$ in AdaScale.

- LoRA Rank $r = 16$, LoRA alpha $\alpha = 16$, apply LoRA to q and k.

