# OpenReview forum: "AdaRoPE: Not All Attention Heads Should Rotate and Scale Equally"
_ICML.cc/2026/Conference — ICML 2026 regular_

### Official Review · Reviewer_DWRk · 2026-02-21

**Soundness:** 3
**Presentation:** 3
**Significance:** 2
**Originality:** 3
**Overall Recommendation:** 4
**Confidence:** 4

**Summary:**

This paper argues that standard RoPE is overly uniform: all attention heads share the same frequency schedule and scaling behavior, despite known head specialization. The authors propose **AdaRoPE**, which adds (1) head-wise, learnable rotary frequencies (**AdaFreq**) and (2) head-wise, length-aware attention temperature scaling (**AdaScale**). The method is presented as a drop-in RoPE replacement with negligible parameter/runtime overhead. Empirically, the paper reports consistent gains in pretraining quality across multiple backbones (up to 2.7B, trained on 100B tokens), and improved long-context extension (8k→64k) compared to YaRN in both extrapolation and continued pretraining settings.

**Compliance With Llm Reviewing Policy:**

Affirmed.

**Final Justification:**

The author solved my concerns.

**Key Questions For Authors:**

1. Can you add direct comparisons with stronger recent long-context baselines (e.g., LongRoPE/LongRoPE2, Ms-PoE, MoICE-style methods) under matched training/token budgets? If AdaRoPE remains competitive, I would likely raise my recommendation.
2. Can you report multi-seed results (mean ± std) and statistical significance for Table 1–4? Stable gains would increase confidence in the claimed improvements.
3. Can you broaden long-context evaluation beyond current retrieval-heavy settings (e.g., more diverse long-context reasoning/QA benchmarks)? Stronger breadth would materially improve the paper’s significance.
4. Can you provide explicit runtime metrics (tokens/s, latency, peak memory) for RoPE vs YaRN vs AdaRoPE at inference and during extension training? This is important to validate the “negligible overhead” claim.
5. In the 8B extension setting, can you provide more detailed component ablations (AdaFreq-only, AdaScale-only, shared-vs-head-wise) under the same protocol? This would clarify which component drives gains at larger scale.

**Limitations:**

The paper includes some discussion, but it would benefit from a clearer treatment of: (i) missing stronger baseline comparisons, (ii) benchmark coverage limits, and (iii) sensitivity to initialization/training protocol choices.

**Strengths And Weaknesses:**

### Strengths
- **Clear motivation and design fit:** The central claim (head heterogeneity) is well aligned with the proposed per-head frequency and scaling mechanisms.
- **Broad empirical coverage:** Pretraining experiments span multiple architectures/scales; context-extension experiments include both extrapolation and continued pretraining.
- **Useful interpretability analyses:** Effective-length and frequency-use analyses support the intuition behind head specialization.

### Weaknesses
- **Baseline gap:** Long-context comparisons focus mainly on YaRN; comparisons against stronger recent methods (e.g., LongRoPE/LongRoPE2, Ms-PoE, MoICE-like head-adaptive methods) are missing.
- **Evaluation scope for long context is limited:** Results are mainly on RULER-style retrieval/general NLU; broader long-context stress tests would strengthen claims of robustness.
- **Statistical confidence is underreported:** No clear multi-seed variance/significance reporting for key tables.
- **Efficiency claims are not fully quantified:** “Negligible overhead” is plausible, but wall-clock throughput/latency/memory measurements are not thoroughly reported.
- **Theory-to-practice gap:** Theoretical results are built on simplified settings; practical implications are suggestive rather than definitive.

---

> ### Author Rebuttal · Authors · 2026-03-31
>
> We thank Reviewer DWRk for the detailed review and for the positive assessments of the clear motivation, broad empirical coverage, and interpretability analyses. We address each concern directly.
>
> (1) Stronger baselines. We compare all methods under the same context extrapolation protocol on SmolLM-1.7B with a target length of 64K. Dynamic NTK, YaRN, and Ms-PoE are training-free; LongRoPE and LongRoPE-2 use an equivalent FLOPs budget for evolutionary search; MoICE and AdaRoPE are both trained on the same 100-sample corpus with matched compute.
>
> |Method|Head-wise?|Frequency Flexibility|Scaling Flexibility|Parameter Source|NLU|RULER Avg|
> |-|-|-|-|-|-|-|
> |Dynamic NTK|No|Global only|None|Analytic rule|62.47|17.42|
> |YaRN|No|Global only|Global only|Heuristic rule|62.56|17.32|
> |Ms-PoE|Yes|Static per-head|None|Heuristic rule|62.58|22.31|
> |LongRoPE|No|Per-dimension (shared across heads)|None|Evolutionary search|62.73|20.88|
> |LongRoPE-2|No|Per-dimension (hybrid, shared across heads)|None|Search|62.41|21.57|
> |MoICE|Yes|Discrete, token-wise (via routing)|None|Router learning|62.66|20.36|
> |**AdaRoPE**|**Yes**|**Continuous, head-specific + dimension-specific**|**Continuous, head-specific**|**Gradient learning**|**62.73**|**22.57**|
>
> Compared under this matched protocol, AdaRoPE attains the best RULER average while preserving competitive NLU.
>
> (2) Statistical confidence. We agree that multi-seed reporting would further strengthen the empirical results, but running repeated full pretraining experiments is beyond the scope of the rebuttal budget. That said, the improvements of AdaRoPE are consistently observed across a wide range of settings, including different architectures (MHA, GQA, MoE, Gated Attention) and both pretraining and context-extension regimes (extrapolation and long-context continued pretraining). The same trend appearing across these substantially different setups suggests that the gains are systematic rather than tied to a particular run or configuration. We will include multi-seed results in the camera-ready version to further validate statistical stability.
>
> (3) Broader long-context evaluation. We agree that RULER/PG19 do not fully cover realistic long-context reasoning. Due to the limited rebuttal time, we additionally evaluated the stronger LLaMA-3-8B model on several HELMET sub-tasks at 64K context length, under both the context extrapolation and long-context continued pretraining settings.
>
> **Context extrapolation (64K, LLaMA-3-8B):**
>
> |Method|Recall EM|RAG F1|Rerank MRR|ICL EM|
> |-|-:|-:|-:|-:|
> |YaRN|18.50|46.42|17.62|64.40|
> |AdaRoPE|**28.50**|**52.76**|**17.95**|**80.50**|
>
> **Long-context continued pretraining (64K, LLaMA-3-8B):**
>
> |Method|Recall EM|RAG F1|Rerank MRR|ICL EM|
> |-|-:|-:|-:|-:|
> |YaRN+LoRA|48.00|**57.82**|20.60|81.90|
> |AdaRoPE+Joint|**53.50**|55.13|**31.83**|**82.50**|
>
> These results provide evidence beyond retrieval-heavy RULER/PG19: AdaRoPE improves substantially on retrieval-style recall and in-context learning, and remains competitive on RAG-style generation and reranking under realistic long-context evaluation.
>
> (4) Efficiency. We additionally measured decoding throughput on a single H100 using LLaMA-430M, with sequence length 8192, generation length 128, batch size 1, averaged over 100 sequences. We also report training throughput in iterations/sec, measured at sequence length 4096, per-device batch size 2, on 8 H100 GPUs. The results are:
>
> |Method|Prefill speed (tok/s)|Decode speed (tok/s)|Inference peak mem|Training iter/s|Training peak mem|
> |-|-:|-:|-:|-:|-:|
> |RoPE|137237.34|36.18|2230M|2.6465|11.07G|
> |YaRN|137148.00 (-0.06%)|36.19 (+0.03%)|2230M|2.6463 (-0.01%)|11.07G|
> |AdaRoPE|136412.21 (-0.60%)|36.06 (-0.33%)|2261M|2.6429 (-0.14%)|11.19G|
>
> The overhead remains negligible because AdaFreq preserves the same rotary compute pattern as RoPE, and AdaScale is only a per-head scalar applied to the query.
>
> (5) Llama3-8B ablations. We additionally report the LLaMA-3-8B results below using only NLU and mean RULER performance.
>
> **Extrapolation (LLaMA-3-8B):**
>
> |Method|NLU|RULER Avg|
> |-|-:|-:|
> |AdaRoPE|63.05|70.59|
> |w/o AdaFreq|63.25|65.78|
> |w/o AdaScale|63.67|62.12|
> |Share Freq.|63.09|64.87|
> |YaRN|62.53|56.63|
>
> **Long-context continued pretraining (LLaMA-3-8B):**
>
> |Method|NLU|RULER Avg|
> |-|-:|-:|
> |AdaRoPE|63.55|78.02|
> |w/o AdaFreq|63.48|77.31|
> |w/o AdaScale|63.67|75.33|
> |Share Freq.|63.91|76.27|
> |LoRA-Only|63.99|74.75|

---

> > ### Author Rebuttal · Reviewer_DWRk · 2026-04-01
> >
> > The author solved my concerns.

---

> > > ### Author Response · Authors · 2026-04-01
> > >
> > > Thank you for your positive reconsideration of our paper. We truly appreciate the time and effort you have devoted to reviewing our work.

---

### Official Review · Reviewer_qyrv · 2026-03-11

**Soundness:** 3
**Presentation:** 3
**Significance:** 3
**Originality:** 3
**Overall Recommendation:** 5
**Confidence:** 5

**Summary:**

The classic RoPE encodes positional information by rotating query and key vectors using a fixed, shared frequency schedule across all attention heads. This paper introduces AdaRoPE arguing that attention heads are functionally heterogeneous, we have in fact local heads, retrieval heads, and global aggregation heads that work at different spatial scales and so they need distinct frequency regimes and distinct attention temperature profiles.
The paper studies a combination of synthetic theoretical analysis (the Needle Retrieval from Window task), empirical observation of sparse and window-dependent frequency utilization in single-layer transformers, and direct measurement of effective sequence length and dimension activation patterns in LLaMA-3-8B.
The proposed method, AdaRoPE, switch the fixed frequency table with learnable per-head log-frequencies (AdaFreq) and introduces a head-specific, length-dependent inverse temperature (AdaScale) to counteract attention dilution heterogeneously across heads. Both components are very lightweight, and both are designed as drop-in replacements fully compatible with FlashAttention and KV caching.
For context extension, the best result is that fine-tuning only the AdaRoPE parameters on 100 samples achieves dramatically better long context performance than YaRN on LLaMA 8B at 64k tokens, while better preserving short-context capability.

**Compliance With Llm Reviewing Policy:**

Affirmed.

**Final Justification:**

In the rebuttal the authors fully answered to my questions.

**Key Questions For Authors:**

- Do the learned frequencies actually recover interpretable head specialization?
- Do you think that AdaRoPE could have some interaction with the attention sink phenomenon? (Attention sinks are crelated to the low frequency, near-identity regime of RoPE at large relative distances. If AdaFreq allows some heads to learn near-zero frequencies (effectively approaching NoPE locally), does this create, suppress, or redistribute attention sinks?)

**Limitations:**

Some of the limitations are briefly mentioned (like the GQA related ones) but the paper would benefit from a proper limitation section.

**Strengths And Weaknesses:**

Strengths:
- I agre that uniform frequency assignment could be not ideal and lead to sparse and window-size-dependent dimension usage.
- The effective sequence length metric $E_{\beta}(w)$ is a nice way to formally distinguish retrieval vs aggregation heads.
- I think that achieving meaningful context extension by fine tuning only the RoPE parameters on 100 samples is really great.
- The ablations are honest and well studied.


Weaknsses:
- The paper is missing a theoretical explanation of how heads organize themselves at scale, or whether the learned frequencies actually recover the functional specialization story (local vs. global heads). The connection between the toy theory and actual LLaMA heads isn't strong enough. About this I suggest this paper: https://aclanthology.org/2025.acl-long.303/
- The log-polynomial schedule $\lambda(L) = (1/\tau) \cdot \ln(1 + L/L_{ref})^{\gamma}$ is motivated post-hoc by theorems that basically just require monotone sublinear growth. And there isn't an analysis of how sensitive results are to this specific functional form or to alternatives.
- RULER and PG19 is a reasonable test but it's also quite limited. Maybe you should add a proper evaluation on realistic long-context tasks.
- In GQA architectures the method learns frequencies at the KV group level instead of per head, and that contradicts the head heterogeneity motivation I think. Since most of large modern models use GQA, this is a practical limitation that the paper acknowledges only briefly without analyzing its impact on performance or the theoretical justification.

---

> ### Author Rebuttal · Authors · 2026-03-31
>
> We thank Reviewer qyrv for the expert, detailed, and constructive review. The positive assessments of the effective sequence length metric, the 100-sample context extension result, and the ablation quality are greatly appreciated. We address each concern directly.
>
> (1) Head specialization at scale. We appreciate this question and note that we have included several analyses of head-level functional specialization. Specifically, Fig. 5 (right) shows the AdaScale exponent $\gamma$ negatively correlates with effective length (larger $\gamma$ attends more locally); Fig. 7 shows learned AdaFreq aligns with utilized frequencies. This is further supported by Fig. 11, where AdaRoPE preserves local heads while selectively extending global ones, linking learned parameters to functional head behavior. Beyond these existing analyses, we have now additionally computed the correlation between each head's mean log-frequency (derived from the learned AdaFreq frequencies) and its attention-weighted relative distance (the average token distance each head attends to). We find a clear negative correlation ($r = -0.61$): heads whose learned frequencies are higher attend to more local tokens on average, consistent with the specialization story that different heads learn to operate at different positional scales. We will include this additional analysis and cite the paper mentioned by the reviewer in the camera-ready version.
>
> (2) Why the log-polynomial schedule? We agree this deserves clearer justification. The key point is that the head-wise scaling coefficient $$\lambda^{(h)}(L):=\frac{1}{\tau^{(h)}} \left[\ln\left(1+\frac{\max(\{L,L\_{\mathrm{ref}}\})}{L\_{\mathrm{ref}}}\right)\right]^{\gamma^{(h)}}$$
> directly multiplies the attention logit before the softmax normalization, so it effectively appears inside the exponential of the attention weights. For this reason, the growth with $L$ must be carefully controlled: if the scale grows too aggressively, the logits become numerically unstable and attention over-sharpens; if it grows too slowly, it does not sufficiently correct attention dilution at long lengths. The logarithm is therefore introduced mainly to keep the growth numerically stable, while the learnable exponent $\gamma^{(h)}$ still allows different heads to adapt how strongly they depend on length. To test whether this form matters in practice, we also tried several alternative scale functions in the LLaMA-430M pretraining setting:
>
> |Scale function|Formula|NLU Avg|Loss|
> |-|-|-:|-:|
> |Fixed Log Scaling (non-learnable)|$\ln(1+L/L\_{\mathrm{ref}})$|50.58|2.687|
> |Sqrt scaling|$\sqrt{aL}$, with learnable $a$|--|diverged|
> |Power scaling|$(aL)^b$, with learnable $a,b$|50.73|2.690|
> |AdaRoPE log-polynomial|$\lambda^{(h)}(L)=\frac{1}{\tau^{(h)}}\left[\ln(1+\max(\{L,L\_{\mathrm{ref}})\}/L\_{\mathrm{ref}})\right]^{\gamma^{(h)}}$|**51.23**|**2.663**|
>
> These results support this motivation: the proposed log-polynomial schedule is more stable than more aggressive alternatives such as $\sqrt{aL}$, and empirically better than the simpler fixed-log or power-law variants.
>
> (3) Evaluation breadth. We additionally evaluated the stronger LLaMA-3-8B model on several HELMET sub-tasks at 64K context length, under both the context extrapolation and long-context continued pretraining settings.
>
> **Context extrapolation (64K, LLaMA-3-8B):**
>
> |Method|Recall EM|RAG F1|Rerank MRR|ICL EM|
> |-|-:|-:|-:|-:|
> |YaRN|18.50|46.42|17.62|64.40|
> |AdaRoPE|**28.50**|**52.76**|**17.95**|**80.50**|
>
> **Long-context continued pretraining (64K, LLaMA-3-8B):**
>
> |Method|Recall EM|RAG F1|Rerank MRR|ICL EM|
> |-|-:|-:|-:|-:|
> |YaRN+LoRA|48.00|**57.82**|20.60|81.90|
> |AdaRoPE+Joint|**53.50**|55.13|**31.83**|**82.50**|
>
> These results provide evidence beyond retrieval-heavy RULER/PG19: AdaRoPE improves substantially on retrieval-style recall and in-context learning, and remains competitive on RAG-style generation and reranking under realistic long-context evaluation.
>
> (4) GQA limitation. While group-sharing partially relaxes the ideal per-head frequency design, the empirical CoV reduction (0.92-> 0.67 on LLaMA-1.3B) and consistent gains on GQA backbones show that the benefit survives this practical constraint. We refer the reviewer to the detailed response to Reviewer KUJD Q4.
>
> (5) Attention Sinks. We define early-prefix sink mass at scale $k$ as $$\text{mass@}k = \mathbb{E}\_q\left[\sum\_{j=1}^{k} A(q,j)\right]$$
> that is, the average attention mass assigned by the last 128 queries to the first $k$ prefix tokens in a sequence of 4096. In a paired LLaMA-430M experiment, we obtain:
>
> |Metric|RoPE|AdaRoPE|Delta|
> |-|-:|-:|-:|
> |mass@1|0.1186|0.1790|+0.0604|
>
> So, in this setting, AdaRoPE increases early-prefix sink mass rather than suppressing it. We also find a clear layer dependence: the increase is mainly concentrated in middle layers, while most early and late layers show little change or a slight decrease.

---

> > ### Author Rebuttal · Reviewer_qyrv · 2026-04-02
> >
> > The authors fully responded to my questions, I'm gonna raise my score (4 -> 5)

---

> > > ### Author Response · Authors · 2026-04-03
> > >
> > > Thank you very much for your thoughtful, expert, and constructive review, and for your positive reconsideration of our paper. Your feedback gave us many valuable ideas for strengthening both the analysis and the presentation, and we truly appreciate the time and care you devoted to the review.

---

### Official Review · Reviewer_KUJD · 2026-03-11

**Soundness:** 3
**Presentation:** 3
**Significance:** 3
**Originality:** 3
**Overall Recommendation:** 4
**Confidence:** 2

**Summary:**

This paper proposes AdaRoPE, a lightweight and learnable extension to Rotary Position Embeddings (RoPE) that introduces head-specific rotation frequencies (AdaFreq) and length-aware attention scaling factors (AdaScale).

**Compliance With Llm Reviewing Policy:**

Affirmed.

**Final Justification:**

The author solved my concerns.

**Key Questions For Authors:**

1. The paper notes that for GQA, frequencies are shared across groups. It would be beneficial to see if this "group-sharing" leads to the same "dimension underutilization" that standard RoPE suffers from.
2. AdaScale assumes a monotonic schedule in $L$. Given the "lost-in-the-middle" phenomenon, it’s worth questioning if non-monotonic scaling could provide further benefits.

**Limitations:**

Yes

**Strengths And Weaknesses:**

Strengths:

The paper provides a rigorous theoretical foundation, using Fourier analysis to prove that different target window sizes (local vs. global) necessitate distinct frequency mass concentrations.

Weaknesses

1. While the paper compares against YaRN, PROPE, and Alibi, it lacks comparison with other modern training-free or dynamic scaling methods like Dynamic NTK-RoPE or other head-wise multi-scale variants like Ms-PoE in all settings.
2. Although 2.7B parameters is respectable, it remains to be seen if the benefits of head-specific RoPE parameters persist or become redundant in larger models (e.g., 7B+), where head redundancy might already be present.
3. The theoretical analysis is based on "simplified retrieval tasks" (NRW). It is unclear how perfectly these scenarios map to the complexities of natural language reasoning.
4. The paper notes that for GQA, frequencies are shared across groups. It would be beneficial to see if this "group-sharing" leads to the same "dimension underutilization" that standard RoPE suffers from.

---

> ### Author Rebuttal · Authors · 2026-03-31
>
> We thank Reviewer KUJD for recognizing the paper's rigorous theoretical foundation and for the detailed, specific concerns. We address each directly.
>
> (1) Relation to other methods. We compare all methods under the same context extrapolation protocol on SmolLM-1.7B with a target length of 64K. Dynamic NTK, YaRN, and Ms-PoE are training-free; LongRoPE and LongRoPE-2 use an equivalent FLOPS budget for evolutionary search; MoICE and AdaRoPE are both trained on the same 100-sample corpus with matched compute.
>
> |Method|Head-wise?|Frequency Flexibility|Scaling Flexibility|Parameter Source|NLU|RULER Avg|
> |-|-|-|-|-|-|-|
> |Dynamic NTK|No|Global only|None|Analytic rule|62.47|17.42|
> |YaRN|No|Global only|Global only|Heuristic rule|62.56|17.32|
> |Ms-PoE|Yes|Static per-head|None|Heuristic rule|62.58|22.31|
> |LongRoPE|No|Per-dimension (shared across heads)|None|Evolutionary search|62.73|20.88|
> |LongRoPE-2|No|Per-dimension (hybrid, shared across heads)|None|Search|62.41|21.57|
> |MoICE|Yes|Discrete, token-wise (via routing)|None|Router learning|62.66|20.36|
> |**AdaRoPE**|**Yes**|**Continuous, head-specific + dimension-specific**|**Continuous, head-specific**|**Gradient learning**|**62.73**|**22.57**|
>
> (2) Larger models and head redundancy. We agree that larger-scale evidence is important. To address this, we have extended our OLMoE experiment to **7.2B parameters** (1.2B active) and pretrained on 10B tokens from FineWeb-Edu (limited by rebuttal time):
>
> |Configuration|NLU Avg|Loss|
> |-|-|-|
> |RoPE (baseline)|58.02|2.614|
> |AdaRoPE|**59.43 (+1.41)**|**2.599 (-0.015)**|
>
> The advantage of AdaRoPE is preserved, suggesting that head-specific positional adaptation does not become redundant as model size grows. We also refer the reviewer to our LLaMA-3-8B context extension results (Table 2), where AdaRoPE substantially outperforms YaRN, providing further evidence that the method scales well.
>
> (3) Simplified theory. We do not intend the NRW analysis as a full model of natural language reasoning. Its role is to isolate one mechanism: different tasks require different frequency bands. We then validate this mechanism empirically in real LLMs through head-wise effective-length/frequency analyses and the downstream long-context gains.
>
> (4) GQA group sharing. In GQA backbones we share AdaFreq within each KV group for implementation compatibility, but AdaScale remains per-head. Empirically, AdaRoPE still improves on GQA models (e.g., LLaMA-1.3B in pretraining and LLaMA-3-8B in extension), suggesting that group-sharing does not collapse the benefit in practice. To make the utilization effect explicit, we compare the median CoV (A lower CoV indicates more uniform and better dimension utilization) across heads below:
>
> |Setting|RoPE median CoV|AdaRoPE median CoV|
> |-|-:|-:|
> |LLaMA-430M (MHA, Figure 3)|0.81|0.63|
> |LLaMA-1.3B (GQA)|0.92|0.67|
>
> The dimension-underutilization problem persists under GQA (median CoV = 0.92 for LLaMA-1.3B), and AdaRoPE still substantially reduces it (0.92 → 0.67), demonstrating that group-shared frequencies do not prevent AdaRoPE from improving utilization in practice.
>
> (5) Why a monotone schedule? Our clarification is that AdaScale is applied as a head-specific scalar on the query, and the scalar depends only on the available context length $L$. Therefore, AdaScale is designed to correct length-induced attention dilution as the sequence gets longer; it is not a position-dependent mechanism over token indices within the sequence. The "lost-in-the-middle" phenomenon is about where a token appears inside the context, whereas AdaScale only changes the overall temperature seen by a head at a given context length. For this reason, allowing a non-monotonic schedule in $L$ would not directly address lost-in-the-middle. Exploring position-dependent or non-monotonic scaling mechanisms is interesting future work, but it is outside the scope of the current design.

---

> > ### Author Rebuttal · Reviewer_KUJD · 2026-04-01
> >
> > Thank you so much for the rebuttal. My comments have been addressed.

---

> > > ### Author Response · Authors · 2026-04-01
> > >
> > > Thank you for your positive reconsideration of our paper. We truly appreciate the time and effort you have devoted to reviewing our work.

---

### Official Review · Reviewer_Cxbq · 2026-03-11

**Soundness:** 4
**Presentation:** 4
**Significance:** 3
**Originality:** 3
**Overall Recommendation:** 4
**Confidence:** 4

**Summary:**

This paper proposes AdaRoPE, an extension of RoPE that allows per-head learnable rotation frequencies and scaling factors. The key motivation is that different attention heads serve different roles (e.g., local vs global aggregation), and therefore should not share the same positional frequency schedule or scaling.

**Compliance With Llm Reviewing Policy:**

Affirmed.

**Final Justification:**

The author solved my concerns.

**Key Questions For Authors:**

Questions:

(1) What is the effect of adaptive frequency or scaling only? I know they can enhance the model performance significantly if these two are considered jointly. What if only one of them is considered and what happened?

(2) The motivation of the method is that different heads require different positional frequencies. However, for different heads, their Q, K, V matrices are also different. I think they can still learn some head-wise frequency and scaling from these matrices. So, in this sense, whether the additional learnable scaling and frequency are not necessary?

(3) How sensitive is AdaRoPE to initialization of frequencies?

(4) Does AdaRoPE improve inference stability for extrapolation beyond training length? In general RoPE, it performs well in length extrapolation. What is the inference performance of AdaRoPE when inference length exceed training length? Does it outperforms other positional encoding methods? If yes, why AdaRoPE has better generalization in terms of length extrapolation, compared to RoPE? Do you have any theoretical or intuitive explanations?

**Limitations:**

Yes

**Strengths And Weaknesses:**

**Strength:**

(1) The paper provides empirical observations that different heads utilize RoPE frequency bands differently.

(2) A theoretical perspective on connecting frequency bands to effective sequence length and head specialization is discussed in the appendix.

(3) The experiments on pre-training and extrapolation are convincing and noticeable performance improvements can be observed across several settings.

(4) The proposed method is simple and practical. It can be easily implemented and seamlessly incorporated into most LLM architectures. The motivation behind the method is also clearly presented and intuitive.

(5) The theoretical analysis, although in appendix, is interesting.


**Weaknesses:**

(1) The proposed method can be viewed as a combination or extension of several existing ideas applied to RoPE. Compared with the broader literature on positional encoding methods, the level of novelty may be somewhat limited.

(2) The theoretical analysis is restricted to NRW problem.

---

> ### Author Rebuttal · Authors · 2026-03-31
>
> We thank the reviewer for the positive assessment. We address each concern below.
>
> (1) Limited Novelty as Combination of Existing Ideas. We acknowledge that AdaRoPE builds on prior work, but believe it goes beyond a simple combination. Theoretically, both components are grounded in formal analysis (Fourier-analytic necessity proof for AdaFreq; information-theoretic entropy preservation for AdaScale), not ad-hoc design. Practically, AdaRoPE is — to our knowledge — the only method that is simultaneously **(i)** head-specific and dimension-specific in frequency, **(ii)** head-specific and length-dependent in scaling, and **(iii)** effective in both pretraining and context extension with negligible overhead. We refer the reviewer to the comparison table in our response to Reviewer KUJD Q1 (and Reviewer DWRk Q1), which contrasts AdaRoPE with other methods.
>
> (2) Theoretical Analysis Restricted to NRW Problem. The NRW simplification is a deliberate choice for analytical tractability via Fourier analysis — it does not claim natural language is equivalent to NRW, but establishes a necessity condition: even in this simple setting, different tasks already require different frequencies, underscoring the need for head-specific frequency allocation in more complex scenarios.
>
> (3) Effect of each component. We agree a clearer component-wise presentation would strengthen the paper. Several ablations are already included: Table 1 reports pretraining ablations and Tables 2 & 4 report context extension ablations. We have now completed the missing AdaScale-only (W/O AdaFreq) configuration in pretraining, and summarize the key ablations below:
>
> Settings: **PT** = Pretraining (LLaMA 430M, Table 1); **Ext** = Context Extension (SmolLM 1.7B, Table 2); **CPT** = Long-Context Continue Pretrain (SmolLM 1.7B, Table 4).
>
> |Configuration|PT NLU|PT Loss|Ext NLU|Ext RULER|CPT NLU|CPT RULER|
> |-|-|-|-|-|-|-|
> |Baseline (RoPE/YaRN)|50.65|2.692|62.56|17.32|62.90|29.26|
> |AdaFreq-only|50.77|2.683|**64.01**|15.75|**64.43**|33.32|
> |AdaScale-only|50.75|2.690|63.15|19.18|63.31|31.56|
> |AdaRoPE (full)|**51.23**|**2.663**|62.73|**22.57**|63.16|**37.61**|
>
>
> (4) Why Q/K/V alone are not enough. The RoPE attention logit between positions $m$ and $n$ is $s\_{m,n}^{(h)}=\sum\_f \left\| q\_f^{(h)} \right\|\left\| k\_f^{(h)} \right\|\cos\!\big(\phi\_f^{(h)} + (m-n)\theta\_f^{(h)}\big)$
> where $\left\| q\_f^{(h)} \right\|\left\| k\_f^{(h)} \right\|$ is the amplitude, $\phi\_f^{(h)}$ is the content-determined phase offset, and $\theta\_f^{(h)}$ is the rotation frequency. $W\_Q$ and $W\_K$ can change the amplitudes and phases, but cannot change the coefficient of $(m-n)$, namely $\theta\_f^{(h)}$, because that frequency schedule is fixed by the positional encoding. This is exactly the degree of freedom AdaFreq introduces.
>
> With AdaScale, the logit becomes $\tilde{s}\_{m,n}^{(h)} = \beta^{(h)}(L)\, s\_{m,n}^{(h)}$, where $\beta^{(h)}(L)$ is a head-specific inverse temperature depending on sequence length $L$. Since the same $W\_Q, W\_K$ are applied regardless of $L$, static Q/K/V projections cannot reproduce a head-specific, length-dependent temperature. Empirically, if Q/K/V were sufficient, removing AdaFreq or AdaScale would not degrade performance — yet our ablations show otherwise across all settings.
>
> (5) Initialization sensitivity. AdaFreq optimizes $\xi\_f^{(h)}$ in log-frequency space with $\theta\_f^{(h)}=\exp(\xi\_f^{(h)})$. The default initialization reproduces the standard RoPE schedule $\theta\_f = \mathrm{base}^{-2f/d}$ for a fair comparison.
>
> We tested two random initialization schemes on LLaMA-430M. Let $\beta = \log(\mathrm{base})$ (standard frequency range $[-\beta, 0]$ in log-space). We consider: (i) $\xi\_f^{(h)} \sim U(-\beta, 0)$ i.i.d., and (ii) a hierarchical variant drawing $\beta\_h \sim U(0, 2\beta)$ per head then $\xi\_f^{(h)} \sim U(-\beta\_h, 0)$:
>
> |Initialization|NLU|Loss|
> |-|-:|-:|
> |Standard RoPE init|51.23|2.663|
> |Random init (i)|51.25|2.663|
> |Random init (ii)|51.16|2.664|
>
> All three initializations converge to near-identical performance.
>
> (6) Extrapolation beyond training length. Unmodified RoPE generalizes poorly beyond the training context — out-of-distribution rotation angles cause rapid NLL growth. The standard remedy is global frequency rescaling (Dynamic NTK, YaRN, LongRoPE). AdaRoPE goes further: AdaFreq learns *head-specific* frequencies via gradient descent rather than a single global rule, and AdaScale adds a head-specific, length-dependent temperature correction. Compared to unmodified RoPE and YaRN, AdaRoPE maintains lower NLL and higher accuracy at 16k/32k/64k (Tables 2–4), because different heads can independently optimize their local-vs-global positional trade-off. This is directly visible in Figure 4 (left), where AdaRoPE's NLL curve stays flat beyond the training length while YaRN's rises.

---

> > ### Author Rebuttal · Reviewer_Cxbq · 2026-04-01
> >
> > Thank you so much for the rebuttal. My comments have been addressed. I only have one question: Does the performance benefit come from more trainable parameters? According to the scaling law, models with more parameters can achieve better performance.

---

> > > ### Author Response · Authors · 2026-04-01
> > >
> > > Thank you for this follow-up question, and for the positive assessment of our rebuttal. We agree that, in general, increased parameter count can improve performance under scaling laws. However, the additional parameter count introduced by AdaRoPE is extremely small.
> > > For example, in the LLaMA-430M backbone, AdaRoPE adds $$\Delta P= L \cdot H_{kv} \cdot \frac{d}{2} + 2LH= 13{,}056 \approx 0.013\text{M}$$ extra parameters (with $L=24$, $H=H_{kv}=16$, and $d=64$). This is only on the order of $10^{-5}$ relative to the full model size and has negligible overhead. Therefore, while parameter count can matter in general, we believe the gain here is unlikely to be explained by model size, and instead mainly comes from the improved positional parameterization itself.

---

### Decision · Program_Chairs · 2026-04-30

**Decision:**

Accept (regular)

**Comment:**

This paper proposes AdaRoPE, a head-wise adaptive rotary position encoding method with learnable frequencies and length-aware scaling for Transformers.
All reviewers give positive scores (4, 4, 5, 4). The strengths include clear motivation, solid theoretical and empirical support, lightweight design, strong long-context performance, comprehensive ablations. The weaknesses include minor novelty concerns, limited theoretical scope, GQA practical constraints, narrow early evaluation.

The authors fully addressed all concerns with supplementary experiments, clarifications, and broader baselines, resolving nearly all reviewer doubts.

Overall, the work provides a sound, effective, and practical improvement to positional encoding. The AC recommends acceptance.